# A Distributed Generative AI Approach for Heterogeneous Multi-Domain Environments under Data Sharing constraints

**Youssef Tawfilis** *youssef.albert@guc.edu.eg*
*The Faculty of Information Engineering and Technology*
*The German University in Cairo*

**Hossam Amer** *hossam.amer@guc.edu.eg*
*The Faculty of Media Engineering and Technology*
*The German University in Cairo*

**Minar Elaasser** *minar.elaasser@guc.edu.eg*
*The Faculty of Information Engineering and Technology*
*The German University in Cairo*

**Tallal Elshabrawy** *tallal.elshabrawy@guc.edu.eg*
*The Faculty of Information Engineering and Technology*
*The German University in Cairo*

**Reviewed on OpenReview:** *https://openreview.net/forum?id=rpbL7pfPYH*

## Abstract

Federated Learning has gained increasing attention for its ability to enable multiple nodes to collaboratively train machine learning models without sharing their raw data. At the same time, Generative AI—particularly Generative Adversarial Networks (GANs)—have achieved remarkable success across a wide range of domains, such as healthcare, security, and Image Generation. However, training generative models typically requires large datasets and significant computational resources, which are often unavailable in real-world settings. Acquiring such resources can be costly and inefficient, especially when many underutilized devices—such as IoT devices and edge devices—with varying capabilities remain idle. Moreover, obtaining large datasets is challenging due to privacy concerns and copyright restrictions, as most devices are unwilling to share their data. To address these challenges, we propose a novel approach for decentralized GAN training that enables the utilization of distributed data and underutilized, low-capability devices while not sharing data in its raw form. Our approach is designed to tackle key challenges in decentralized environments, combining KLD-weighted Clustered Federated Learning to address the issues of data heterogeneity and multi-domain datasets, with Heterogeneous U-Shaped split learning to tackle the challenge of device heterogeneity under strict data sharing constraints—ensuring that no labels or raw data, whether real or synthetic, are ever shared between nodes. Experimental results shows that our approach demonstrates consistent and significant improvements across key performance metrics, where it achieves an average 10% boost in classification metrics (up to 60% in multi-domain non-IID settings), $1.1\times$—$3\times$ higher image generation scores for the MNIST family datasets, and $2\times$—$70\times$ lower FID scores for higher resolution datasets, in much lower latency compared to several benchmarks. Our code is available at https://distributed-gen-ai.github.io/huscf-gan.github.io/.

## 1 Introduction

Generative artificial intelligence has captured global attention across every domain, including healthcare (Showrov et al., 2024),security (Lim et al., 2024), image synthesis (Lang et al., 2021), and data

Table 1: Evaluation of Distributed GAN Techniques Against Key Challenges

| Criterion | (Hardy et al., 2019) | (Rasouli et al., 2020) | (Guerraoui et al., 2020) | (Fan & Liu, 2020) | (Zhao et al., 2021) | (Zhang et al., 2021) | (Kortoçi et al., 2022) | (Li et al., 2022) | (Ekblom et al., 2022) | (Wijesinghe et al., 2024a) | (Wijesinghe et al., 2024b) | (Chiaro et al., 2023) | (Cao et al., 2022) | (Fan et al., 2024) | (Maliakel et al., 2024) | (Ma et al., 2023) | (Zhao et al., 2025) | (Zhang et al., 2023) | (Haghbin et al., 2025) | (Petch et al., 2025) | (Quan et al., 2024b) | (Quan et al., 2024a) | (Kasturi & Hota, 2023) | (Jain & Wilfred Godfrey, 2022) | (Wijesinghe et al., 2023) | HuSCF-GAN |
|---|---|---|---|---|---|---|---|---|---|---|---|---|---|---|---|---|---|---|---|---|---|---|---|---|---|---|
| Data Heterogeneity | ✓ | ✓ | ✓ |  | ✓ | ✓ |  | ✓ | ✓ | ✓ | ✓ | ✓ | ✓ | ✓ | ✓ | ✓ | ✓ | ✓ | ✓ | ✓ | ✓ | ✓ | ✓ | ✓ | ✓ | ✓ |
| Device Heterogeneity |  |  |  |  |  |  | ✓ |  |  |  |  |  |  |  |  |  |  |  |  |  |  |  |  |  |  | ✓ |
| Resource Constrained Environments | ✓ |  |  | ✓ |  |  | ✓ |  |  |  |  | ✓ |  |  |  | ✓ |  | ✓ |  |  | ✓ | ✓ |  | ✓ |  | ✓ |
| Multi-Domain |  |  |  |  |  |  |  |  |  |  |  |  |  |  |  |  |  |  |  |  |  |  |  |  | ✓ | ✓ |
| No Raw Data/Labels Shared |  | ✓ | ✓ |  |  |  |  | ✓ | ✓ | ✓ | ✓ |  |  | ✓ |  | ✓ | ✓ |  |  | ✓ | ✓ | ✓ | ✓ |  | ✓ | ✓ |

augmentation (Amer et al., 2024) . These models not only understand and analyze data but also generate entirely new content that ideally reflects the underlying distributions of their training data. However, training such models demands both massive volumes of diverse data and vast computational power (Manduchi et al., 2024; Amer et al., 2026).

Meeting these requirements is often challenging, as the majority of today's data remains siloed due to privacy, security, and proprietary concerns. These restrictions prevent clients and devices from sharing their data to a centralized entity, limiting the breadth and diversity of centralized training datasets—and, by extension, the performance of centralized models depending on them. Moreover, centralized training infrastructures—whether on-premises servers or cloud-based clusters—must be powerful enough to accommodate growing model sizes and are correspondingly costly in terms of cloud subscription fees or, if on-premises, hardware, electricity, cooling, and land costs, making scalability difficult and fees prohibitive (Abul-Fazl et al., 2025). This is especially true as the amount of computing power used to train AI models is growing at a staggering pace—doubling roughly every 3.4 months. That is far faster than the rate predicted by Moore's Law, which estimates computing capacity to double approximately every 24 months (OpenAI & Stanford HAI, 2023). Meanwhile, countless underutilized devices at the edge—smartphones, tablets, IoT devices—sit idle despite possessing significant collective computational capacity; individually. However, none can handle the workload of a full-scale generative model, such as GANs (Goodfellow et al., 2020).

Decentralized paradigms such as Federated Learning (McMahan et al., 2017) and Split Learning (Vepakomma et al., 2018) offer promising alternatives by enabling collaborative model training across distributed, privacy-preserving devices without exposing raw data or relying on centralized computational resources. Federated Learning allows multiple devices to collaboratively train a shared model without exchanging raw data. Each device trains a local copy of the model on its private dataset for several epochs, and then server collects them, aggregates them, and redistributes them. Split Learning, on the other hand, is specifically designed to address the limitations of resource-constrained devices—such as IoT devices—that are incapable of training an entire model locally. In its basic form, the model is divided into two segments: a client-side portion and a server-side portion. Training begins at the client, which sends its activations to the server to continue training.

While decentralized paradigms offer a promising solution to the problems regarding training centralized generative models, they introduce their own challenges. First, data heterogeneity—devices often hold non-IID data with varying label distributions, skewness, and dataset sizes—can destabilize the global model when participants' local distributions differ widely. Second, utilizing underutilized devices introduces the challenge of device heterogeneity in resource-constrained environments—edge devices vary in compute power and data rates, so assigning equal workloads can cause bottlenecks and slow down training. Third, devices may hold data from different domains, which can degrade performance if aggregation ignores these differences; effective methods must detect and adapt to such variation. Finally, ensuring data sharing constraints remains paramount: no device should ever share its raw (or generated) data or labels, and the training process must guarantee that sensitive information never leaves the device.

Most recent efforts on decentralized GANs have typically addressed only one or two of the aforementioned challenges, as shown in Table 1. While these approaches offer promising ideas, they fall short of providing a comprehensive, real-world solution suitable for heterogeneous environments. In this paper, we introduce Heterogeneous U-shaped Split Clustered Federated GANs (HuSCF-GANs), a novel approach that enables collaborative training in heterogeneous settings across underutilized edge and IoT devices with the support of an intermediary server—without relying on it exclusively. Although we demonstrate our approach using conditional GANs (cGANs) (Mirza & Osindero, 2014), the same concepts can be applied to any generative model.

Our approach proceeds in five stages.

1. First, we employ a genetic algorithm to determine the optimal cut points in the model for each client, based on its computational capacity and data transmission rate, with the remainder of the model hosted on the server.

2. Next, we perform Heterogeneous U-shaped Split Learning on each client's portion of the model according to these tailored cuts, sending and receiving the activations/gradients to and from the server depending on the cut layers to continue its training.

3. Every several epochs, we apply a clustering technique to the clients' activations of the Discriminator's intermediate layer hosted on the server, grouping them into domain-specific clusters.

4. Within each cluster, we execute a federated learning routine that weighs each device's parameter updates by both its dataset size and its Kullback–Leibler divergence score.

5. Finally, we test and evaluate HuSCF-GAN against alternative decentralized frameworks across multiple benchmark datasets, demonstrating its superior performance.

HuSCF-GAN successfully addresses several key challenges: **data heterogeneity**, characterized by non-IID data distributions across clients; **multi-domain data**, where clients can possess data from different domains; and **device heterogeneity**, involving variability in computational power and data transmission rates among participating clients. Importantly, all of this is achieved under strict data-sharing constraints, specifically: **no sharing of raw data**—neither real nor generated data is exchanged, with only intermediate activations and gradients being communicated; and **no sharing of labels**, thereby preserving the confidentiality of local annotations.

The proposed approach was evaluated against several baselines, including MD-GAN (Hardy et al., 2019), FedGAN (Rasouli et al., 2020), Federated Split GANs (Kortoçi et al., 2022), PFL-GAN (Wijesinghe et al., 2023), and HFL-GAN (Petch et al., 2025). HuSCF-GAN achieves up to $2.2\times$ higher image generation scores and an average 10% improvement in classification metrics (with gains of up to 50% in some test cases) while maintaining lower latency compared to other approaches, demonstrating its effectiveness without introducing significant computational or communication overhead.

The remainder of this paper is organized as follows. Section 2 provides the necessary background to understand the core concepts presented in this work. Section 3 reviews the existing literature related to distributed GANs. Section 4 details the proposed methodology of our HuSCF-GAN Approach. Section 5 describes the experimental setup, while Section 6 presents and analyzes the results. Finally, Section 8 concludes the paper and discusses potential directions for future research.

## 2 Background

### 2.1 Federated Learning

Federated Learning (FL), introduced in 2016, is a distributed learning framework that enables multiple nodes to collaboratively train machine learning models without sharing their local data as shown in Figure 1. Instead, after several training epochs, each node shares its model parameters, which are then averaged to produce a global model—effectively learning from all datasets without compromising data privacy (McMahan

Figure 1: Comparison between Federated Learning, traditional Split Learning, and U-shaped Split Learning. In Federated Learning, each client trains a local model for several epochs and then sends its model weights to a central server. The server aggregates these weights—typically by averaging—and sends the updated global model back to the clients. This process is repeated for multiple rounds. In traditional Split Learning, the model is divided into two parts: the client holds the initial segment, and the server holds the remaining part. In U-shaped Split Learning, the model is split into three segments: the client retains both the initial and final segments, while the server manages the middle segment.

et al., 2017). The overall objective in FL is to minimize the global loss function defined over the data distributed across all participating clients.

## 2.2   Split Learning

Another distributed learning paradigm is Split Learning, which in its vanilla form operates as follows: the neural network is split into two parts—the client part and the server part—where the first part resides on the client device, such as an edge node, and the other part resides on the server. These two parts work sequentially, where the client part is trained first on the data (which remains on the client), then it sends the "smashed data" (activations) to the server to complete the forward pass and calculate the loss. The process is reversed in the backward pass with the gradients being shared instead of the activations. However, in the vanilla form, the labels must be shared with the server for loss calculation. This issue is mitigated in the U-shaped variant of Split Learning, where the model is divided into three parts: the head, the tail, and the middle part. In this setup, the head and tail reside on the client so that both the labels and the raw data remain local, while the middle part resides on the server (Vepakomma et al., 2018) as shown in Figure 1.

## 2.3   Split Federated Learning

Split Federated Learning (SFL) is a distributed AI framework that combines the principles of both FL and SL, as described in Subsections 2.1 and 2.2, respectively (Thapa et al., 2022). In this framework, the model is split between each client and the server. The client-side segments of the model perform federated learning among themselves, enabling collaborative training while preserving data locality and reducing individual device workloads. The overall architecture of SFL is illustrated in Figure 2.

## 2.4   Conditional Generative Adversarial Networks

Conditional Generative Adversarial Network (cGAN) is an instance of a generative model, particularly GAN architecture, designed to enable class- or condition-specific data generation. Unlike vanilla GANs (Goodfellow et al., 2020), which generate outputs solely from a noise vector, cGANs condition both the Generator and the Discriminator on auxiliary information—typically class labels or other side information—allowing the model to produce outputs that adhere to specific categories (Mirza & Osindero, 2014).

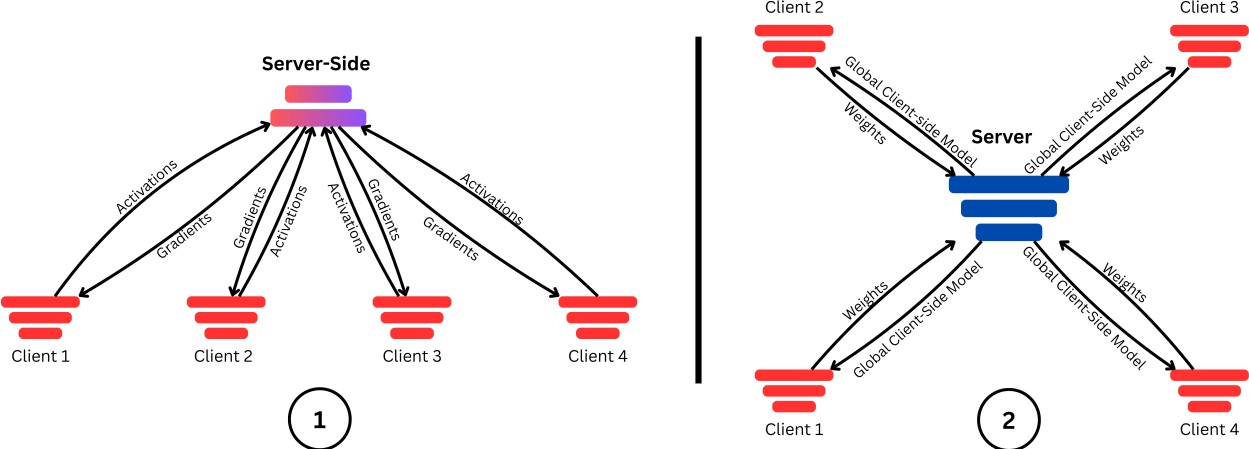

Figure 2: **SFL architecture:** In the first step (split learning), clients train their local (client-side) model segments and send the resulting activations to the server, which continues training on the server-side segments. During backpropagation, the server returns the gradients to the clients. In the second step, after several local epochs, clients send their updated client-side models to the server for federated aggregation and redistribution.

In the cGAN framework, both networks receive the conditioning variable $y$ as an additional input. The Generator $G$ learns to map a noise vector $z \sim p_z(z)$ and a condition $y$ to the data space, i.e., $G(z|y)$, while the Discriminator $D$ is trained to distinguish between real data samples paired with their true condition and synthetic samples generated by $G$.

The training process is formulated as a two-player minimax game. The objective function for a cGAN is defined as follows:

$$\min_G \max_D V(D, G) = \mathbb{E}_{x \sim p_{\text{data}}(x)}[\log D(x|y)] + \mathbb{E}_{z \sim p_z(z)}[\log(1 - D(G(z|y)|y))] \tag{1}$$

Here, $x \sim p_{\text{data}}(x)$ denotes real data samples from the true data distribution, $z \sim p_z(z)$ is the input noise vector, and $y$ is the conditioning information. Both $G$ and $D$ are explicitly conditioned on $y$, enabling class-specific synthesis and discrimination.

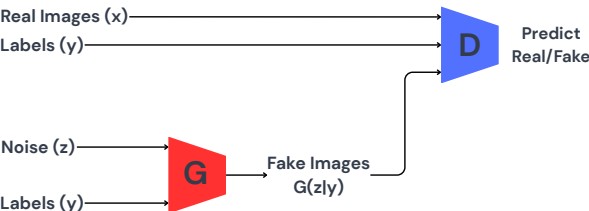

Figure 3: Architecture of a Conditional GAN (cGAN). The generator and discriminator are both conditioned on auxiliary information, such as class labels, allowing the model to generate data that adheres to specific conditions.

As illustrated in Figure 3, the incorporation of the conditioning variable allows for more controlled and targeted generation. This makes cGANs particularly effective in applications such as class-conditional image synthesis, image-to-image translation, and text-to-image generation, where the ability to direct the output is essential.

The reason for choosing a conditional GAN (cGAN) instead of a vanilla GAN as the generative model is to enable the generation of synthetic data along with their corresponding labels. This allows us to train

a classifier on the generated data and evaluate its performance on a real test dataset by calculating the appropriate metrics, thereby assessing the effectiveness of the distributed approach.

## 3   Related Work

Several distributed learning frameworks and paradigms have been proposed to distribute GAN models across multiple devices or nodes. MD-GAN (Hardy et al., 2019) was the first distributed GAN framework, in which a single generator resides on the server to reduce the computational burden on clients, while multiple discriminators are deployed on edge devices. The generator produces batches of synthetic data and sends them to the discriminators. Each discriminator computes its own loss, as well as the generator's loss, which is then sent back to the server and averaged. Additionally, the discriminators are periodically swapped among clients to prevent overfitting. Multi-Generator MD-GAN (Jain & Wilfred Godfrey, 2022) extends MD-GAN by training label-specific generators on the server side. UA-GAN (Zhang et al., 2021) addresses data heterogeneity (non-i.i.d. data) in distributed GAN training. In this framework, a central Generator resides on the server to alleviate client-side computational demands, while multiple discriminators are deployed on the clients. The method aggregates all distributed discriminators into a simulated centralized discriminator, where the overall odds value is computed as a weighted mixture of the odds values from local discriminators. Also, CAP-GAN (Zhang et al., 2023) proposes a novel approach for federated GAN training within the Mobile Edge Computing (MEC) paradigm. However, it suffers from limitations such as reliance on a large number of edge servers and the transmission of generated data over the air. These four approaches address data heterogeneity and resource-constrained environments but doesn't consider device heterogeneity, multi-domain clients, and data sharing constraints, as they involve sending synthetic raw images from the server to clients. Although the data is synthetic, it still reflects the underlying distribution of the original datasets on which it was trained. This distribution should never be shared with any entity other than the participating clients, not even with the server.

FedGAN (Rasouli et al., 2020) applied Federated Learning (FL) to GANs, adopting the standard FL approach using the FedAVG algorithm to aggregate model updates. Similarly, GANs are used in federated settings in (Fan & Liu, 2020), where either both the generator and discriminator, or only one of them, is federated. These methods address data sharing constraints by ensuring that no data or labels are shared outside of clients and also handle data heterogeneity in terms of varying local dataset sizes. However, they overlook challenges such as differing data distributions, device heterogeneity, resource-constrained environments, and multi-domain datasets.

FeGAN (Guerraoui et al., 2020) introduced a novel aggregation strategy by assigning scores to clients based on both the size of their local datasets and the Kullback–Leibler (KL) divergence between their local label distributions and a global reference distribution, as shown in Equation 2. However, this method requires clients to share their label distribution statistics with the server, potentially raising privacy concerns. Fed-TGAN (Zhao et al., 2021) follows a similar methodology but focuses on tabular data. It uses centralized column encoders trained on shared client data statistics to initialize the model before proceeding with federated learning. FL-Enhance (Chiaro et al., 2023) and PerFed-GAN (Cao et al., 2022) utilize GANs to support or replace traditional FL frameworks. These approaches often involve sharing data—either real or synthetic—with the server, which introduces significant data privacy risks. FLIGAN (Maliakel et al., 2024) performs federated GAN training for incomplete tabular data by using federated encoding of columns, followed by GAN training with node grouping based on label distribution. Although these methods address data heterogeneity, they share the data distribution with the server and doesn't tackle device heterogeneity, resource-constrained environments, and multi-domain datasets.

$$D_{\mathrm{KL}}(P \parallel Q) = \sum_i P(i) \log\left(\frac{P(i)}{Q(i)}\right) \tag{2}$$

Federated Split GANs (Kortoçi et al., 2022) combine split learning with federated learning to address device heterogeneity. In this approach, the generator resides on the server, while multiple discriminators are distributed and split across edge devices based on their capabilities. The discriminators are federated and

aggregated every few epochs using the FedAVG algorithm. While this approach handles device heterogeneity, resource-constrained environments, and varying client dataset sizes, it overlooks challenges related to multi-domain datasets and data heterogeneity in terms of different data distributions. Additionally, it transmits synthetic data over the air to end devices.

Other approaches such as IFL-GAN (Li et al., 2022) and EFFGAN (Ekblom et al., 2022) attempt to mitigate data heterogeneity using different strategies. IFL-GAN incorporates maximum mean discrepancy (MMD) into the model averaging process to reduce distributional differences across clients. EFFGAN addresses the issue by ensembling fine-tuned federated generators to produce synthetic data. PS-FedGAN (Wijesinghe et al., 2024a) trains only the discriminator in a distributed fashion, while HFL-GAN (Petch et al., 2025) uses hierarchical federated learning to manage data heterogeneity by grouping clients based on cosine similarity and performing local and global federations. OS-GAN (Kasturi & Hota, 2023) performs one-shot distributed GAN training by sharing locally trained GANs and classifiers, which are used to infer labels for generated samples and construct a global dataset. These methods, along with FedGen (Zhao et al., 2025) and FLGAN (Ma et al., 2023), address data heterogeneity and avoids raw data sharing, but they fall short in addressing device heterogeneity, resource constraints, and multi-domain scenarios.

Further, U-FedGAN (Wijesinghe et al., 2024b) operates by training the discriminators both on the client side and the server side, while all generators are hosted on the server. To preserve privacy, only the gradients of the discriminators—trained on real client data—are shared with the server to continue the collaborative training with the generators. GANFed (Fan et al., 2024) embeds a discriminator within the Federated Learning network, where it interacts with the shallow layers of the generator to form a complete GAN model. AFL-GAN (Quan et al., 2024b) and RCFL-GAN (Quan et al., 2024a) incorporate reinforcement learning and maximum mean discrepancy (MMD) to handle data heterogeneity and enhance training efficiency in resource-constrained environments by selecting a subset of clients for each training round. Another approach, AuxFedGAN (Haghbin et al., 2025), integrates GANs into federated learning by utilizing a pre-trained Auxiliary Classifier-GAN to support a federated classifier. While these methods effectively address data heterogeneity, resource constraints, and data sharing concerns, they do not consider the challenges posed by device heterogeneity or multi-domain datasets.

PFL-GAN (Wijesinghe et al., 2023) is an innovative approach that replaces traditional FL frameworks with a GAN-based solution. It trains a conditional GAN (cGAN) locally on each client's dataset. These locally trained cGANs are then sent to the server, which generates synthetic data and constructs refined datasets for each client. The similarity between client datasets is calculated using the Kullback–Leibler Divergence (KLD) from latent representations of synthetic data, obtained via a pre-trained encoder. This method effectively handles data heterogeneity and multi-domain datasets but still does not address device heterogeneity or training on resource-constrained devices.

Three notable approaches, HSFL (Sun et al., 2025), ESFL (Zhu et al., 2024) and DFL (Samikwa et al., 2024), combine Split Learning and Federated Learning across multiple nodes, where each node holds only part of the model while the remainder resides on the server. Both methods address the challenge of selecting the optimal cut point for each client based on their computational capabilities, employing heterogeneous cuts—i.e., each client may have a different cut—to adapt accordingly. However, both approaches implement standard Split Learning, meaning that labels are still transmitted from the client to the server. Moreover, the problem they tackle is simpler than ours: they only require selecting a single cut per client. In contrast, our method involves selecting four cuts per client, as we apply U-shaped Split Learning to both the generator and the discriminator. This not only increases the complexity of the problem but also enhances the security of the system. Another approach (Wu et al., 2025) applies heterogeneous split federated learning, assuming each client has a different cut point. However, it does not specify how the optimal cut point is determined for each client—It just assigns different cut points to different clients.

As discussed above, while many existing works on decentralized GANs address the challenge of data heterogeneity, only a few consider device heterogeneity and the limitations of resource-constrained devices. Even fewer tackle the scenario where clients possess datasets from different domains. To the best of our knowledge, none of the existing approaches simultaneously address all of these challenges while adhering to strict data sharing constraints—namely, that no data is shared in its raw form, whether real or synthetic,

and labels are never shared outside the client. Only intermediate activations or gradients are permitted to be exchanged.

# 4 Methodology

## 4.1 Overview

HuSCF-GAN, as illustrated in Figure 4, operates as follows. First, Each client's model is split between the client device and the server. The server determines the optimal cut points for each client using a genetic algorithm, taking into account both computational capacity and data transmission rate, with the objective of minimizing overall training latency across all devices. Both the Generator and Discriminator are divided into three segments: Head, Server, and Tail. The client retains the Generator Head ($G_H$), Generator Tail ($G_T$), Discriminator Head ($D_H$), and Discriminator Tail ($D_T$), each of which contains at least one layer. The intermediate segments—Generator Server ($G_S$) and Discriminator Server ($D_S$)—are hosted on the server, shared among all clients, and each must also contain at least one layer, corresponding to the central layer of the Generator or Discriminator, respectively. The server maintains a mapping of clients to their associated server-side layers, which may differ across clients due to heterogeneous cut points.

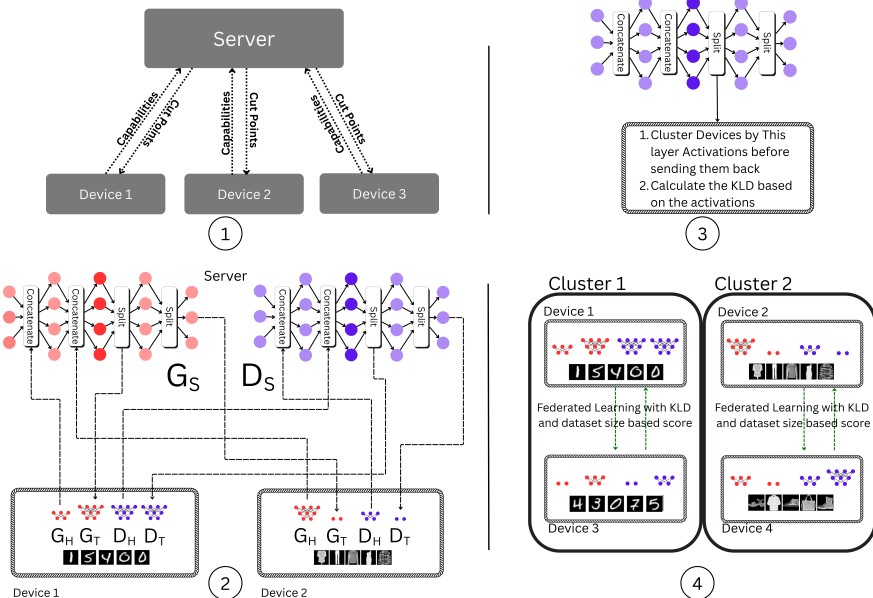

Figure 4: **HuSCF-GAN Overview:** Clients first send device capabilities to the server, which uses a Genetic Algorithm to assign optimal cut points. Clients then perform U-shaped split learning, exchanging intermediate activations/gradients with the server. Every $E$ epochs, the server clusters discriminator activations and computes intra-cluster KLD scores and perform an intra-cluster federated learning round whose aggregation weights consider both data size and KLD.

Second, Once the setup is complete, Heterogeneous U-Shaped Split Learning begins. In this stage, each client trains its "head" section and sends the resulting activations to the server. The server concatenates these activations with those from other clients participating in the same layer and continues the forward pass, combining activations across layers as needed. When a client's final server-side layer is reached, the server sends the corresponding activations back so the client can continue the forward pass through its "tail" section. This process is applied to both the generator and the discriminator during training. The backward pass follows the reverse path with the gradients being shared instead of the activations.

Third, after $E$ epochs, the server applies a clustering algorithm to the activations from the intermediate layer of the discriminator residing on the server (while processing real data). This algorithm groups clients into clusters based on activation similarity.

Fourth, within each cluster, federated learning is performed using a scoring mechanism that considers both the size of each client's local dataset and the Kullback-Leibler (KL) divergence of its activations—which is used as an alternative to sharing data labels. This helps address data heterogeneity across clients.

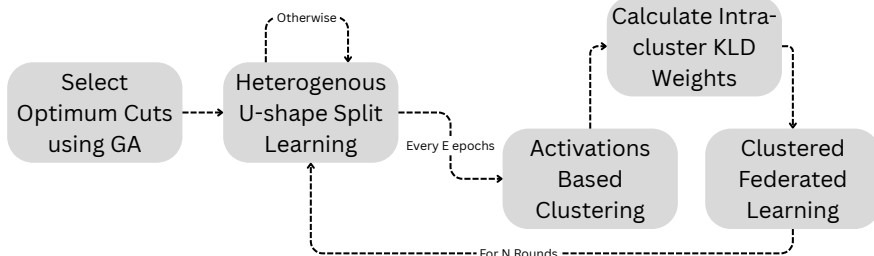

Figure 5: Overview of the proposed HuSCF-GAN procedure.

Figure 5 present the complete procedure of our proposed approach as a diagram.

Table 2: Comprehensive List of Symbols and Their Definitions Used in This Work

| Symbol | Description |
| --- | --- |
| $G$ | Generator |
| $D$ | Discriminator |
| $G_H$, $G_T$ | Generator Head and Tail |
| $D_H$, $D_T$ | Discriminator Head and Tail |
| $\theta_{G_H}$ | Parameters of $G_H$ |
| $\theta_{G_T}$ | Parameters of $G_H$ |
| $\theta_{D_H}$ | Parameters of $D_H$ |
| $\theta_{D_T}$ | Parameters of $D_T$ |
| $\theta_{G_S}$ | Parameters of The Server-side Generator Part |
| $\theta_{D_S}$ | Parameters of The Server-side Discriminator Part |
| $E$ | Number of Epochs |
| $K$ | The Clients |
| $b$ | Batch Size |
| $FLOPs$ | Floating Point Operations |
| $FLOPS$ | Floating Point Operations per second |
| $n_k$ | The local dataset size on client $k$ |

## 4.2 Latency Model

HuSCF-GANs implement Split Learning by selecting optimal cut points for each of the $K$ clients. While the number of cut points is flexible and depends on the number of devices over which the model is distributed, we choose to distribute each client's model between the client and the server only, thereby selecting four cut points per client. This results in the following model components: the client-side Generator Head and Tail, the client-side Discriminator Head and Tail, and the server-side Generator Segment and Discriminator Segment. This selection is performed using a genetic algorithm aimed at minimizing the total training iteration latency for the clients participating in the training process, based on their computational capabilities and data transmission rates. To achieve this, we first need to define the latency model for the system, similar to the model described in (Sun et al., 2025), but with some differences as our model is a GAN model with 4 cut points but their model is a traditional classifier with only one cut point.

The computational latencies for the clients depend on the total number of FLOPs required for $G_H$, $G_T$, $D_H$, and $D_T$, the batch size used during these operations, and the computational capabilities of the clients, as shown in:

$$T_{k,x}^{C,F} = \frac{b \cdot \gamma_{C,x}^{F}(l_{k,x})}{f_k \kappa_k}, \quad T_{k,x}^{C,B} = \frac{b \cdot \gamma_{C,x}^{B}(l_{k,x})}{f_k \kappa_k}, \quad \text{for } x \in \{G_H, G_T, D_H, D_T\} \tag{3}$$

Here, $T_{k,x}^{C,F}$ and $T_{k,x}^{C,B}$ denote the Client's forward and backward propagation computational latencies, respectively, for client $k$ and model segment $x \in \{G_H, G_T, D_H, D_T\}$. The term $\gamma_{C,x}^F(l_{k,x})$ represents the number of floating point operations (FLOPs) required for forward propagation—either up to the cut layer during head training or starting from the cut layer during tail training—at layer $l_{k,x}$, while $\gamma_{C,x}^B(l_{k,x})$ denotes the FLOPs required for backward propagation. The variable $b$ is the batch size used during training, $f_k$ is the CPU frequency of client $k$, and $\kappa_k$ represents the number of FLOPs executable per CPU cycle on client $k$.

While the computational latencies for the server segments per layer are described as follows:

$$T_{G,i}^{S,F} = \frac{b\gamma_{s,G,i}^F}{f_s\kappa_s}, \quad T_{G,i}^{S,B} = \frac{b\gamma_{s,G,i}^B}{f_s\kappa_s}, \quad T_{D,i}^{S,F} = \frac{b\gamma_{s,D,i}^F}{f_s\kappa_s}, \quad T_{D,i}^{S,B} = \frac{b\gamma_{s,D,i}^B}{f_s\kappa_s} \tag{4}$$

Here, $T_{G,i}^{S,F}, T_{G,i}^{S,B}, T_{D,i}^{S,F}$, and $T_{D,i}^{S,B}$, represent the computational latency for the server-side Generator, and Discriminator respectively for a given layer $i$, where $\gamma$ represent the number of FLOPs needed while training the Generator, and Discriminator during Forward and Backward Propagation for the $i$th layer, $f_s$, and $\kappa_s$ represents the CPU frequency and The FLOPs per CPU cycle respectively for the server.

For the transmission latency, we describe the latency for both clients and the server as follows:

$$T_{k,x}^{u,\phi} = \frac{b \cdot \xi_x(l_{k,x})}{R_k}, \quad \text{for } x \in \{G_H, G_T, D_H, D_T\}, \; \phi \in \{F, B\} \tag{5}$$

$$T_{k,x}^{d,F} = \frac{b \cdot \xi_x(l_{k,x_T} - 1)}{R_s}, \quad T_{k,x}^{d,B} = \frac{b \cdot \xi_x(l_{k,x_H} + 1)}{R_s}, \quad \text{for } x \in \{G, D\} \tag{6}$$

Here, $T_{k,x}^{u,\phi}$ represents the uplink transmission latency for client $k$ and model segment $x \in \{G_H, G_T, D_H, D_T\}$, during propagation direction $\phi \in \{F, B\}$. This term accounts for the time taken to send smashed data or gradients from the client to the server. Similarly, $T_{k,x}^{d,F}$ and $T_{k,x}^{d,B}$ denote the downlink transmission latencies from the server to client $k$ for the Generator or Discriminator ($x \in \{G, D\}$) during forward and backward propagation, respectively. The forward transmission ($T_{k,x}^{d,F}$) occurs just before the client-side tail cut points ($l_{k,x_T}$), while the backward transmission ($T_{k,x}^{d,B}$) begins just after the head cut points ($l_{k,x_H}$). $\xi_x(l)$ represents the size, in bytes, of the smashed data or gradients at layer $l$ for model component $x$. The variables $R_k$ and $R_s$ denote the uplink and downlink transmission rates (in bytes per second) for client $k$ and the server, respectively. The batch size is denoted by $b$.

To define the total latency of the system, we first have to introduce the following equations:

$$S_{x,i}^F = \max\left(S_{x,i-1}^F + T_{x,i}^{S,F}N_i, \max_{\substack{k \in \mathcal{K} \\ l_{k,x_H}=i}}\left(T_{k,x_H}^{C,F} + T_{k,x_H}^{u,F}\right)\right), \quad \text{for } x \in \{G, D\} \tag{7}$$

$$S_{x,i}^B = \max\left(S_{x,i+1}^B + T_{x,i}^{S,B}N_i, \max_{\substack{k \in \mathcal{K} \\ l_{k,x_T}=i}}\left(T_{k,x_T}^{C,B} + T_{k,x_T}^{u,B}\right)\right), \quad \text{for } x \in \{G, D\} \tag{8}$$

Here, $S_{x,i}^F$ and $S_{x,i}^B$ represent the maximum cumulative latency from the first layer in the head segments up to layer $i$ and from the final layer in the tail segments down to layer $i$ during forward and backward propagation, respectively, for $x \in \{G, D\}$ on the server side. The forward latency $S_{x,i}^F$ accumulates from the input (layer 0) up to layer $i$, where the initial condition is $S_{x,0}^F = 0$. Similarly, the backward latency $S_{x,i}^B$ accumulates from the output layer $n+1$ down to layer $i$, with initial condition $S_{x,n+1}^B = 0$, where $n$ is the index of the final layer of the server-side model for both Generator and Discriminator. The first term inside the outer $\max(\cdot)$

represents the sequential server-side computation at layer $i$ scaled by the number of participating clients $N_i$. The second term accounts for the maximum latency among all clients whose cut layer (i.e., the boundary between client and server) is at layer $i$. Specifically, this includes the local client computation and the uplink transmission time needed to deliver activations (in forward) or gradients (in backward) to the server. The term $N_i$ is the number of clients participating in this layer on the server side. If no clients are assigned to a given layer on the server, $N_i = 0$, and that layer incurs no server-side latency.

Now, the total latency is defined by the following equations:

$$L_x^F = \max_{k \in \mathcal{K}} \left( S_{x,l_{k,x_T}-1}^F + T_{k,x}^{d,F} + T_{k,x_T}^{C,F} \right), \quad L_x^B = \max_{k \in \mathcal{K}} \left( S_{x,l_{k,x_H}+1}^B + T_{k,x}^{d,B} + T_{k,x_H}^{C,B} \right), \quad \text{for } x \in \{G, D\} \quad (9)$$

$$L_T = L_G^F + L_G^B + 3 \left( L_D^F + L_D^B \right) \quad (10)$$

Here, $L_x^F$ and $L_x^B$ represent the total forward and backward propagation latency, respectively, for model $x \in \{G, D\}$ (Generator or Discriminator). The total system latency $L_T$ reflects the complete duration of one training iteration. It includes the Generator's forward and backward passes and three instances of the Discriminator's forward and backward passes. The factor of three accounts for the Discriminator being trained on a batch of real data, a batch of fake data, and once more when computing gradients used to update the Generator.

### 4.3 Selecting The Optimum Cuts

For our optimization problem, we employ a genetic algorithm to minimize the total latency of the system by optimizing the variable, which represents the optimum four cut points for each client $k$

$$\mathbf{l} \in \mathcal{L} = \left\{ (l_1, l_2, \dots, l_K) \,|\, l_k = (l_{k,G_H}, l_{k,G_T}, l_{k,D_H}, l_{k,D_T}) \right\}.$$

A major challenge lies in its high complexity: since we aim to determine four cut points for each client, the search space grows exponentially with the number of clients. This makes convergence of the genetic algorithm increasingly difficult as the client population increases. To address this issue and reduce the search space, we employ an reduction strategy where we reduce the total number of clients to the number of device profiles (Devices having the same computation capabilities). This approach maintains representativeness while significantly shrinking the optimization domain. , as illustrated in Figure 6.

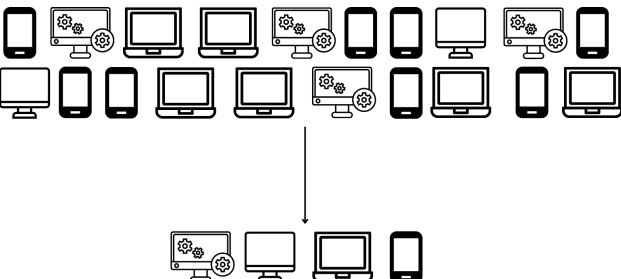

Figure 6: Downsampling the number of clients while representing all different profiles.

This approach is shown to be able to reach the same sub-optimal value as a full genetic algorithm in appendix D. To ensure the accuracy of latency evaluation, the fitness of each individual is always measured after upsampling the solution back to the original number of clients.

The procedure begins with the initialization of a population of 1000 individuals. Each individual represents a possible set of cut points for the clients and is typically encoded as an array of client-specific cut points initialized randomly.

A fitness function is used to evaluate how well each individual (i.e., each potential solution) performs. In our case, the fitness function is defined as:

$$\text{Fit}(\mathbf{l}) = -L_T(\mathbf{l}), \tag{11}$$

where $L_T(\mathbf{l})$ denotes the total latency corresponding to the solution $\mathbf{l}$. Since the genetic algorithm is designed to maximize the fitness function, and our objective is to minimize latency, we introduce a negative sign to reverse the optimization direction. Alternative formulations, such as using the reciprocal of latency, can also be employed and are expected to yield equivalent results under our strategy.

After initializing the population of candidate cut-point configurations, the genetic algorithm runs for a predefined number of generations. In each generation, we evolve the population by applying the following steps, gradually searching for a configuration that minimizes total system latency:

1. **Selection:** We begin by selecting parents based on their fitness, which in our case is the negative of the total latency—where parents are the individuals with the highest fitness (lowest latency). To do this, we use tournament selection: five individuals are randomly chosen from the current population, and the one with the highest fitness (i.e., the lowest latency) is selected as a parent. This process is repeated to select a second parent.

2. **Crossover:** Next, the two selected parents are combined to create offspring according to a selected crossover rate. Crossover happens to explore new combinations of cut points across clients. We alternate with equal probability between:

   - **Uniform crossover:** Each parent shares half of its clients' cut points randomly.
   - **Two-point crossover:** Each parent is split into three segments (Of clients' cut points) at two points, and segments are interchanged to produce diversity in the offspring.

3. **Mutation:** After crossover, we introduce small random changes by moving the cut points for some clients to other layers according to a selected mutation rate. This adds variation to the population and helps prevent the algorithm from getting stuck in a local minimum.

4. **Elitism:** To make sure we don't lose our best solutions, we carry over the top two individuals (combination of clients' cut points) from the current generation directly into the next one without any changes. This guarantees that the best configurations discovered so far are always preserved.

The algorithm repeats this process for several generations. Over time, the population improves, and the best solution found—denoted as $\mathbf{l}^*$—represents the set of cut points that achieves the lowest overall system latency.

### 4.4 Heterogeneous U-Shaped Split Learning

The split learning process proceeds as follows. After the server determines the optimal cut points for each client—based on their computational capabilities—it records which clients participate in which server-side layers. Due to heterogeneous cuts, this assignment varies across clients; however, the middle layer of both the generator and discriminator must reside in the server, thus shared by all clients.

For each training batch of size $b$ as shown in Figure 7, the forward pass begins with each client executing its local Generator Head ($G_H$) and transmitting its activation outputs to the server. At the first server-side layer, the server concatenates the activations received from all clients participating in that layer. For subsequent layers, the server concatenates its previous layer's output with the activations from the $G_H$ modules of clients whose participation starts at that specific layer. This continues until the middle layer is reached, which is shared by all clients. Beyond the middle layer, the server sends activation outputs back to the clients whose participation ends at each layer, so they can process their corresponding Generator Tail ($G_T$). The same procedure applies to the Discriminator.

During the backward pass, the same communication pattern is followed in reverse. Gradients are propagated from the tail to the head, and the exchanged data are gradients rather than activations.

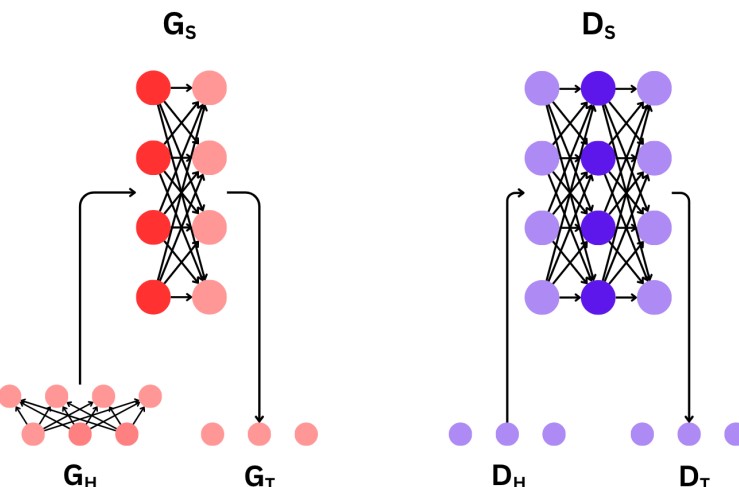

Figure 7: **Heterogeneous U-Shaped Split Learning:** The model is divided between the client and the server—hence the term *split learning*. It is described as *U-shaped* because the initial and final segments of the model reside on the client side, while the intermediate portion is hosted on the server, forming a U-like structure as indicated by the arrows. The term *heterogeneous* indicates that different clients may use different cut points for splitting the model.

## 4.5 Clustered Federated Learning

Every $E$ epochs, a round of federation is performed. During the first two rounds, no clustering or Kullback-Leibler Divergence (KLD) weighting is applied, FedAVG(McMahan et al., 2017) is applied in its vanilla form allowing the local models to mature and produce meaningful outputs. After these initial rounds, and before each subsequent round of federation, we apply a clustering algorithm to the activations of the middle layer of the discriminator, which is shared across all clients as described in Subsection 4.4.

This clustering is performed during the Discriminator's training on real data, as real data typically yields more informative feature representations. For each client $k$, let $\alpha_{k,D}^{(n_D/2)}$ denote the average activation vector from the middle layer of the Discriminator, where $n_D$ is the total number of Discriminator layers. The middle layer, $n_D/2$, is shared across all clients, and clustering is performed on its activation vectors.

We collect the activation vectors from all clients and input them into a KMeans clustering algorithm:

$$\text{labels} = \text{KMeans}\left(\left\{\alpha_{k,D}^{(n_D/2)}\right\}_{k \in \mathcal{K}}\right) \tag{12}$$

where each cluster label $\text{label}_k \in \text{labels}$ corresponds to client $k$. Each cluster represents clients whose datasets originate from the same or similar domains. After clustering, we calculate how each client's distribution diverges from the other clients' distributions within the same cluster by using the Kullback-Leibler Divergence (KLD) score.

To compute this, we follow these steps:

First, we take the averaged activations of all clients and apply the softmax function to each:

$$P_k = \text{Softmax}\left(\alpha_{k,D}^{(n_D/2)}\right), \quad \forall k \in \mathcal{K} \tag{13}$$

Next, for each client $k$, we compute the average distribution of the other clients in its cluster $C_k$ as

$$P_j = \frac{\displaystyle\sum_{\substack{x \in C_k \\ x \neq k}} P_x}{|C_k| - 1} \tag{14}$$

Then, we calculate the KLD between $P_k$ and $P_j$ using (2), and denote it by $\text{KLD}_k$.

Finally, we use the KLD score along with the dataset size to define the weighting score for intra-cluster federated learning as follows:

$$s_k = \frac{n_k \; e^{-\beta \; \text{KLD}_k}}{\displaystyle\sum_{j \in C_k} n_j \; e^{-\beta \; \text{KLD}_j}} \tag{15}$$

where $\beta$ is a scaling parameter. The equation is written such that clients with larger dataset sizes should have greater weight, and clients with higher divergence from the group should have lower weight. The choice of exponential decay as an inverse function is due to its smoothness and numerical stability, ensuring the avoidance of exploding weights when the divergence approaches zero, unlike the use of $\frac{1}{\text{KLD}}$.

Then, Federated Learning is applied to update the parameters of the client-side model components for all clients within the same cluster as follows:

$$\boldsymbol{\theta}^{t+1} = \sum_{k \in C_i} s_k \, \boldsymbol{\theta}_k^{t+1}, \quad \forall C_i \in \mathcal{C}, \quad \boldsymbol{\theta} \in \{\boldsymbol{\theta}_{G_H}, \boldsymbol{\theta}_{G_T}, \boldsymbol{\theta}_{D_H}, \boldsymbol{\theta}_{D_T}\} \tag{16}$$

For the server-side model components, the parameters are updated using all clients collectively. In this case, the scoring mechanism described in (14) and (15) is applied globally across all clients, rather than on a per-cluster basis, since the server-side model is shared among all clients. Then the server-side model components are updated similar to that in (16) but with the global scores.

## 5   Experimental Setup

We set up an experimental environment using a **conditional GAN (cGAN)** implemented in PyTorch (Paszke, 2019), following the architecture described in **Table 3**, which comprises **3M parameters**. This architecture is adopted as a proof of concept to demonstrate that we can effectively reduce total latency while still achieving strong performance. While real-world applications may require more complex architectures, the same methodology remains applicable. To simulate a heterogeneous setting, we consider 100 clients whose profiles are randomly sampled from the device configurations listed in **Table 4** of which many are resource constrained IoT devices. The reason for selecting this setup is that it includes profiles representing clients with varying computational capabilities—ranging from weak to medium devices—thereby simulating a real-world scenario. Moreover, the setup mimics configurations similar to those of actual devices, with realistic variations in computational power, rather than randomly assigning CPU frequency, FLOPs per cycle, or transmission rates. Setups from other papers were also experimented on, and yielded similar consistent results.

We evaluate the proposed system using the following benchmark datasets: **MNIST** (LeCun et al., 1998), **Fashion-MNIST (FMNIST)** (Xiao et al., 2017), **Kuzushiji-MNIST (KMNIST)** (Clanuwat et al., 2018), and **NotMNIST** (Bulatov, 2011). We also evaluate our framework using **CIFAR10** (Krizhevsky et al., 2009) and **SVHN** (Netzer et al., 2011) datasets to test the adaptability of our approach to higher resolution datasets. We also test our approach on medical imaging datasets (Yang et al., 2021): BloodMNIST and DermaMNIST which are Blood Cell Microscope and Dermatoscope imaging to evaluate how our framework would perform on a real world use case. Finally, to examine the cross-modal adaptability of our framework, we conduct experiments on the AudioMNIST Becker et al. (2024) dataset, which represents the audio modality.

We compare **HuSCF-GAN** with $E = 5$, hyperparameter $\beta = 150$, population size $PS = 1000$, crossover rate $CR = 0.7$, and mutation rate $MR = 0.01$ against several benchmarks **MD-GAN** (Hardy et al., 2019),

**FedGAN** (Rasouli et al., 2020), **Federated Split GANs** (Kortoçi et al., 2022), **HFL-GAN** (Petch et al., 2025), and **PFL-GAN** (Wijesinghe et al., 2023), all with the same cGAN architecture explained earlier to ensure fairness. The system's performance is evaluated and compared against baseline methods using three metrics:

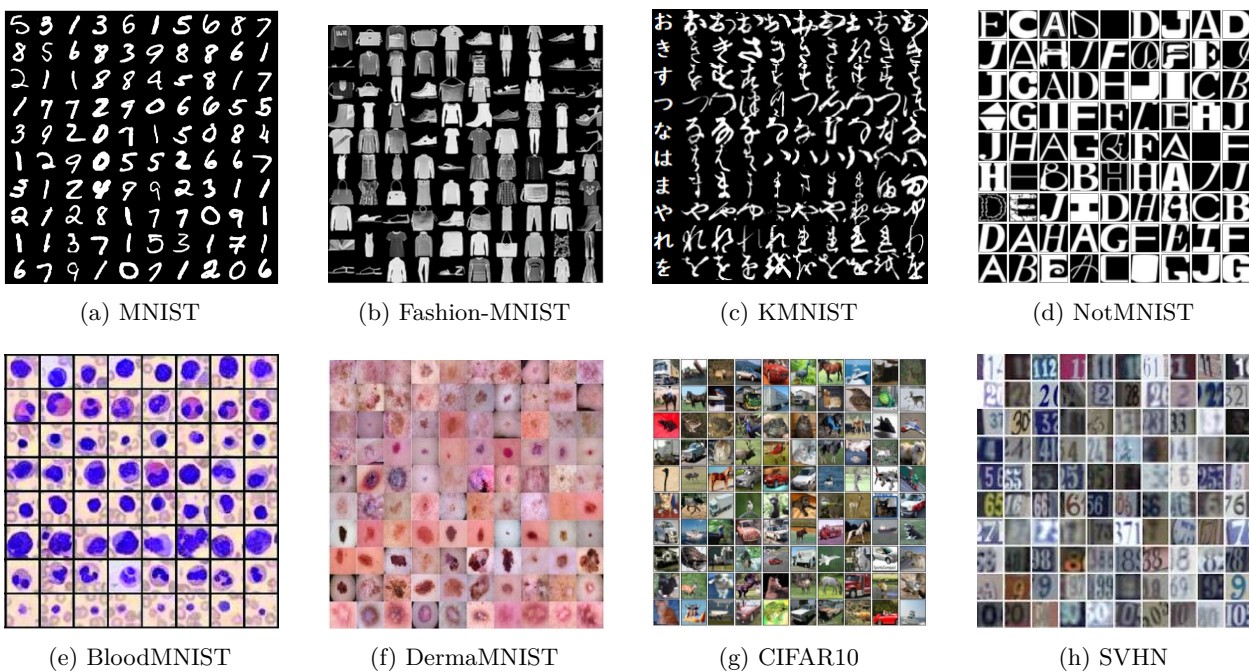

(a) MNIST  (b) Fashion-MNIST  (c) KMNIST  (d) NotMNIST

(e) BloodMNIST  (f) DermaMNIST  (g) CIFAR10  (h) SVHN

Figure 8: Sample images from the eight datasets used in our experiments: MNIST, Fashion-MNIST, KMNIST, NotMNIST, BloodMNIST, DermaMNIST, CIFAR10, and SVHN.

1. **Classification Metrics:** A CNN model is trained exclusively on 30,000 generated samples (i.e., without using any real data), with labels uniformly distributed across classes. The model is then evaluated on a real test set, which was not seen during training, to compute metrics such as Accuracy, Precision, Recall, F1 Score, and False Positive Rate (One-vs-All). This evaluation serves to assess the generative model by measuring how closely the distribution of generated data matches that of the real data. If the distributions are similar, the classifier learns meaningful features and achieves better scores. Conversely, if the distributions differ significantly, the classifier performs poorly.

2. **Image Generation Quality:** To evaluate the quality of generated images:

   - For the MNIST family datasets, we adopt the metric introduced in (Hardy et al., 2019), inspired by the Inception Score (Salimans et al., 2016). This metric is computed using a pre-trained, dataset-specific classifier in place of the Inception V3 model. The evaluation is performed separately for each dataset within the MNIST family.
   - For higher-resolution datasets such as CIFAR-10 and SVHN, we employ the Fréchet Inception Distance (FID) score (Heusel et al., 2017), as it provides meaningful assessments for these domains. In contrast, the FID score is less appropriate for MNIST-family datasets due to their structural simplicity relative to the Inception V3 model's expectations. (Lower FID scores indicate better image quality.)

3. **Average Training Latency per Iteration:** This measures the average computational time required for a single training iteration, providing insight into the training efficiency of the model.

Eight distinct scenarios are considered to evaluate the proposed approach across various test cases. The first scenario involves single-domain IID data, followed by single-domain non-IID data. Next, two-domain

Table 3: Generator and Discriminator Architectures

| Generator | Discriminator |
|---|---|
| Input: $z \in \mathbb{R}^{100}$, label $\in \mathbb{R}^{10}$ | Input: image 28×28, label $\in \mathbb{R}^{10}$ |
| Label embedding and concatenation | Label embedding reshaped and concatenated with image |
| FC $\to$ 256×7×7 + BatchNorm + ReLU | Conv 2→64, kernel 4×4, stride 2 + BatchNorm + L.ReLU |
| ConvT 256→128, kernel 4×4, stride 2 + BatchNorm + ReLU | Conv 64→128, kernel 4×4, stride 2 + BatchNorm + L.ReLU |
| ConvT 128→128, kernel 3×3, stride 1 + BatchNorm + ReLU | Conv 128→128, kernel 3×3, stride 1 + BatchNorm + L.ReLU |
| ConvT 128→64, kernel 4×4, stride 2 + BatchNorm + ReLU | Conv 128→256, kernel 4×4, stride 2 + BatchNorm + L.ReLU |
| ConvT 64→1, kernel 3×3, stride 1 + Tanh | Flatten $\to$ FC $\to$ Sigmoid |
| Output: generated image 28×28 | Output: probability (real/fake) |

Table 4: Computation and Communication Capabilities of Devices

| Device | Frequency (MHz) | FLOPs/cycle | Transm. Rate (Bytes/s) |
|---|---|---|---|
| Device 1 | 480 | 1 | $50 \times 10^6$ |
| Device 2 | 6000 | 8 | $150 \times 10^6$ |
| Device 3 | 15600 | 8 | $1000 \times 10^6$ |
| Device 4 | 5720 | 8 | $300 \times 10^6$ |
| Device 5 | 4000 | 4 | $50 \times 10^6$ |
| Device 6 | 9000 | 4 | $100 \times 10^6$ |
| Device 7 | 12000 | 10 | $800 \times 10^6$ |
| Server | 42000 | 16 | $1000 \times 10^6$ |

IID and two-domain non-IID scenarios are examined, including a more challenging two-domain non-IID setting. To assess scalability, a four-domain IID scenario is also included. Finally, two specialized scenarios are considered: a two-domain non-IID setting for medical imaging applications, and a two-domain highly non-IID scenario for higher-resolution images.

# 6 Evaluation & Results

The experimental results are evaluated using three key metrics: dataset-specific image generation scores, classifier performance with 95% Wald confidence intervals, and training latency. These metrics are assessed across both single-domain and multi-domain settings, under IID and Non-IID data distributions. Table 5 summarizes the different test scenarios upon which the evaluation is assessed. It is important to note that an ablation study conducted to evaluate different components of the approach is shown in appendix A, Further results and comparisons are also demonstrated in appendix C.

Table 5: Summary of Test Scenarios

| # Domains | Data Distribution | Datasets Used |
|---|---|---|
| 1 | IID | MNIST |
| 1 | Non-IID | MNIST |
| 2 | IID | MNIST + FMNIST |
| 2 | Non-IID | MNIST + FMNIST |
| 2 | Highly Non-IID | MNIST + FMNIST |
| 4 | IID | MNIST + FMNIST + KMNIST + NotMNIST |
| 2 | Non-IID | BloodMNIST + DermaMNIST |
| 2 | Highly Non-IID | CIFAR10 + SVHN |
| 1 | Non-IID | AudioMNIST |

## 6.1 Image Generation Scores & Classifier Performance Comparison

We evaluate the performance of our algorithms against several baselines across six different scenarios, each representing a distinct data distribution.

### 6.1.1 Single-Domain IID Data

In the first scenario, all clients possess local datasets drawn from the same domain, with data being independent and identically distributed (IID). Specifically, we use the MNIST dataset, where each client holds 600 images.

In this scenario, all algorithms demonstrate similar performance in terms of the MNIST score, as illustrated in Figure 9, and classification metrics, as shown in Table 6. However, our algorithm achieves slightly higher classification metrics compared to the others. This can be attributed to the simplicity of the scenario—since it involves only a single domain with IID data distribution, all algorithms perform well under these favorable conditions.

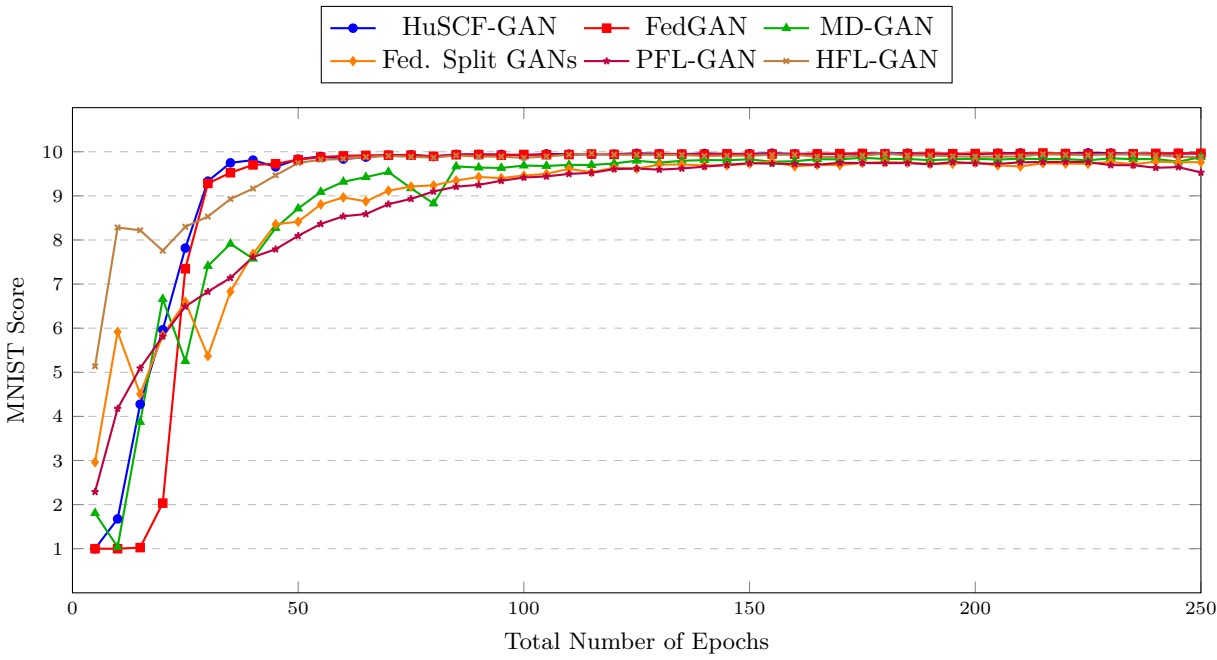

Figure 9: **MNIST score vs. Training Epochs - Single-Domain IID Data:** This Plot shows similar performance across all approaches

Table 6: Classifier Performance - Single-Domain IID Data

| | MNIST Dataset | | | | |
| --- | --- | --- | --- | --- | --- |
| | Accuracy ↑ | Precision ↑ | Recall ↑ | F1 Score ↑ | FPR ↓ |
| FedGAN (Rasouli et al., 2020) | 97.3%±0.32 | 97.3%±0.32 | 97.31%±0.32 | 97.29%±0.32 | 0.3%±0.11 |
| MD-GAN (Hardy et al., 2019) | 96.53%±0.36 | 96.58%±0.36 | 96.51%±0.36 | 96.5%±0.36 | 0.34%±0.12 |
| Fed. Split GANs (Kortoçi et al., 2022) | 94.72%±0.44 | 94.89%±0.43 | 94.69%±0.44 | 94.71%±0.43 | 0.59%±0.15 |
| PFL-GAN (Wijesinghe et al., 2023) | 97.11%±0.33 | 97.12%±0.33 | 97.11%±0.33 | 97.18%±0.32 | 0.32%±0.11 |
| HFL-GAN (Petch et al., 2025) | 93.84%±0.47 | 93.92%±0.33 | 93.82%±0.33 | 93.8%±0.33 | 0.68%±0.16 |
| HuSCF-GAN | **97.71%±0.29** | **97.73%±0.29** | **97.7%±0.29** | **97.69%±0.29** | **0.29%±0.1** |

### 6.1.2 Single-Domain Non-IID Data

To increase the complexity of the previous test case, we introduce non-IID characteristics to the system. In this scenario all clients possess local datasets originating from the same domain (MNIST), but the data

distribution is non-independent and non-identically distributed (non-IID). The heterogeneity is simulated as follows: while some clients have access to the full set of labels, 40 clients have 2 labels excluded, 10 clients have 3 labels excluded, and another 10 clients have 4 labels excluded. Additionally, the quantity of data varies across clients, with some holding 600 images and others only 400.

In this test case, performance differences between algorithms begin to emerge due to the added complexity introduced by the non-IID data distribution, despite the data still being from a single domain. As shown in Figure 10, HuSCF-GAN, FedGAN, and HFL-GAN achieve the highest MNIST scores, with HuSCF-GAN demonstrating a faster convergence rate. Furthermore, Table 7 shows that HuSCF-GAN achieves the highest classification metrics among all evaluated algorithms, with up to a 4% increase in accuracy compared to the others.

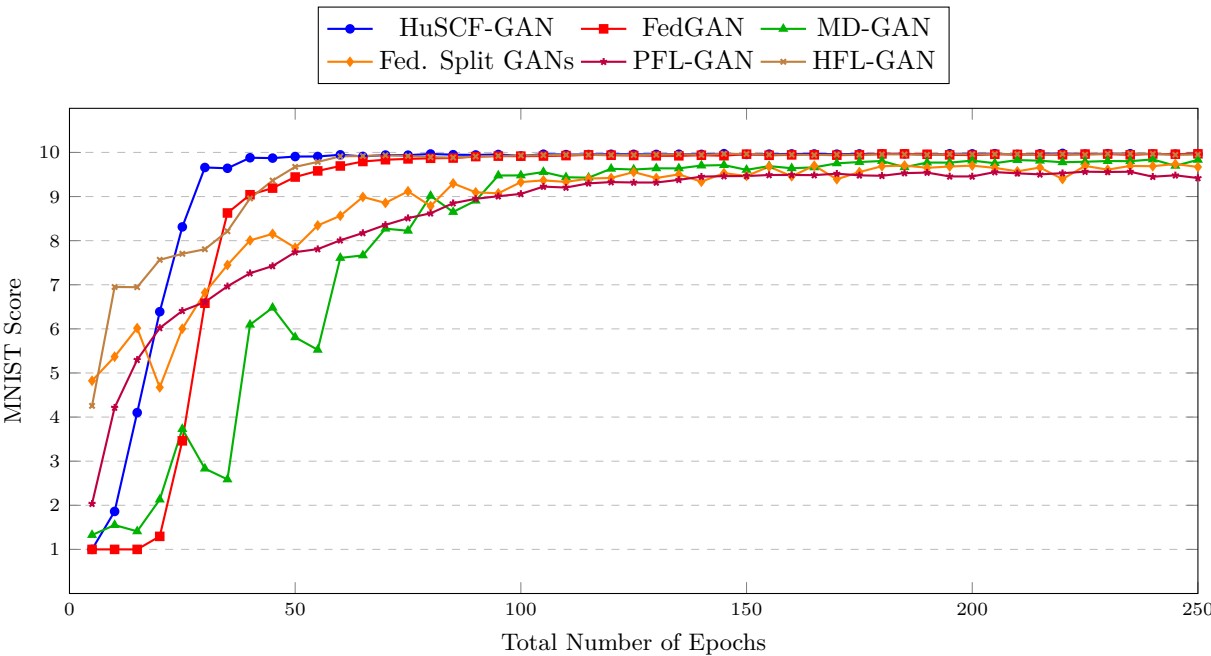

Figure 10: **MNIST Score vs. Training Epochs – Single-Domain Non-IID Data:** This plot shows similar performance across all algorithms, with HuSCF-GAN, FedGAN, and HFL-GAN achieving slightly higher scores than the others. Among them, HuSCF-GAN also demonstrates a slightly faster convergence.

Table 7: Classifier Performance - Single-Domain Non-IID Data

| | MNIST Dataset | | | | |
| --- | --- | --- | --- | --- | --- |
| | Accuracy ↑ | Precision ↑ | Recall ↑ | F1 Score ↑ | FPR ↓ |
| FedGAN (Rasouli et al., 2020) | 97.03%±0.33 | 97.04%±0.33 | 97.04%±0.33 | 97.02%±0.33 | 0.33%±0.11 |
| MD-GAN (Hardy et al., 2019) | 96.01%±0.38 | 96.14%±0.38 | 95.97%±0.38 | 96.99%±0.33 | 0.44%±0.13 |
| Fed. Split GANs (Kortoçi et al., 2022) | 93.86%±0.47 | 93.96%±0.47 | 93.82%±0.47 | 93.84%±0.47 | 0.68%±0.16 |
| PFL-GAN (Wijesinghe et al., 2023) | 92.41%±0.52 | 92.41%±0.52 | 92.41%±0.52 | 92.39%±0.52 | 0.84%±0.12 |
| HFL-GAN (Petch et al., 2025) | 94.46%±0.45 | 94.46%±0.45 | 94.44%±0.45 | 94.41%±0.45 | 0.61%±0.15 |
| HuSCF-GAN | **97.17%±0.32** | **97.21%±0.32** | **97.18%±0.32** | **97.15%±0.32** | **0.31%±0.11** |

### 6.1.3 Two-Domains IID Data

We then begin to introduce a multi-domain environment by allowing clients to hold data from two different distributions. In this scenario, clients draw data from two distinct domains: 50 clients possess IID data sampled from the MNIST dataset, while the other 50 clients possess IID data from the FMNIST dataset. All clients have an equal dataset size of 600 images.

This scenario marks the beginning of the multi-domain experiments with the inclusion of a second domain, FMNIST. Among all evaluated approaches, only our method and PFL-GAN demonstrate strong performance across both the MNIST and FMNIST datasets, as illustrated in Figures 11a and 11b. This is due to the fact that only these two approaches are capable of effectively adapting to multi-domain environments, with HuSCF-GAN converging faster than PFL-GAN. In terms of evaluation metrics, as shown in Table 8, our method achieves slightly better results than PFL-GAN, with equal or slightly higher false positive rates (FPR), and significantly outperforms all other approaches with a 20% to 80% increase in evaluation metrics such as accuracy.

It is evident from Figures 11a and 11b that the remaining methods struggle to maintain consistent performance across the two datasets, often exhibiting fluctuations and instability. While Federated Split GANs demonstrate strong latency performance, It remains incapable of handling multi-domain environments. In contrast, our approach maintains robust and stable results across both domains, achieving 1.3× to 2× higher MNIST and FMNIST scores.

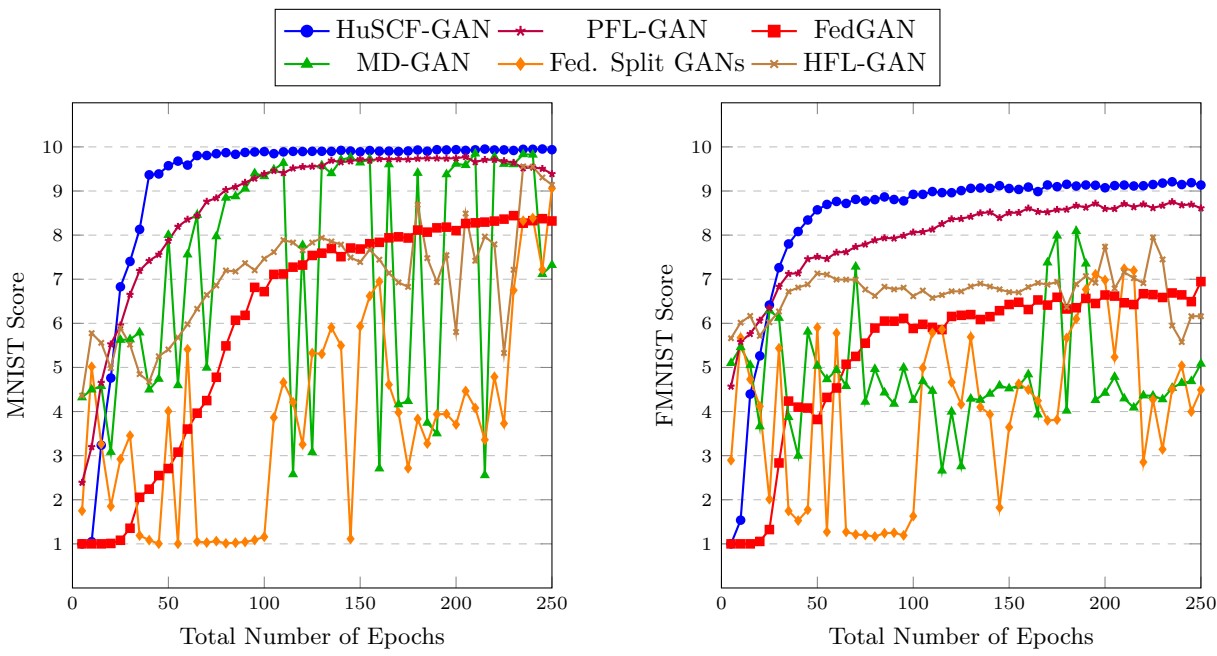

(a) **MNIST Score vs. Training Epochs :** This plot shows the superior performance of HuSCF-GAN and PFL-GAN compared to other algorithms, with HuSCF-GAN converging faster and achieving a slightly higher score than PFL-GAN. In contrast, the other approaches exhibit noticeable fluctuations and instability.

(b) **FMNIST score vs. Training Epochs:** This plot shows the superior performance of HuSCF-GAN and PFL-GAN compared to other algorithms, with HuSCF-GAN converging faster and achieving a slightly higher score than PFL-GAN. In contrast, the other approaches exhibit noticeable fluctuations and instability.

Figure 11: Image Generation Scores — Two-Domains IID Data

Table 8: Classifier Performance - Two-Domains IID Data

| | MNIST Dataset | | | | | FMNIST Dataset | | | | |
|---|---|---|---|---|---|---|---|---|---|---|
| | Accuracy ↑ | Precision ↑ | Recall ↑ | F1 Score ↑ | FPR ↓ | Accuracy ↑ | Precision ↑ | Recall ↑ | F1 Score ↑ | FPR ↓ |
| FedGAN (Rasouli et al., 2020) | 87.55%±0.65 | 87.91%±0.64 | 87.58%±0.65 | 87.16%±0.66 | 1.38%±0.23 | 61.30%±0.95 | 68.77%±0.91 | 61.30%±0.95 | 59.86%±0.96 | 4.30%±0.4 |
| MD-GAN (Hardy et al., 2019) | 61.75%±0.95 | 66.60%±0.93 | 61.95%±0.95 | 60.74%±0.95 | 4.25%±0.4 | 32.32%±0.91 | 37.77%±0.95 | 32.32%±0.91 | 30.42%±0.90 | 7.52%±0.52 |
| Fed. Split GANs (Kortoçi et al., 2022) | 64.63%±0.94 | 67.82%±0.92 | 64.68%±0.94 | 65.25%±0.94 | 3.92%±0.38 | 11.92%±0.63 | 12.47%±0.65 | 11.92%±0.63 | 9.67%±0.59 | 9.79%±0.58 |
| PFL-GAN (Wijesinghe et al., 2023) | 96.80%±0.34 | 96.81%±0.34 | 96.69%±0.34 | 96.80%±0.34 | **0.35%±0.12** | 81.34%±0.77 | 81.17%±0.77 | 81.34%±0.77 | 81.14%±0.77 | **1.85%±0.26** |
| HFL-GAN (Petch et al., 2025) | 93.33%±0.49 | 93.43%±0.49 | 93.29%±0.49 | 93.27%±0.49 | 0.74%±0.17 | 44.37%±0.98 | 55.13%±0.99 | 44.37%±0.98 | 42.39%±0.98 | 6.18%±0.47 |
| HuSCF-GAN | **97.23%±0.33** | **96.96%±0.34** | **97.07%±0.34** | **97.21%±0.33** | 0.35%±0.12 | **83.93%±0.72** | **83.77%±0.72** | **83.91%±0.72** | **83.54%±0.73** | 1.75%±0.26 |

### 6.1.4 Two-Domains Non-IID Data

Building on the multi-domain setup, we introduce non-IID characteristics to further assess how well our algorithm performs compared to others. In this scenario, clients draw data from two distinct domains: 50 clients possess non-IID data sampled from the MNIST dataset, and the other 50 clients possess non-IID data from the FMNIST dataset. Within each domain, some clients have access to the full set of labels, while 20 clients have two labels excluded, 5 clients have three labels excluded, and another 5 clients have four labels excluded. Additionally, data quantity varies, with some clients holding 600 images and others 400.

In this scenario, data heterogeneity is introduced through the non-IID distribution, increasing the overall complexity of the system. It is evident that our approach outperforms all other methods in terms of MNIST and FMNIST scores, as well as classification metrics—including PFL-GAN. While PFL-GAN delivers competitive results, it exhibits some difficulty in handling non-IID data, as shown in Figures 12a and 12b, and in terms of classification accuracy in Table 9. Our method achieves $1.1\times$ to $1.125\times$ better MNIST and FMNIST scores than PFL-GAN, and up to $2\times$ better scores compared to other algorithms. HuSCF-GAN achieves up to 5% higher evaluation metrics than PFL-GAN and a 10% to 80% improvement over the remaining methods.

Our approach consistently maintains stable and high performance across both datasets, in contrast to the performance degradation observed in PFL-GAN and the highly unstable behavior of the other algorithms.

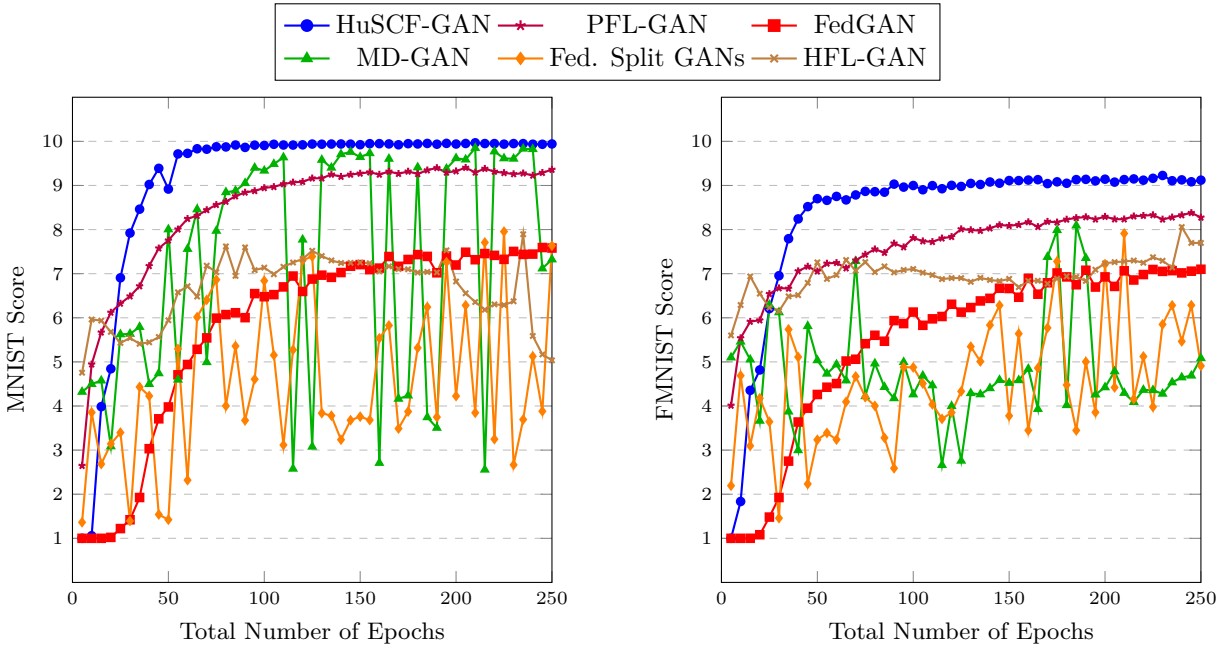

(a) **MNIST Score vs. Training Epochs:** This plot shows that HuSCF-GAN achieves the highest score, with performance approximately $1.1\times$ higher than PFL-GAN and up to $1.6\times$ higher than other algorithms.

(b) **FMNIST score vs. Training Epochs:** This plot shows that HuSCF-GAN achieves the highest score, with performance approximately $1.125\times$ higher than PFL-GAN and up to $2\times$ higher than other algorithms.

Figure 12: Image Generation Scores — Two-Domains Non-IID Data

Table 9: Classifier Performance - Two-Domains Non IID Data

| | MNIST Dataset | | | | | FMNIST Dataset | | | | |
|---|---|---|---|---|---|---|---|---|---|---|
| | Accuracy ↑ | Precision ↑ | Recall ↑ | F1 Score ↑ | FPR ↓ | Accuracy ↑ | Precision ↑ | Recall ↑ | F1 Score ↑ | FPR ↓ |
| FedGAN (Rasouli et al., 2020) | 84.02%±0.72 | 86.73%±0.66 | 84.01%±0.72 | 83.02%±0.73 | 1.77%±0.26 | 61.08%±0.96 | 67.28%±0.91 | 61.08%±0.96 | 60.76%±0.95 | 4.32%±0.4 |
| MD-GAN (Hardy et al., 2019) | 14.36%±0.69 | 17.13%±0.75 | 15.21%±0.71 | 8.83%±0.55 | 9.47%±0.57 | 72.84%±0.88 | 74.78%±0.86 | 72.84%±0.88 | 71.82%±0.89 | 3.02%±0.34 |
| Fed. Split GANs (Kortoçi et al., 2022) | 56.20%±0.98 | 63.49%±0.95 | 55.83%±0.99 | 54.92%±0.99 | 4.86%±0.42 | 20.92%±0.80 | 21.81%±0.82 | 20.92%±0.80 | 16.58%±0.74 | 8.79%±0.55 |
| PFL-GAN (Wijesinghe et al., 2023) | 91.15%±0.56 | 91.35%±0.55 | 91.15%±0.56 | 91.14%±0.56 | 0.98%±0.19 | 79.37%±0.80 | 79.84%±0.79 | 79.37%±0.80 | 79.41%±0.80 | 2.29%±0.29 |
| HFL-GAN (Petch et al., 2025) | 33.16%±0.93 | 69.61%±0.90 | 33.92%±0.94 | 29.17%±0.89 | 7.37%±0.51 | 69.30%±0.90 | 72.35%±0.88 | 69.30%±0.90 | 67.99%±0.90 | 3.41%±0.36 |
| HuSCF-GAN | **96.21%±0.37** | **96.28%±0.37** | **96.16%±0.38** | **96.19%±0.38** | **0.42%±0.13** | **81.90%±0.75** | **82.60%±0.74** | **81.90%±0.75** | **81.75%±0.75** | **2.01%±0.27** |

### 6.1.5 Two-Domains Highly Non-IID Data

Another highly non-IID scenario is conducted in which clients draw data from two distinct domains: 50 clients possess non-IID data sampled from the MNIST dataset, and the other 50 clients possess non-IID data from the FMNIST dataset. Within each domain, 20 clients have two labels excluded, and another 30 clients have three labels excluded. Furthermore, dataset sizes vary across clients—some have 600 entries, others have 200, and a few have only 100. This scenario introduces a more intense level of data heterogeneity compared to the previous one, further increasing the challenge for all algorithms.

Despite the added complexity, HuSCF-GAN continues to demonstrate stable and superior performance. As illustrated in Figures 13a and 13b, our method consistently achieves high MNIST and FMNIST scores, with a significant performance margin over all other algorithms—achieving $1.2\times$ to $2.1\times$ higher scores. Additionally, our approach attains the highest classification accuracy, as shown in Table 10, with improvements ranging from 10% to 40% on MNIST and from 10% to 80% on FMNIST compared to all other algorithms. These results highlight the robustness and adaptability of HuSCF-GAN in highly heterogeneous settings.

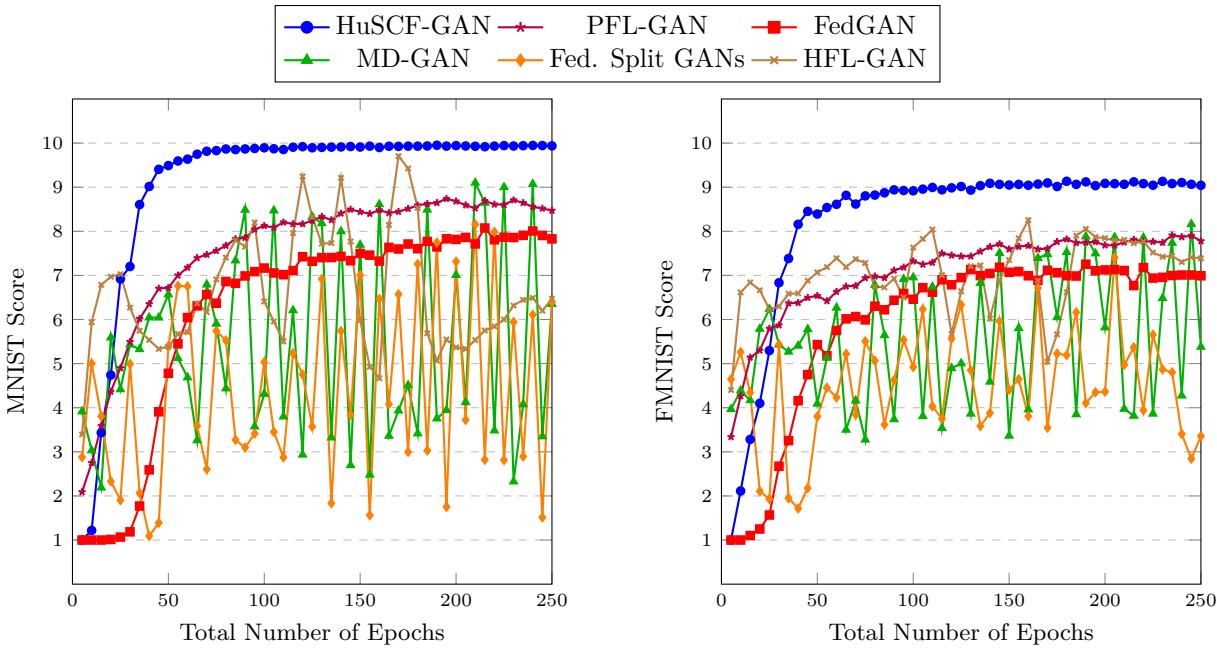

(a) **MNIST Score vs. Training Epochs:** HuSCF-GAN achieves scores that are $1.2\times$ to $2\times$ higher and significantly more stable compared to other approaches.

(b) **FMNIST score vs. Training Epochs:** HuSCF-GAN achieves scores that are $1.15\times$ to $2\times$ higher and significantly more stable compared to other approaches.

Figure 13: Image Generation Scores — Two-Domains Highly Non-IID Data

Table 10: Classifier Performance - Two-Domains Highly Non IID Data

| | MNIST Dataset | | | | | FMNIST Dataset | | | | |
|---|---|---|---|---|---|---|---|---|---|---|
| | Accuracy ↑ | Precision ↑ | Recall ↑ | F1 Score ↑ | FPR ↓ | Accuracy ↑ | Precision ↑ | Recall ↑ | F1 Score ↑ | FPR ↓ |
| FedGAN (Rasouli et al., 2020) | 86.63%±0.67 | 87.37%±0.65 | 86.57%±0.67 | 86.14%±0.68 | 1.48%±0.24 | 62.28%±0.95 | 70.26%±0.90 | 62.28%±0.95 | 62.41%±0.95 | 4.19%±0.39 |
| MD-GAN (Hardy et al., 2019) | 69.71%±0.91 | 72.08%±0.89 | 70.11%±0.91 | 69.19%±0.92 | 3.36%±0.35 | 28.10%±0.88 | 35.02%±0.94 | 28.10%±0.88 | 23.26%±0.83 | 7.99%±0.53 |
| Fed. Split GANs (Kortoçi et al., 2022) | 56.50%±0.99 | 67.99%±0.92 | 56.90%±0.99 | 56.60%±0.99 | 4.82%±0.42 | 9.06%±0.57 | 8.46%±0.56 | 9.06%±0.57 | 6.88%±0.51 | 10.10%±0.59 |
| PFL-GAN (Wijesinghe et al., 2023) | 85.86%±0.68 | 86.04%±0.68 | 85.86%±0.68 | 85.83%±0.68 | 1.57%±0.24 | 77.75%±0.83 | 78.31%±0.83 | 77.75%±0.83 | 77.39%±0.84 | 2.47%±0.3 |
| HFL-GAN (Petch et al., 2025) | 77.48%±0.83 | 83.51%±0.72 | 77.48%±0.83 | 77.05%±0.83 | 2.50%±0.3 | 68.87%±0.91 | 71.81%±0.89 | 68.87%±0.91 | 67.31%±0.92 | 3.46%±0.36 |
| HuSCF-GAN | **96.15%±0.38** | **96.11%±0.38** | **96.10%±0.38** | **96.10%±0.38** | **0.45%±0.13** | **81.46%±0.75** | **81.32%±0.75** | **81.46%±0.75** | **80.61%±0.77** | **1.95%±0.27** |

### 6.1.6 Four-Domains IID Data

In this scenario, clients draw data from four distinct domains: 25 clients possess IID data sampled from the MNIST dataset, 25 from the FMNIST dataset, 25 from the KMNIST dataset, and the remaining 25 from the NotMNIST dataset. All clients have an equal dataset size of 600 images.

In the final scenario, the number of domains is increased from two to four by introducing the KMNIST and NotMNIST datasets. This setup is designed to evaluate the scalability of the algorithms in multi-domain environments. Among all methods, only our approach and PFL-GAN are able to adapt effectively to this increased complexity. However, our method significantly outperforms PFL-GAN, achieving $1.2\times$ to $1.58\times$ higher image generation scores, and outperforms all other approaches by up to $2.5\times$.

Table 11 presents the classification metrics across the different domains and algorithms. HuSCF-GAN achieves 1% to 5% higher metrics than PFL-GAN in most domains, with the exception of the FMNIST and NotMNIST domains, where the results are comparable. Nonetheless, HuSCF-GAN achieves up to 50% higher classification metrics than all other algorithms. These results demonstrate both the scalability and consistent effectiveness of our approach in increasingly complex multi-domain settings.

Table 11: Classifier Performance - Four-Domains IID Data

| | MNIST Dataset | | | | | FMNIST Dataset | | | | |
| | Accuracy ↑ | Precision ↑ | Recall ↑ | F1 Score ↑ | FPR ↓ | Accuracy ↑ | Precision ↑ | Recall ↑ | F1 Score ↑ | FPR ↓ |
| --- | --- | --- | --- | --- | --- | --- | --- | --- | --- | --- |
| FedGAN (Rasouli et al., 2020) | 41.25%±0.96 | 57.90%±0.97 | 42.00%±0.96 | 41.17%±0.96 | 6.53%±0.48 | 50.58%±0.99 | 49.47%±0.99 | 50.58%±0.99 | 47.81%±0.98 | 5.49%±0.52 |
| MD-GAN (Hardy et al., 2019) | 26.07%±0.88 | 29.67%±0.91 | 26.44%±0.88 | 23.95%±0.86 | 8.22%±0.54 | 38.49%±0.97 | 41.42%±0.98 | 38.49%±0.97 | 36.41%±0.96 | 6.83%±0.49 |
| Fed. Split GANs (Kortoçi et al., 2022) | 16.05%±0.72 | 17.90%±0.76 | 16.35%±0.73 | 13.61%±0.69 | 9.33%±0.57 | 22.41%±0.66 | 19.73%±0.63 | 22.41%±0.66 | 19.36%±0.63 | 8.62%±0.55 |
| PFL-GAN (Wijesinghe et al., 2023) | 94.45%±0.46 | 94.52%±0.46 | 94.45%±0.46 | 94.44%±0.46 | 0.62%±0.15 | **82.01%±0.54** | 82.09%±0.54 | **82.09%±0.54** | 82.11%±0.54 | 1.96%±0.27 |
| HFL-GAN (Petch et al., 2025) | 45.49%±0.98 | 62.12%±0.97 | 46.18%±0.98 | 46.98%±0.98 | 6.04%±0.47 | 50.86%±0.99 | 56.01%±0.99 | 50.86%±0.99 | 45.56%±0.98 | 5.46%±0.45 |
| HuSCF-GAN | **95.94%±0.38** | **95.97%±0.38** | **95.96%±0.38** | **95.93%±0.38** | **0.45%±0.13** | 81.94%±0.55 | **82.26%±0.55** | 81.94%±0.55 | 81.98%±0.55 | 2.01%±0.28 |
| | KMNIST Dataset | | | | | NotMNIST Dataset | | | | |
| | Accuracy ↑ | Precision ↑ | Recall ↑ | F1 Score ↑ | FPR ↓ | Accuracy ↑ | Precision ↑ | Recall ↑ | F1 Score ↑ | FPR ↓ |
| FedGAN (Rasouli et al., 2020) | 27.66%±0.88 | 30.42%±0.91 | 27.66%±0.88 | 27.81%±0.88 | 8.04%±0.53 | 38.53%±0.96 | 38.93%±0.96 | 38.31%±0.96 | 32.32%±0.91 | 6.84%±0.49 |
| MD-GAN (Hardy et al., 2019) | 33.36%±0.91 | 43.15%±0.98 | 33.36%±0.91 | 32.46%±0.90 | 7.40%±0.51 | 37.09%±0.95 | 40.55%±0.98 | 33.01%±0.91 | 6.98%±0.5 |
| Fed. Split GANs (Kortoçi et al., 2022) | 22.04%±0.82 | 24.50%±0.86 | 22.04%±0.82 | 20.40%±0.79 | 8.66%±0.55 | 16.37%±0.73 | 18.85%±0.78 | 16.24%±0.73 | 16.74%±0.74 | 9.28%±0.57 |
| PFL-GAN (Wijesinghe et al., 2023) | 67.72%±0.92 | 71.50%±0.90 | 67.72%±0.92 | 68.15%±0.92 | 3.59%±0.36 | 88.11%±0.65 | 88.23%±0.65 | 88.11%±0.65 | 88.14%±0.65 | **1.17%±0.21** |
| HFL-GAN (Petch et al., 2025) | 33.07%±0.91 | 38.59%±0.97 | 33.07%±0.91 | 33.13%±0.91 | 7.44%±0.51 | 36.10%±0.94 | 42.31%±0.98 | 35.80%±0.94 | 32.79%±0.91 | 7.12%±0.5 |
| HuSCF-GAN | **72.91%±0.85** | **74.31%±0.84** | **72.91%±0.85** | **73.09%±0.85** | **3.01%±0.33** | **88.30%±0.65** | **88.54%±0.65** | **88.28%±0.65** | **88.32%±0.65** | 1.30%±0.22 |

### 6.1.7 Medical Imaging Datasets

To further evaluate the robustness of our algorithm, we introduce a **non-IID multi-domain scenario** designed to reflect a realistic **medical imaging use case**. In this setting, 100 clients are divided across two distinct domains: **50 clients** are assigned data sampled from the BloodMNIST dataset, and the remaining **50 clients** from the DermaMNIST dataset. Within each domain, data heterogeneity is introduced through non-IID sampling — some clients have access to the full set of labels, while **20 clients** are missing two labels, **5 clients** are missing three labels, and another **5 clients** are missing four labels.

This scenario is specifically designed to showcase the applicability of our approach to **medical imaging**, where **privacy preservation** is crucial — no patient images or labels can be shared — and **domain separation** (e.g., between different imaging types) plays a vital role.

It is evident that our proposed approach consistently outperforms all competing methods on both BloodMNIST and DermaMNIST datasets, as well as across various classification metrics. Although PFL-GAN demonstrates competitive performance, it shows limitations when dealing with non-IID data distributions, as illustrated in Figures 15a and 15b, and reflected in the classification accuracies reported in Table 12. In contrast, our method achieves between $1.2\times$ and $3\times$ higher BloodMNIST and DermaMNIST scores compared to the other algorithms. HuSCF-GAN delivers an improvement in classifier metrics ranging from 9% to 70% over the remaining approaches.

Table 12: Classifier Performance - Medical Imaging Datasets

| | BloodMNIST Dataset | | | | | DermaMNIST Dataset | | | | |
| | Accuracy ↑ | Precision ↑ | Recall ↑ | F1 Score ↑ | FPR ↓ | Accuracy ↑ | Precision ↑ | Recall ↑ | F1 Score ↑ | FPR ↓ |
| --- | --- | --- | --- | --- | --- | --- | --- | --- | --- | --- |
| FedGAN (Rasouli et al., 2020) | 37.91%±1.61 | 60.72%±1.63 | 41.04%±1.68 | 33.07%±1.60 | 8.51%±0.97 | 62.74%±2.13 | 22.18%±1.84 | 20.08%±1.76 | 20.30%±1.77 | 12.12%±1.49 |
| MD-GAN (Hardy et al., 2019) | 7.40%±0.89 | 13.16%±1.14 | 12.38%±1.09 | 2.54%±0.53 | 12.48%±1.13 | 29.33%±2.00 | 9.56%±1.01 | 12.56%±1.18 | 8.87%±0.94 | 16.70%±1.66 |
| Fed. Split GANs (Kortoçi et al., 2022) | 10.06%±1.00 | 13.12%±1.15 | 14.43%±1.21 | 4.33%±0.69 | 12.43%±1.13 | 40.10%±2.18 | 14.40%±1.39 | 16.41%±1.47 | 13.17%±1.33 | 11.72%±1.17 |
| PFL-GAN (Wijesinghe et al., 2023) | 68.99%±1.56 | 73.12%±1.47 | 68.99%±1.56 | 70.22%±1.53 | 4.38%±0.64 | 26.08%±1.97 | 69.42%±2.01 | 26.08%±1.97 | 33.46%±2.10 | 10.04%±1.36 |
| HFL-GAN (Petch et al., 2025) | 32.83%±1.52 | 43.40%±1.67 | 30.72%±1.46 | 28.02%±1.40 | 9.93%±1.00 | 20.10%±1.76 | 15.08%±1.42 | 13.37%±1.33 | 11.20%±1.21 | 16.21%±1.62 |
| HuSCF-GAN | **77.55%±1.37** | **78.44%±1.34** | **76.10%±1.40** | **75.52%±1.41** | **3.25%±0.60** | **63.59%±2.10** | **34.74%±2.08** | **31.95%±2.04** | **28.05%±1.99** | **8.59%±1.26** |

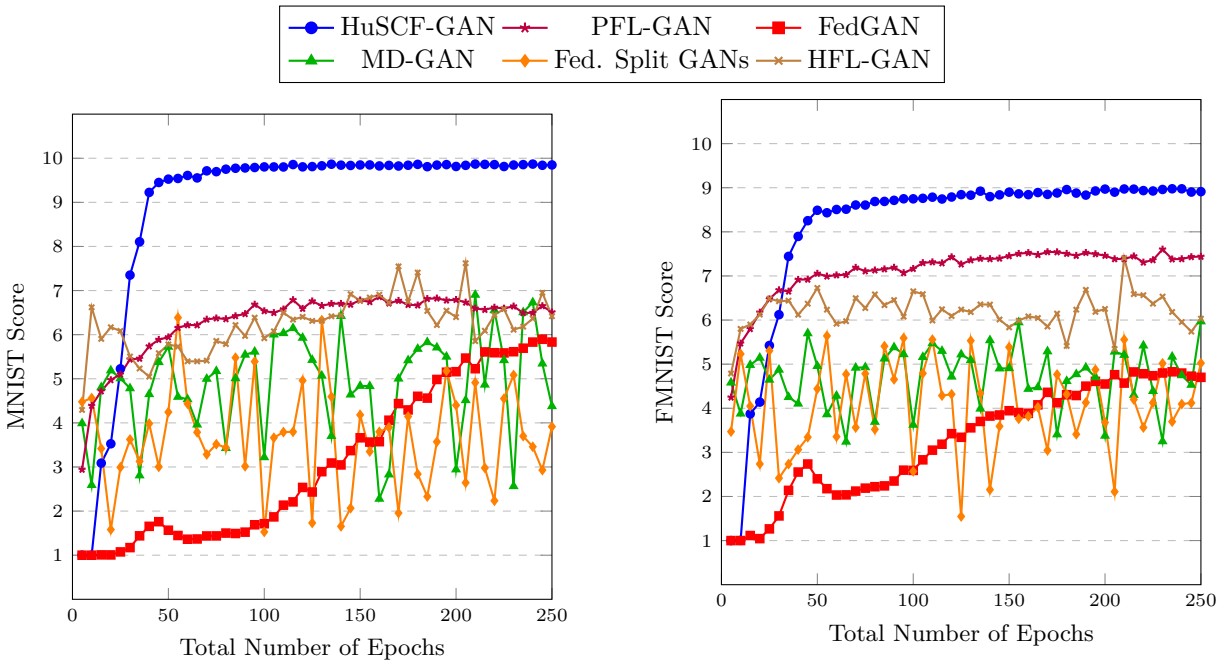

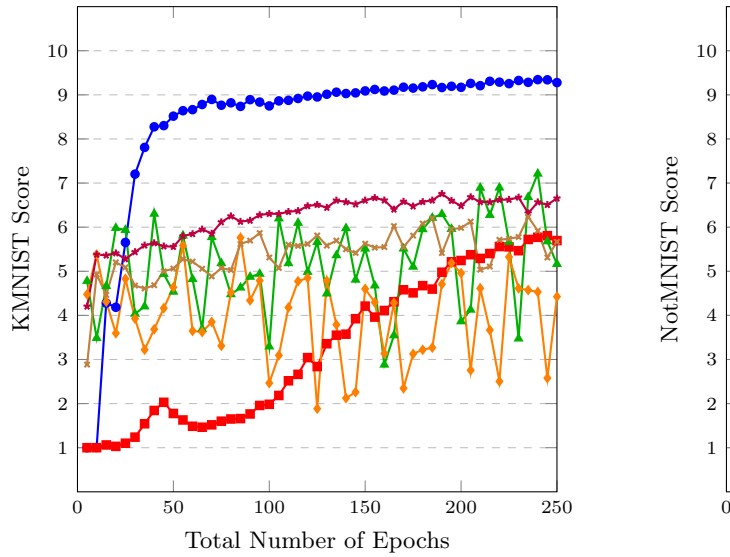

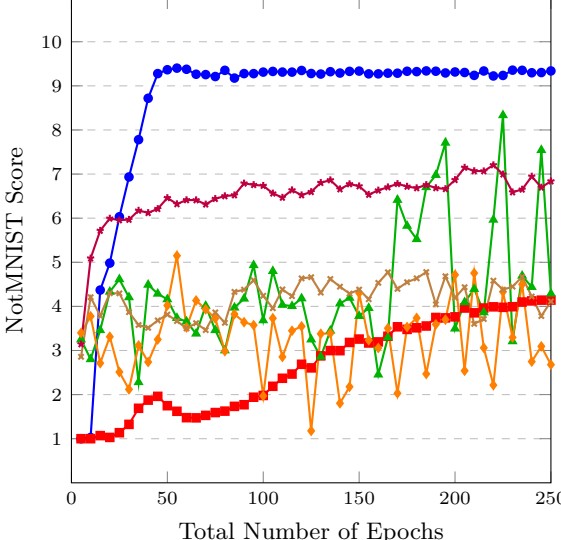

(a) **MNIST score vs. Training Epochs:** HuSCF-GAN achieves up to 2× higher scores than other approaches.

(b) **FMNIST score vs. Training Epochs:** HuSCF-GAN achieves up to 2× higher scores than other approaches.

(c) **KMNIST score vs. Training Epochs:** HuSCF-GAN achieves up to 2× higher scores than other approaches.

(d) **NotMNIST score vs. Training Epoch:** HuSCF-GAN achieves up to 2.1× higher scores than other approaches.

Figure 14: Image Generation Scores — Four-Domains IID Data

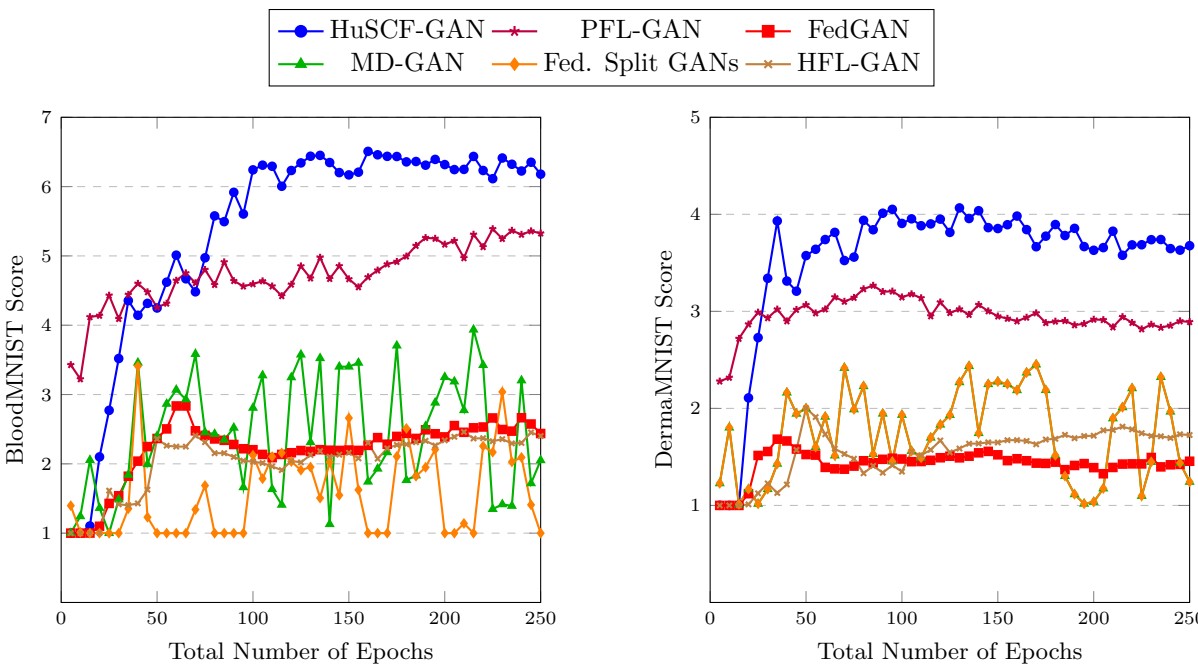

(a) **BloodMNIST Score vs. Training Epochs:** HuSCF-GAN achieves scores that are 1.2× to 3× higher compared to other approaches.

(b) **DermaMNIST Score vs. Training Epochs:** HuSCF-GAN achieves scores that are 1.33× to 2.2× higher compared to other approaches.

Figure 15: Image Generation Scores — Medical Imaging

### 6.1.8 Higher Resolution Datasets

Another experimental scenario is designed to evaluate the adaptability of our framework on higher-resolution datasets, namely CIFAR10 and SVHN. This setup follows a structure similar to that described in Subsection 6.1.5, where clients draw data from two distinct domains: 50 clients hold non-IID data sampled from the CIFAR10 dataset, while the remaining 50 clients hold non-IID data from the SVHN dataset. Within each domain, 20 clients have two labels excluded, and another 30 clients have three labels excluded. Furthermore, the dataset sizes vary among clients—some possess 600 samples, others 200, and a few as few as 100 samples.

For this scenario, we assess image generation quality using the Fréchet Inception Distance (FID) score, as these higher-resolution RGB datasets are more suitable for FID-based evaluation compared to the other datasets used in this study.

HuSCF-GAN continues to exhibit stable and superior performance. As illustrated in Figures 16a and 16b, our method consistently achieves substantially lower FID scores, outperforming all other algorithms with margins ranging from 2× to 70× lower values. Moreover, our approach attains the highest classification performance, as reported in Table 13, with improvements ranging from 20% to 45% on CIFAR10 and from 10% to 60% on SVHN compared to all competing methods. These findings highlight the robustness and adaptability of HuSCF-GAN when applied to highly heterogeneous, high-resolution datasets.

Table 13: Classifier Performance - Higher Resolution Images

| | CIFAR10 Dataset | | | | | SVHN Dataset | | | | |
|---|---|---|---|---|---|---|---|---|---|---|
| | Accuracy ↑ | Precision ↑ | Recall ↑ | F1 Score ↑ | FPR ↓ | Accuracy ↑ | Precision ↑ | Recall ↑ | F1 Score ↑ | FPR ↓ |
| FedGAN (Rasouli et al., 2020) | 26.26%±0.87 | 27.66%±0.88 | 26.26%±0.87 | 24.24%±0.85 | 8.19%±0.53 | 52.98%±0.97 | 50.84%±0.98 | 48.71%±0.98 | 48.02%±0.97 | 5.2%±0.44 |
| MD-GAN (Hardy et al., 2019) | 20.12%±0.79 | 20.44%±0.8 | 20.11%±0.79 | 19.41%±0.74 | 8.88%±0.55 | 10.83%±0.39 | 10.42%±0.38 | 10.40%±0.28 | 7.47%±0.32 | 9.98%±0.5 |
| Fed. Split GANs (Kortoçi et al., 2022) | 14.61%±0.68 | 13.87%±0.66 | 14.61%±0.68 | 11.31%±0.61 | 9.49%±0.57 | 11.24%±0.4 | 12.56%±0.42 | 10.8%±0.39 | 8.67%±0.35 | 9.88%±0.49 |
| PFL-GAN (Wijesinghe et al., 2023) | 39.66%±0.96 | 41.11%±0.97 | 39.66%±0.96 | 39.89%±0.96 | 6.7%±0.49 | 37.81%±0.93 | 38.4%±0.93 | 37.81%±0.93 | 35.01%±0.91 | 6.96%±0.49 |
| HFL-GAN (Petch et al., 2025) | 23.83%±0.84 | 24%±0.85 | 23.83%±0.84 | 23.42%±0.84 | 8.46%±0.54 | 37.7%±0.93 | 37.67%±0.93 | 34.38%±0.91 | 34.17%±0.91 | 6.96%±0.49 |
| HuSCF-GAN | **63.67%±0.94** | **67.84%±0.91** | **61.67%±0.95** | **61.87%±0.95** | **4.48%±0.41** | **73.35%±0.88** | **70.51%±0.9** | **71.92%±0.89** | **70.61%±0.9** | **2.96%±0.33** |

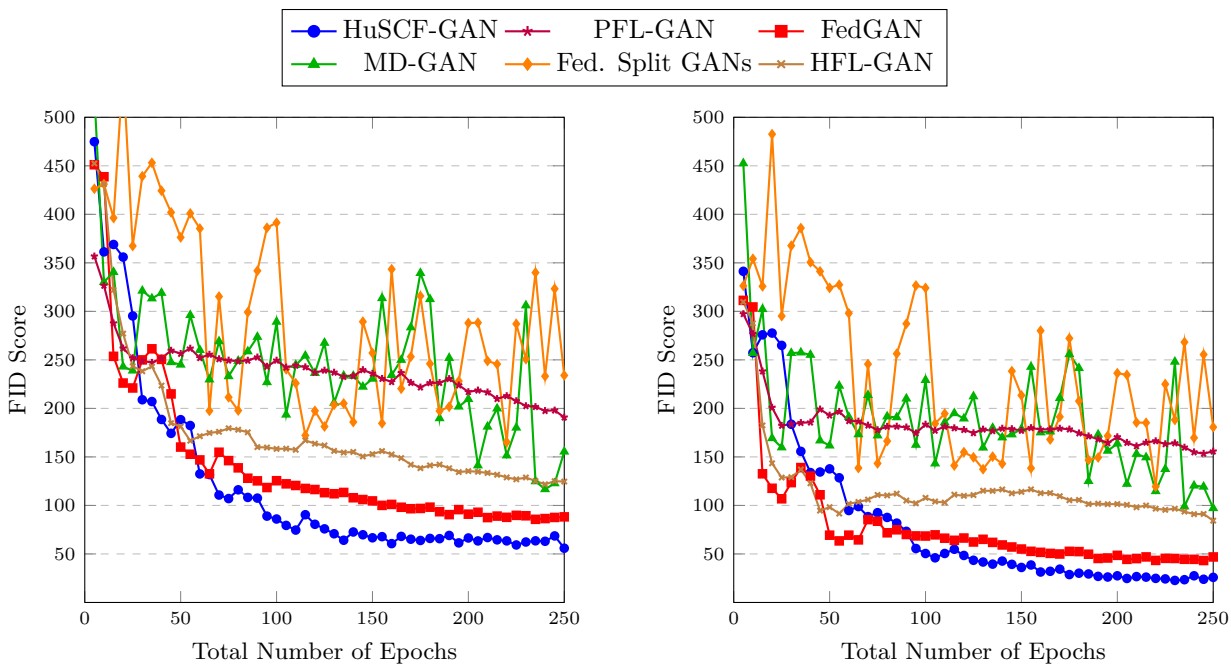

(a) **CIFAR10 FID Score vs. Training Epochs:** HuSCF-GAN achieves FID scores that are 1.8× to 60× lower compared to other approaches.

(b) **SVHN FID Score vs. Training Epochs:** HuSCF-GAN achieves scores that are 2× to 70× higher and significantly more stable compared to other approaches.

Figure 16: FID Scores — Higher Resolution Datasets

### 6.1.9   Audio Dataset

The final experimental scenario aims to evaluate the adaptability of our framework across different data modalities, specifically audio. To this end, we employ the AudioMNIST dataset Becker et al. (2024), which consists of spoken digits from zero to nine. This setup follows a similar configuration to that described in Subsubsection 6.1.2, where clients draw data from a single domain. A total of 100 clients participate in this experiment, holding non-IID partitions of the AudioMNIST dataset: 40 clients have 2 labels excluded, 10 clients have 3 labels excluded, and another 10 clients have 4 labels excluded.

Since the base model architecture relies on convolutional and deconvolutional layers, using raw audio signals directly would be incompatible. Therefore, each audio file was processed using the Librosa McFee et al. (2015) library to compute a 128-bin mel-spectrogram, which was then converted to a log-scaled power representation and resized to $28 \times 28$ pixels. The resulting spectrograms were converted to grayscale format for input to our model architecture.

While the base model is capable of generating audio samples, the output quality remains limited due to the low input resolution ($28 \times 28$), which inevitably results in the loss of fine-grained spectral information. Nonetheless, this setup effectively demonstrates the proof of concept for extending our framework to the audio domain.

HuSCF-GAN continues to exhibit stable and superior performance. As reported in Table 14, it consistently outperforms competing approaches with improvements ranging from 1.5% to 13% across all evaluation metrics. These results highlight the robustness and cross-modal adaptability of HuSCF-GAN compared to existing federated generative learning algorithms.

It is important to note that, due to the different data modality, traditional image-based evaluation metrics such as Inception Score and FID are not applicable. Instead, audio-specific generation metrics should be considered for a more accurate assessment as a direction for future work.

Table 14: Classifier Performance - Audio Experiment

| | AudioMNIST Dataset | | | | |
|---|---|---|---|---|---|
| | Accuracy ↑ | Precision ↑ | Recall ↑ | F1 Score ↑ | FPR ↓ |
| FedGAN (Rasouli et al., 2020) | 96.67%±0.45 | 96.71%±0.45 | 96.67%±0.45 | 96.67%±0.45 | 0.37%±0.15 |
| MD-GAN (Hardy et al., 2019) | 90.29%±0.75 | 91.13%±0.72 | 90.29%±0.75 | 90.19%±0.75 | 1.08%±0.26 |
| Fed. Split GANs (Kortoçi et al., 2022) | 89.34%±0.78 | 90.01%±0.76 | 89.90%±0.76 | 88.93%±0.79 | 1.06%±0.26 |
| HFL-GAN (Petch et al., 2025) | 93.62%±0.62 | 93.66%±0.62 | 93.62%±0.62 | 93.62%±0.62 | 0.71%±0.21 |
| PFL-GAN (Wijesinghe et al., 2023) | 85.00%±0.90 | 71.26%±1.15 | 85.30%±0.90 | 77.07%±1.06 | 3.78%±0.48 |
| HuSCF-GAN | **98.05%±0.35** | **98.06%±0.35** | **98.04%±0.35** | **98.03%±0.35** | **0.22%±0.12** |

## 6.2 Latency Comparison

Training latency is a critical factor when deploying generative algorithms on underutilized or resource-constrained devices. In this comparison, we evaluate the latency per training iteration—specifically, the time required to train a single batch per client.

As shown in Table 15, our approach achieves lower training latency compared to all other benchmark algorithms, with the exception of Federated Split GANs (Kortoçi et al., 2022), which demonstrates comparable performance. This substantial performance difference arises because both PFL-GAN (Wijesinghe et al., 2023) and FedGAN (Rasouli et al., 2020) train the entire GAN model on each client device. Meanwhile, HFL-GAN (Petch et al., 2025) exhibits the highest latency due to its dual-generator structure, where two generators are trained per client—effectively doubling the latency compared to FedGAN. This significantly amplifies training time, particularly on resource-constrained devices.

In contrast, MD-GAN (Hardy et al., 2019) trains only the discriminator on the client side, resulting in improved latency relative to PFL-GAN and FedGAN. Notably, both our method and Federated Split GANs dynamically adapt to the computational capabilities of individual clients, achieving optimal latency even in the presence of weaker devices.

While Federated Split GANs achieves comparable latency, it struggles considerably with non-IID data and multi-domain settings, where our method (HuSCF-GAN) demonstrates superior performance. Additionally, although PFL-GAN attains good scores and classification metrics, its latency remains significantly higher compared to HuSCF-GAN, which achieves both high performance and low latency.

These results highlight the effectiveness of our method in utilizing underpowered devices, enabling faster training while preserving strong generative and classification performance.

Table 15: **Latency Comparison Across Approaches:** HuSCF-GAN achieves the lowest latency, offering up to 58× reduction compared to other methods.

| Approach | HuSCF-GAN | PFL-GAN | FedGAN | HFL-GAN | MD-GAN | Fed. Split GANs |
|---|---|---|---|---|---|---|
| **Latency (s)** | **7.8** | 251.37 | 234.6 | 454.22 | 47.73 | **8.68** |

Table 16 presents the generator and discriminator head and tail layers corresponding to the various device profiles detailed in Table 4. It is important to note that auxiliary layers such as Batch Normalization, activation functions (e.g., ReLU), and label embeddings contribute negligibly to the overall computational cost (in terms of FLOPs) when compared to the primary layers, namely fully connected layers, convolutional layers, and transposed convolutional layers. Therefore, for clarity and relevance, only these major layers are included in the table. As shown, devices with weaker capabilities—such as devices 1 & 5—are assigned fewer layers, while more capable devices—such as device 7—incorporate additional layers within both the generator and discriminator components. A thorough analysis of the computational complexity of HuSCF-GAN is demonstrated in appendix 6.4.1

Table 16: Client-side layers per device

| Device | Generator Head | Generator Tail | Discriminator Head | Discriminator Tail |
|---|---|---|---|---|
| Device 1 | FC 256×7×7 | ConvT, 3x3, s1 | Conv, 4x4, s2 | FC 1 (Sigmoid) |
| Device 2 | FC 256×7×7 
 ConvT, 4x4, s2 | ConvT, 4x4, s2 
 ConvT, 3x3, s1 | Conv, 4x4, s2 | Conv, 4x4, s2 
 FC 1 (Sigmoid) |
| Device 3 | FC 256×7×7 
 ConvT, 4x4, s2 | ConvT, 4x4, s2 
 ConvT, 3x3, s1 | Conv, 4x4, s2 | Conv, 4x4, s2 
 FC 1 (Sigmoid) |
| Device 4 | FC 256×7×7 
 ConvT, 4x4, s2 | ConvT, 4x4, s2 
 ConvT, 3x3, s1 | Conv, 4x4, s2 | Conv, 4x4, s2 
 FC 1 (Sigmoid) |
| Device 5 | FC 256×7×7 | ConvT, 3x3, s1 | Conv, 4x4, s2 
 Conv, 4x4, s2 | FC 1 (Sigmoid) |
| Device 6 | FC 256×7×7 
 ConvT, 4x4, s2 | ConvT, 4x4, s2 
 ConvT, 3x3, s1 | Conv, 4x4, s2 
 Conv, 4x4, s2 | FC 1 (Sigmoid) |
| Device 7 | FC 256×7×7 
 ConvT, 4x4, s2 | ConvT, 4x4, s2 
 ConvT, 3x3, s1 | Conv, 4x4, s2 
 Conv, 4x4, s2 | Conv, 4x4, s2 
 FC 1 (Sigmoid) |

### 6.3 Comparison Between Label Distribution-Based and Activation-Based KLD for FL Weights Calculation

This section provides a detailed analysis of the KLD (Kullback–Leibler Divergence) computation by comparing two approaches: the activation-based KLD introduced in this work (HuSCF-GAN), and the label distribution-based KLD approach (Guerraoui et al., 2020), which requires clients to share label information with the server—thereby compromising data privacy.

For this evaluation, we focus on the second test case scenario described in Subsubsection 6.1.2, where the dataset originates from a single domain and follows a Non-IID distribution. This setup is deliberately chosen to focus on the effect of the KLD component, ensuring that the evaluation of KLD is not affected by the performance variations introduced by multi-domain settings.

As shown in Figure 17, both approaches converge to the same MNIST Score at a similar rate. Furthermore, Table 17 presents classifier performance metrics for both methods, which are nearly identical. These results demonstrate that the proposed activation-based KLD not only preserves client privacy by avoiding label sharing but also matches the performance of the label-based alternative—thereby offering a more privacy-preserving solution without sacrificing effectiveness.

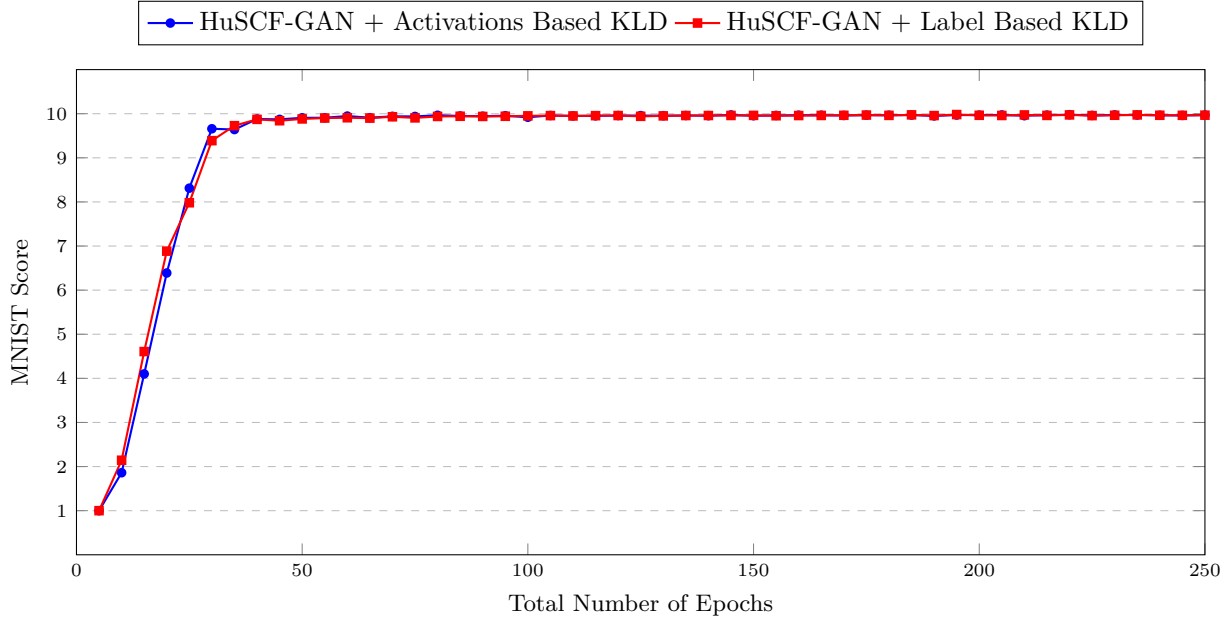

Figure 17: **MNIST Score vs. Training Epochs - Single-Domain Non-IID Data**

Table 17: KLD Comparison – Single-Domain Non-IID Data

| | MNIST Dataset | | | | |
|---|---|---|---|---|---|
| | Accuracy ↑ | Precision ↑ | Recall ↑ | F1 Score ↑ | FPR ↓ |
| HuSCF-GAN + Label-Based KLD | **97.2%±0.32** | 97.19%±0.32 | **97.19%±0.32** | **97.17%±0.33** | **0.31%±0.11** |
| HuSCF-GAN + Activation-Based KLD | 97.17%±0.33 | **97.21%±0.32** | 97.18%±0.32 | 97.15%±0.33 | **0.31%±0.11** |

## 6.4 Complexity Analysis

In this subsection, we discuss the computational (time) and space complexity of **HuSCF-GAN** in comparison with **FedGAN**. All variables and notations used are listed in table 18.

Table 18: Table of Notations and Variables

| Symbol | Description |
|---|---|
| $n_{G_{H,k}}$, $n_{G_{T,k}}$ | Number of parameters in the head and tail of the generator for client $k$ |
| $n_{D_{H,k}}$, $n_{D_{T,k}}$ | Number of parameters in the head and tail of the discriminator for client $k$ |
| $n_{G_{S,i}}$, $n_{D_{S,i}}$ | Number of parameters in the $i$-th server-side layer of the generator and discriminator |
| $n_{G_S}$, $n_{D_S}$ | Total number of parameters for the full server-side generator and discriminator segments |
| $n_G$, $n_D$ | Total number of parameters in the full generator and discriminator, respectively |
| $N_{G,i}$, $N_{D,i}$ | Number of clients per $i$-th layer of the generator and discriminator server segments |
| $N$ | Total number of clients |
| $M$ | Number of federation rounds |
| $I$ | Total number of iterations |
| $b$ | Batch size |
| $A_{G_{H,k}}$, $A_{G_{T,k}}$ | Activation size per sample for the head and tail of the generator for client $k$ |
| $A_{D_{H,k}}$, $A_{D_{T,k}}$ | Activation size per sample for the head and tail of the discriminator for client $k$ |
| $A_{G_{S,i}}$, $A_{D_{S,i}}$ | Activation memory for the $i$-th server-side layer of the generator and discriminator |
| $A_{G_S}$, $A_{D_S}$ | Total activation size for the full server-side generator and discriminator segments |
| $A_G$, $A_D$ | Total activation size for the full generator and discriminator |

### 6.4.1 Computational Complexity

HuSCF-GAN consists of both client-side and server-side components. The model is partitioned such that each client computes the head and tail segments of the generator and discriminator—denoted as $G_H$, $G_T$, $D_H$, and $D_T$—while the server handles the shared segments $G_S$ and $D_S$, in addition to aggregation. We compare the computational complexity of HuSCF-GAN with FedGAN, which places the entire model on the client side.

*Client-Side Computational Complexity*

Each client executes local computations on its assigned model segments. Let $I$ denote the total number of training iterations and $b$ the batch size. Table 19 presents the client-side complexity for both methods. HuSCF-GAN reduces per-client computation by processing only partial model segments, an advantage for resource-limited devices where offloading computation to the server enhances scalability.

It is important to note that $n_G + n_D >> n_{G_{H,k}} + n_{G_{T,k}} + n_{D_{H,k}} + n_{D_{T,k}}$, which means that the load on the client-side in our approach is less, as our approach puts most of the load on the server.

*Server-Side Computational Complexity*

Table 20 summarizes the server-side complexity. While FedGAN's server only performs aggregation, HuSCF-GAN's server also executes part of the forward and backward computations. Although this increases server workload (because it puts a bulk of the generator and discriminator calculations on the server besides the

Table 19: Client-side computational complexity comparison.

| Method | Client-Side Computational Complexity |
|---|---|
| **HuSCF-GAN** | $\mathcal{O}\left( \underbrace{Ib\left(n_{G_{H,k}} + n_{G_{T,k}} + n_{D_{H,k}} + n_{D_{T,k}}\right)}_{\text{Client-side segments training iterations}} \right)$ |
| **FedGAN** | $\mathcal{O}\left( \underbrace{Ib\left(n_G + n_D\right)}_{\text{Client models training iterations}} \right)$ |

aggregation), it substantially reduces client burden (the client no longer need to do the full model calculations), leveraging the server's typically greater computational capacity to improve overall efficiency and scalability.

Table 20: Server-side computational complexity comparison.

| Method | Server-Side Computational Complexity |
|---|---|
| **HuSCF-GAN** | $\mathcal{O}\left( \underbrace{Ib\left(\sum_i N_{G,i} n_{G_{S,i}} + \sum_i N_{D,i} n_{D_{S,i}}\right)}_{\text{Server-side training iterations}} + \underbrace{MN\left(\max_k n_{G_{H,k}} + \max_k n_{G_{T,k}} + \max_k n_{D_{H,k}} + \max_k n_{D_{T,k}}\right)}_{\text{Client-side federated aggregation}} \right)$ |
| **FedGAN** | $\mathcal{O}\left( \underbrace{MN\left(n_G + n_D\right)}_{\text{Client models federated aggregation}} \right)$ |

### 6.4.2 Space Complexity

We now analyze the memory requirements of HuSCF-GAN compared to FedGAN, considering parameter storage and intermediate activations.

*Client-Side Space Complexity*

Each client stores its local model segments ($G_H$, $G_T$, $D_H$, $D_T$). The resulting space complexity is shown in Table 21. HuSCF-GAN lowers client memory consumption by limiting storage to local segments in contrary with FedGAN where clients store the full models. It is important to note $A_G + A_D >> A_{G_{H,k}} + A_{G_{T,k}} + A_{D_{H,k}} + A_{D_{T,k}}$

Table 21: Client-side space complexity comparison.

| Method | Client-Side Space Complexity |
|---|---|
| **HuSCF-GAN** | $\mathcal{O}\left( \underbrace{n_{G_{H,k}} + n_{G_{T,k}} + n_{D_{H,k}} + n_{D_{T,k}}}_{\text{Parameters of client-side segments}} + \underbrace{b(A_{G_{H,k}} + A_{G_{T,k}} + A_{D_{H,k}} + A_{D_{T,k}})}_{\text{Activations of client-side segments}} \right)$ |
| **FedGAN** | $\mathcal{O}\left( \underbrace{n_G + n_D}_{\text{Parameters of client models}} + \underbrace{b(A_G + A_D)}_{\text{Activations of client models}} \right)$ |

*Server-Side Space Complexity*

The server stores parameters and activations for $G_S$ and $D_S$ and retains the maximum client-side parameters during aggregation. The overall complexity is shown in Table 22.

The memory footprint on the server in our approach is larger than FedGAN, because our approach assumes that the server is more capable and can assist the clients in their computations.

Table 22: Server-side space complexity comparison.

| Method | Server-Side Space Complexity |
|---|---|
| **HuSCF-GAN** | $\mathcal{O}\Big( \underbrace{\sum_i (n_{G_{S,i}} + n_{D_{S,i}})}_{\text{Parameters of server-side segments}} + b\underbrace{\sum_i (N_{G,i}A_{G_{S,i}} + N_{D,i}A_{D_{S,i}})}_{\text{Activations of server-side segments}} + \underbrace{\max_k(n_{G_{H,k}} + n_{G_{T,k}} + n_{D_{H,k}} + n_{D_{T,k}})}_{\text{Activations of client-side segments for aggregation}} \Big)$ |
| **FedGAN** | $\mathcal{O}\Big( \underbrace{n_G + n_D}_{\text{Parameters of client models for aggregation}} \Big)$ |

HuSCF-GAN reduces the memory footprint on clients while slightly increasing server memory usage due to shared segment computation and aggregation. This trade-off enhances scalability and system efficiency in federated learning environments.

# 7 Discussion

## 7.1 Limitations

In regards to our approach, several limitations should be acknowledged.

- **Centralized dependency:** Although HuSCF-GAN leverages heterogeneous and underutilized devices, it does not operate in a fully decentralized manner. The framework still relies on a central server to support these devices and to handle critical tasks such as clustering, federation, and the execution of genetic algorithms. The server is also responsible for performing the forward and backward pass computations for the server-side layers.

- **Privacy vulnerabilities:** While our approach avoids sharing raw data or labels, it may still be vulnerable to certain privacy-related threats. In particular, the system could be susceptible to attacks such as data reconstruction and label inference, where adversaries attempt to extract sensitive information from shared gradients or intermediate representations.

- **System complexity and scalability:** Our approach integrates multiple components—including a Genetic Algorithm, Federated Learning, U-shaped Split Learning, Kullback-Leibler Divergence (KLD) calculations, and clustering—which collectively introduce additional complexity. For example, selecting the optimal cut point can create computational overhead, particularly in large-scale environments where many device profiles may introduce variability and unwanted delays.

- **Hyperparameter tuning:** Tuning the hyperparameters for these components across a large number of devices can be challenging. Such tuning not only increases computational burden but may also prolong convergence times, especially as the network scales to thousands of devices. These factors represent added complexity that can affect the scalability and responsiveness of the system.

Recognizing these limitations is essential for guiding future improvements to enhance both the robustness and security of HuSCF-GAN.

## 7.2 Possible Risks & Mitigation Strategies

While our work enables distributed GAN training in heterogeneous, multi-domain environments under data-sharing constraints using split federated learning, it is important to recognize that such systems may be vulnerable to various types of attacks (Shabbir et al., 2025). In this subsection, we discuss potential threats and outline strategies to mitigate them.

One critical category of threats is **data reconstruction attacks**, where an adversary attempts to recover private training data by exploiting shared gradients or activations exchanged between end devices and the server. This risk is particularly serious because preserving data privacy is a central goal of our approach. Examples of such attacks include:

- **Feature Space Hijacking Attacks**: A malicious server manipulates feature representations to align with a target feature space that it knows how to invert (Pasquini et al., 2021; Gawron & Stubbings, 2022).

- **Model Inversion Attacks**: An honest-but-curious server records smashed activations and uses optimization techniques or generative models to reconstruct approximations of clients' raw inputs (Erdoğan et al., 2022).

- **Feature Reconstruction Attacks**: Adversaries attempt to recover input data directly from intermediate feature representations (Xu et al., 2024; Ye et al., 2024).

- **GAN-based Attacks in Split Learning**: An adversary trains a generator to produce fake inputs whose feature representations or gradients closely mimic those observed during training (Zeng et al., 2025).

Another class of threats involves **label inference attacks**, where adversaries seek to infer sensitive label information from client updates or intermediate activations. Examples include:

- **Gradient-based Label Inference**: Correlations between gradient updates and labels are exploited to extract hidden label information (Liu et al., 2024; Xie et al., 2023; Zhao et al., 2024; Bai et al., 2023; Kariyappa & Qureshi, 2023).

- **Smashed Data-based Label Inference**: Relationships between intermediate activations and label distributions are leveraged to infer labels, for instance through clustering or distance-based methods (Zhu et al., 2023; Liu et al., 2024).

Other possible threats include **adversarial attacks** and **model poisoning attacks** (Wu et al., 2024; He et al., 2024; Gamal et al., 2023), which can disrupt model performance or corrupt its learned representations.

To address these risks, several mitigation strategies can be applied to strengthen the security and privacy of our framework:

- Employing **homomorphic encryption** to protect data during transmission and computation (Yi et al., 2014; Kokaj & Mollakuqe, 2025; Khan et al., 2023).

- Applying **differential privacy** techniques to perturb data or gradients, thereby limiting information leakage (Dwork, 2006; Pham et al., 2024).

- Using **function secret sharing** to securely split and process sensitive data (Khan et al., 2024).

- Introducing **randomized activations or layers** to make feature representations less predictable (Mao et al., 2023).

- Implementing **detection mechanisms** for weight or gradient manipulation and anomaly detection (Fu et al., 2023; Erdogan et al., 2022; Erdoğan et al., 2024).

Incorporating these mitigation strategies would further enhance the robustness and privacy-preserving capabilities of our proposed approach.

### 7.3 Applicability to Other Generative Models

The HuSCF-GAN framework can be extended to a wide range of other generative models, including transformers, diffusion models, and large language models (LLMs). The core methodology remains the same: both the input and output of the model—as well as any labels, if applicable—are kept strictly on the client side to preserve data privacy, while distributing the computational load on different clients according to their capabilities.

The primary adjustment lies in the number of cut points required. GANs, with their dual-network architecture, necessitate four cut points. In contrast, architectures with a single network, such as encoder-only networks (e.g BERT (Devlin et al., 2019)) or decoder-only networks (e.g GPT (Radford et al., 2018)), would only require two cut points—one near the input and another near the output to keep the data and the labels on the client—However, the framework should allow for dynamic number of cutpoints in the future. In this setup, the initial portion and final portion of the model remain on the client devices, ensuring that input data and generated outputs (e.g., predictions or text) never leave the client side. However, our framework would allow for a dynamic number of cutpoints according to the application needs, and devices complexity.

Applying this approach to LLMs could enable training on vast amounts of text data stored on underutilized personal devices, such as smartphones, while simultaneously taking advantage of their unused computational resources. Furthermore, the clustering component of our framework could group different LLM instances by domain—for example, medical texts, geographic data, or general knowledge—allowing for the creation of specialized models tailored to specific fields.

Similar to GANs in our approach, these models would share portions of their intermediate parameters, enabling them to learn collaboratively while still specializing in their respective domains. The same principles can also be extended to diffusion models (Ho et al., 2020), vision transformers (Dosovitskiy et al., 2020), and other modern generative architectures.

### 7.4 Illustrative Use Case

With the rapid proliferation of AI-powered assistants and copilots (Stratton, 2024) across domains—from personal and domestic helpers to industrial, medical, and autonomous systems—there is a growing need for these assistants (copilots) to perceive, understand, and generate visual content in real time. These systems are increasingly equipped with sensing and vision capabilities, continuously interacting with their environments and users.

To illustrate the practical impact of our proposed **HuSCF-GAN** framework, we consider a use case involving a network of such intelligent assistants or copilots equipped with vision systems that continuously capture and generate image data. These assistants operate in distinct environments—such as domestic, industrial, or outdoor settings—leading to naturally diverse and highly non-IID image distributions spanning multiple domains. Beyond interpreting visual information, they must also generate synthetic images for purposes such as environment simulation, object recognition enhancement, visual communication with humans, or training on visual tasks involving human collaboration. In this context, **HuSCF-GAN** enables collaborative training among assistants without requiring centralized data sharing, allowing each assistant to learn from collective visual knowledge while keeping its local image data and labels private. By utilizing underused computational resources on the assistants themselves, the framework minimizes infrastructure costs of relying on a central server. It dynamically adjusts participation and update frequencies according to each assistant's computational capacity and data characteristics, ensuring balanced and stable model convergence. Furthermore, its domain-clustering mechanism groups assistants based on visual domain similarities—for instance, indoor versus outdoor imagery—while still maintaining shared global learning to capture cross-domain visual patterns. Through this approach, **HuSCF-GAN** could provide a privacy-preserving, resource-efficient, and scalable solution for collaborative image generation and understanding across heterogeneous assistants, ultimately improving their visual perception and generative capabilities in real-world environments. We believe that **HuSCF-GAN** is a good fit in this example or other examples proposed by the research community

## 8 Conclusion & Future Work

Centralized generative models—such as traditional Generative Adversarial Networks (GANs)—encounter several critical limitations when applied in real-world, distributed environments. One major challenge is data diversity: in practice, most client devices retain their data locally due to privacy concerns, leading to limited access to the full distribution of data and reducing the generalization ability of centralized models. Another issue is the inefficient use of computational resources, where many edge devices, IoT devices, and wearables remain underutilized while powerful centralized servers perform the bulk of training.

To address these limitations, distributed Generative AI training has emerged as a promising paradigm. However, this approach introduces its own set of challenges, such as data heterogeneity (clients possess non-IID data that may differ significantly in distribution), device heterogeneity (clients differ in computational capabilities and network speeds), and domain disparity (client datasets may originate from entirely different domains or modalities). Moreover, the presence of strict data sharing constraints further complicates the design of collaborative learning systems.

In this paper, we propose **HuSCF-GAN**, a novel *Heterogeneous U-Shaped Clustered Federated Generative AI* approach—implemented using a conditional GAN (cGAN) as a proof of concept—that systematically addresses key challenges in federated learning with generative models. Our method partitions the model architecture such that different components are trained across clients and the server, enabling flexible model splitting based on client capabilities and communication bandwidth. By adapting to both data and system heterogeneity, **HuSCF-GAN** significantly improves training efficiency and model performance.

Through extensive experiments, we demonstrate that **HuSCF-GAN** outperforms state-of-the-art benchmarks across multiple datasets and experimental settings. It achieves superior classification performance—across metrics such as Accuracy, Precision, Recall, F1 Score, and False Positive Rate—with an average improvement of 10%, and up to a 60% gain in multi-domain, non-IID environments. In terms of image generation quality, **HuSCF-GAN** achieves between $1.1\times$ and $3\times$ improvement in generation scores for the MNIST family, and between $2\times$ and $70\times$ lower FID scores for higher resolution datasets. Furthermore, it substantially reduces latency in heterogeneous and resource-constrained environments, achieving at least a $5\times$ and up to a $58\times$ reduction compared to existing benchmarks.

Potential directions for future research include the following:

- Distributing the generative model across multiple edge devices without relying on a central server. This would involve selecting a dynamic number of cut points based on the number of available devices, rather than using a fixed number (four, in our case). Such an approach enables full reliance on underutilized low-power devices, eliminating the need for the centralized infrastructure avoiding its costs.

- Optimizing cut point selection based on factors such as energy consumption, data quality and quantity at each node, and the expected battery lifetime of the devices.

- Make dynamic cut selection throughout training to adapt to dynamically changing devices capabilities and configurations.

- Incorporating privacy-preserving techniques such as Differential Privacy or Homomorphic Encryption to enhance data security during training.

- Extending the approach to other generative architectures, such as diffusion models, transformers, or large language models (LLMs), to evaluate its generalizability.

- Evaluating the proposed system on a physical testbed rather than relying solely on simulation, to validate performance under real-world conditions.

- Evaluation the framework on more complex dataset (Higher resolution images) and more diverse modalities (e.g Time-series, text, and 3D objects)

- Investigating alternative generation evaluation metrics tailored to various data modalities

**Broader Impact Statement**

Our work on **HuSCF-GAN** advances distributed generative AI by enabling deployment across heterogeneous devices with varying computational capabilities, diverse data distributions, and multiple domains. It leverages idle resources and can access data constrained by privacy or sharing limitations.

However, this approach carries potential risks, including the possibility that shared activations or gradients could be exploited to reconstruct sensitive input data. We emphasize that careful deployment is essential and

strongly recommend incorporating additional privacy-preserving techniques, such as homomorphic encryption, function secret sharing, differential privacy, and zero-knowledge proofs (ZKPs), to mitigate these risks and ensure responsible use of the technology.

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

## A  Ablation Study on HuSCF-GAN Components

In this section, we present the ablation study conducted to evaluate the contributions of different components of **HuSCF-GAN** toward overall model performance. Specifically, we investigate the impact of: (i) the **Clustering Component**, (ii) the **KLD Component for intra-cluster weighting** to address non-IID data, and (iii) using **both components together**. To evaluate these configurations, we select the **Two-Domain Highly Non-IID** test case under varying conditions.

As illustrated in Figures 18a and 18b, image generation performance is primarily driven by the clustering component. The removal of the KLD weighting component—which is responsible for handling intra-cluster non-IID characteristics—has a marginal effect on performance. However, removing the clustering component results in a significant performance drop. In such a case, the model tends to bias towards the MNIST dataset, leading to a much higher MNIST score compared to FMNIST.

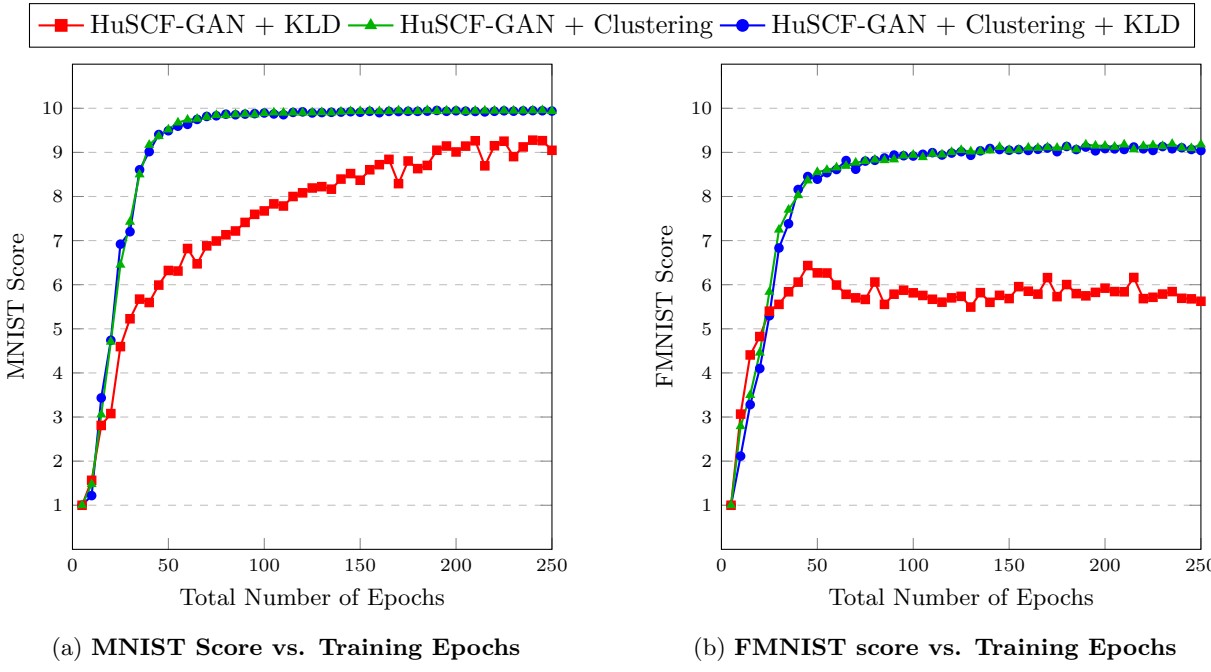

(a) **MNIST Score vs. Training Epochs**    (b) **FMNIST score vs. Training Epochs**

Figure 18: Image Generation Scores — Two-Domains Highly Non-IID Data

Table 23 further supports these findings. The highest evaluation metrics are achieved when both the clustering and KLD weighting components are enabled. Notably, the clustering component has a more substantial impact. Removing it causes a significant reduction (over 20%) in FMNIST performance due to the model overfocusing on the easier MNIST dataset. Although the KLD component plays a smaller role, it still contributes positively: removing it causes approximately a 1% drop in the evaluation metrics across both datasets. The relatively smaller impact is attributed to the shared layers among clients in the model architecture, which reduces the negative effects of intra-cluster variance. Nonetheless, the combination of both components yields the optimal performance.

Table 23: Ablation Study - Two-Domains Intense Non-IID Data

| | MNIST Dataset | | | | | FMNIST Dataset | | | | |
|---|---|---|---|---|---|---|---|---|---|---|
| | Accuracy ↑ | Precision ↑ | Recall ↑ | F1 Score ↑ | FPR ↓ | Accuracy ↑ | Precision ↑ | Recall ↑ | F1 Score ↑ | FPR ↓ |
| HuSCF-GAN + KLD | 94.81%±0.47 | 94.80%±0.47 | 94.79%±0.47 | 94.75%±0.47 | 0.58%±0.04 | 62.00%±0.97 | 65.50%±0.96 | 62.00%±0.97 | 61.69%±0.96 | 4.22%±0.39 |
| HuSCF-GAN + Clustering | 95.03%±0.44 | 95.07%±0.44 | 95.04%±0.44 | 95.01%±0.44 | 0.49%±0.04 | 80.10%±0.57 | 80.50%±0.57 | 80.03%±0.57 | 80.26%±0.57 | 2.17%±0.20 |
| HuSCF-GAN + KLD + Clustering | **96.15%±0.45** | **96.11%±0.45** | **96.10%±0.45** | **96.10%±0.45** | **0.45%±0.04** | **81.46%±0.55** | **81.32%±0.55** | **81.46%±0.55** | **80.61%±0.55** | **1.95%±0.19** |

# B Ablation Study on Genetic Algorithm Hyperparameters

In this section, we present an ablation study to evaluate the impact of different hyperparameters of the genetic algorithm, namely Population Size ($PS$), Crossover Rate ($CR$), and Mutation Rate ($MR$). The ablation is performed by varying each hyperparameter individually while keeping the others fixed at their baseline values.

Since the genetic algorithm is used to select the cutpoints of the model, it significantly affects the system latency. Therefore, we use latency as the primary metric for this study. While changes in the genetic algorithm parameters have minimal impact on other performance metrics, their effect on latency is pronounced and readily observable.

Table 24: **Ablation Study on Genetic Algorithm Hyperparameters**

| GA Hyper-parameters | $PS = 1000$ $CR = 0.7$ $MR = 0.01$ | $PS = 1000$ $CR = 0.3$ $MR = 0.01$ | $PS = 1000$ $CR = 0.5$ $MR = 0.01$ | $PS = 1000$ $CR = 0.9$ $MR = 0.01$ | $PS = 1000$ $CR = 0.7$ $MR = 0.001$ | $PS = 1000$ $CR = 0.7$ $MR = 0.05$ | $PS = 1000$ $CR = 0.7$ $MR = 0.1$ | $PS = 100$ $CR = 0.7$ $MR = 0.01$ | $PS = 500$ $CR = 0.7$ $MR = 0.01$ | $PS = 2000$ $CR = 0.7$ $MR = 0.01$ |
|---|---|---|---|---|---|---|---|---|---|---|
| Latency (s) | **7.8** | 8.21 | **7.8** | **7.8** | **7.8** | 8.3 | 9.7 | 8.22 | 7.9 | 8.13 |

Table 24 shows that our chosen hyperparameters achieve the lowest latency. However, other parameter selections yield comparable latency, suggesting that multiple configurations could be used without compromising performance. The results indicate that reducing the population size (e.g., to 500 or below) leads to a noticeable degradation in the objective function (latency). This is because a smaller population reduces solution diversity, thereby restricting the exploration capability of the algorithm. Conversely, increasing the population size excessively (e.g., beyond 2000) yields minimal latency improvement while substantially increasing computational cost. Similarly, low crossover rates ($\leq 0.3$) hinder performance by limiting genetic diversity, whereas excessively high rates ($\geq 0.9$) offer negligible gains while adding computational overhead. Regarding mutation, higher rates ($\geq 0.1$) degrade performance due to excessive randomness, while very low rates ($\leq 0.001$) increase the risk of premature convergence to local optima.

Overall, these observations validate our initial hypothesis that an appropriate balance between exploration and efficiency is crucial—one that is effectively achieved through our selected hyperparameter configurations.

# C Further Comparison Between HuSCF-GAN and PFL-GAN

This section presents an extended comparison between HuSCF-GAN and PFL-GAN, based on the evaluation scenarios introduced in the original PFL-GAN paper (Wijesinghe et al., 2023). Specifically, two test cases are considered: the *label skewness* and the *Byzantine* scenarios. Both consist of 20 clients, each having 300 samples per label, except for 3 randomly chosen labels that only have 15 samples each. The key difference is that the label skewness scenario uses clients from a single domain (MNIST), while the Byzantine scenario includes 10 MNIST clients and 10 FMNIST clients. In both scenarios, PFL-GAN achieves scores similar to those shown in its original paper.

In the first scenario, as shown in Figure 19, HuSCF-GAN achieves slightly higher image generation scores compared to PFL-GAN. Additionally, HuSCF-GAN yields approximately 1.5% improvement across classification metrics, as reported in Table 25.

In the second scenario, summarized in Table 26, HuSCF-GAN outperforms PFL-GAN in classification metrics on the MNIST dataset, with an improvement of approximately 1%. On the FMNIST dataset, HuSCF-GAN performs comparably, with a slight decrease in scores. Figure 20 shows that HuSCF-GAN also achieves marginally better image generation quality in this setting.

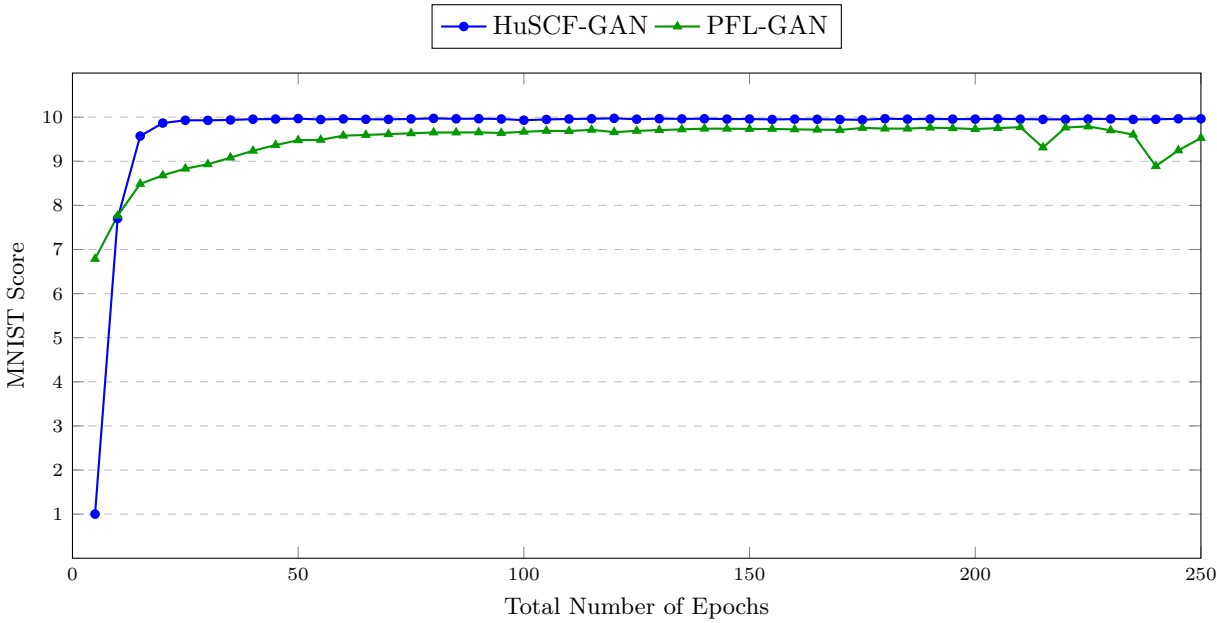

Figure 19: **MNIST Score vs. Training Epochs** — Comparison between HuSCF-GAN and PFL-GAN — label skewness scenario (Wijesinghe et al., 2023).

Table 25: Further Comparison between PFL-GAN & HuSCF-GAN — Label Skewness Scenario

| | MNIST Dataset | | | | |
|---|---|---|---|---|---|
| | Accuracy ↑ | Precision ↑ | Recall ↑ | F1 Score ↑ | FPR ↓ |
| PFL-GAN | 97.18%±0.46 | 97.21%±0.46 | 97.18%±0.46 | 97.17%±0.46 | 0.31%±0.11 |
| HuSCF-GAN | **98.11%±0.44** | **98.13%±0.44** | **98.08%±0.44** | **98.10%±0.44** | **0.21%±0.09** |

Table 26: Further Comparison between PFL-GAN & HuSCF-GAN — Byzantine Scenario

| | MNIST Dataset | | | | | FMNIST Dataset | | | | |
|---|---|---|---|---|---|---|---|---|---|---|
| | Accuracy ↑ | Precision ↑ | Recall ↑ | F1 Score ↑ | FPR ↓ | Accuracy ↑ | Precision ↑ | Recall ↑ | F1 Score ↑ | FPR ↓ |
| PFL-GAN | 96.28%±0.39 | 96.31%±0.39 | 96.28%±0.39 | 96.27%±0.39 | 0.41%±0.04 | **84.75%±0.36** | **84.75%±0.36** | **84.65%±0.36** | **84.58%±0.36** | **1.69%±0.25** |
| HuSCF-GAN | **97.75%±0.33** | **97.76%±0.33** | **97.75%±0.33** | **97.75%±0.33** | **0.25%±0.26** | 83.92%±0.37 | 83.95%±0.37 | 83.92%±0.37 | 83.44%±0.37 | 1.79%±0.26 |

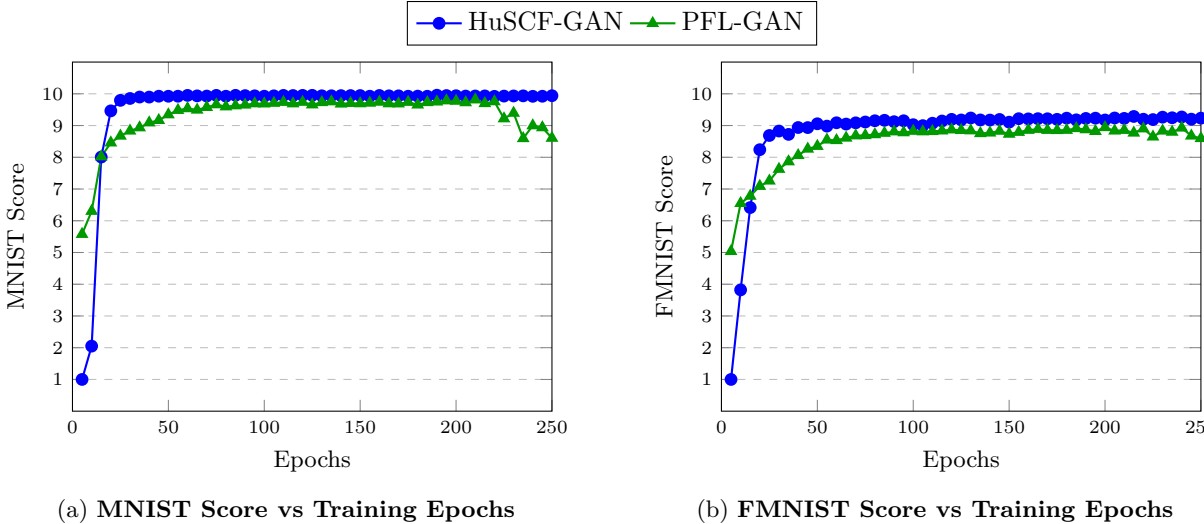

(a) **MNIST Score vs Training Epochs**  (b) **FMNIST Score vs Training Epochs**

Figure 20: Comparison of HuSCF-GAN and PFL-GAN — Byzantine Scenario (Wijesinghe et al., 2023).

# D  Further Elaboration on the Reduction Strategy

## D.1  Overview

To address the scalability limitations of the client-based Genetic Algorithm (GA), we introduce a *profile-based reduction strategy*. In this approach, instead of optimizing over all individual clients, the search space is reduced to a set of unique *device profiles*, where each profile groups clients that share identical computational and communication characteristics (see Figure 6). The GA then operates over these profiles rather than individual clients, effectively transforming the optimization process from client-level to profile-level.

## D.2  Reduction Principle and Scalability

In the client-based configuration, the GA treats every client as an independent entity in the search space. While this ensures maximum fidelity to the real system, the resulting search space grows exponentially with the number of clients, leading to high computational complexity and slow convergence.

The profile-based strategy, in contrast, achieves a *structural reduction* rather than a quantitative one. By aggregating clients with identical characteristics into common profiles, the GA optimizes over a search space that scales with the number of profiles rather than the number of clients. Since the number of profiles typically remains much smaller than the total number of devices, even in large-scale deployments, this strategy substantially reduces computational cost while preserving the representativeness of client behaviors.

## D.3  Impact on Algorithmic Complexity

As the number of clients increases, the computational complexity of the profile-based strategy remains largely constant, because it scales with the number of profiles rather than the total number of clients. In this setting, the GA operates on a compact search space defined by profile-level configurations, resulting in a much lower time and space complexity compared to the client-based GA.

Although this work assumes that all clients within a profile have identical computational and communication capabilities, this assumption serves as a proof of concept. In realistic large-scale systems, clients can still be effectively clustered into representative profiles based on similarity, keeping the number of profiles limited and ensuring the scalability of the approach.

### D.4   Impact on Solution Quality

As the number of clients scales, the profile-based reduction strategy consistently maintains—and in some cases improves—the quality of the obtained solutions while drastically reducing computational complexity. Because the GA operates over a smaller yet more structured search space, it converges faster and more stably without degrading performance.

Experimental results with 100 devices illustrate this effect clearly as shown in Table 27: the profile-based GA achieved the lowest latency within the smallest number of generations, outperforming the client-based GA both in convergence speed and final latency. This improvement arises from the reduced and more structured search space, which allows the GA to explore more efficiently and avoid being trapped in local optima—an issue common in large, complex search spaces.

Table 27: Comparison between profile-based and client-based GA performance using 100 devices.

| Strategy | Achieved Latency (s) | Generations to Convergence |
|---|---|---|
| Profile-based | 7.8 | 12 |
| Client-based | 8.26 | 488 |

### D.5   Effect of System Synchronicity

**Assumption:** The system is assumed to operate in a synchronous setting, where the server waits for all clients participating in a given layer—effectively for the slowest client—before proceeding. Under this assumption, the profile-based strategy maintains reachability to all solutions achievable by the client-based GA. Any configuration change that could improve overall latency either (i) provides no net benefit when applied to individual clients within a profile, or (ii) can be equivalently represented at the profile level, since all clients within a profile share identical computational and communication characteristics.

If the system were instead asynchronous, where the server does not wait for slower clients, some configurations accessible to the client-based GA might not be representable at the profile level. However, even under such a setting, the profile-based strategy is expected to achieve reasonable (even if not the best) latency while preserving significantly lower computational cost.

### D.6   Effect of Profiling Assumptions

The performance of the profile-based strategy depends on the assumption regarding client similarity within each profile. In this work, we assume that clients in the same profile are identical in their computational and communication characteristics. Under this assumption, there is no information loss, and the profile-based GA can explore the same solution space as the client-based GA while maintaining quality and reachability.

In more realistic scenarios, this assumption can be relaxed by clustering clients with similar (rather than identical) characteristics into shared profiles. This introduces a minor approximation error, as each profile represents an averaged abstraction of its clients. Nevertheless, the resulting model remains highly effective, delivering near-optimal solutions with a fraction of the computational effort required by client-level optimization, making it a scalable and practical approach for large deployments.

### D.7   Summary

In summary, the profile-based reduction strategy offers a highly scalable and efficient alternative to client-level GA optimization. By operating over representative device profiles rather than individual clients, it significantly reduces computational complexity while preserving solution quality and convergence properties.

