# OpenReview forum: "A Distributed Generative AI Approach for Heterogeneous Multi-Domain Environments under Data Sharing constraints"
_TMLR — Accepted by TMLR_

### Review · Reviewer_tuMr · 2025-08-30

**Summary Of Contributions:**

This paper introduces **HuSCF-GAN (Heterogeneous U-Shaped Split Clustered Federated GANs)**, a distributed generative AI framework designed for heterogeneous, multi-domain environments under strict data sharing constraints. The work tackles four central challenges in decentralized GAN training:

1. Data heterogeneity (non-IID distributions)
2. Device heterogeneity (compute + network variation)
3. Multi-domain datasets
4. Strict data sharing constraints (no raw/synthetic data, no labels shared)

**Key contributions:**
- **Novel methodology:** Combines a genetic algorithm for cut-point optimization, heterogeneous U-shaped split learning for privacy-preserving training, and clustered KLD-weighted federated learning for adapting to domain differences.
- **Latency model + optimization:** Explicit mathematical model of client/server compute and communication costs, optimized with a genetic algorithm to minimize per-iteration latency.
- **Privacy-preserving aggregation:** Proposes activation-based KLD scoring (vs. prior label-based methods) to avoid label sharing while achieving equivalent performance.
- **Extensive evaluation:** Across six scenarios (single-domain IID/non-IID, multi-domain IID/non-IID, highly non-IID, and four-domain scalability), HuSCF-GAN consistently outperforms baselines (MD-GAN, FedGAN, Federated Split GANs, HFL-GAN, PFL-GAN):
  - $1.1 \times$–$2.2 \times$ higher image generation scores
  - $10\%$–$50\%$ higher classification metrics in challenging multi-domain non-IID settings
  - $5 \times$–$58 \times$ lower training latency, especially on resource-constrained devices

**Strengths:**
- Addresses all four critical challenges jointly (a first among distributed GAN works).
- Strong technical rigor: clear latency model, formalized optimization, and privacy-preserving weighting scheme.
- Extensive and fair evaluation across multiple datasets, baselines, and scenarios.
- Practical relevance to federated learning in IoT/edge-device ecosystems.

**Weaknesses:**
- Experiments are limited to relatively small image datasets (MNIST-family). The scalability to modern high-dimensional data (e.g., CIFAR, ImageNet, medical images) remains untested.
- The approach depends on a central server for middle-layer hosting, which may limit its applicability in fully decentralized settings.
- Privacy is preserved at the label/data level, but advanced adversarial leakage attacks (e.g., gradient inversion) are not discussed.

**Additional Comments:**

- The paper is well-written and structured, though dense in equations. Adding more **intuitive diagrams** (e.g., GA optimization process, communication flow) would improve accessibility.
- The results section is strong; however, including **confidence intervals or statistical significance testing** would further substantiate performance claims.
- Future work suggestions are thoughtful; in particular, exploring applicability to **diffusion models and LLMs** is an exciting direction that would extend the contribution substantially.

**Audience:**

Yes

**Audience Explanation:**

The paper addresses a timely and important problem at the intersection of **federated learning, distributed training, and generative AI**. TMLR’s readership, which includes researchers in machine learning systems, privacy-preserving ML, and generative modeling, would find this work relevant because:

- It introduces a **novel integration** of split learning, federated learning, and clustering in generative model training.
- It offers a practical framework for **edge/IoT deployment**, a key application domain of interest.
- It provides methodological advances (latency-optimized cut-point selection, activation-based privacy-preserving KLD) that could inspire further research.

The findings advance both the **theoretical understanding** of distributed GANs and their **practical feasibility** under real-world constraints.

**Broader Impact Concerns:**

- The work has **positive societal potential**: enabling privacy-preserving generative AI on distributed, resource-limited devices, including healthcare and sensitive data domains.
- **Concerns:**
  - While data/label privacy is preserved, information leakage via intermediate activations or gradients could still be exploited by adversaries.
  - No explicit safeguards (e.g., differential privacy, homomorphic encryption) are integrated in the current implementation.
- **Recommendation**: The paper should expand its Broader Impact section to explicitly discuss these risks and possible mitigations.

**Claims And Evidence:**

Yes

**Claims Explanation:**

The experiments systematically test the algorithm in conditions ranging from favorable (IID single-domain) to extremely challenging (multi-domain, highly non-IID). In simple scenarios, HuSCF-GAN performs on par with baselines, confirming correctness. In complex scenarios, it clearly outperforms thereby validating its design choices (heterogeneous splitting, clustering, KLD-weighted FL).

The latency analysis is especially convincing. Unlike many works that ignore communication/computation tradeoffs, this paper explicitly models them and validates with empirical device simulations. The privacy claims are also substantiated via comparison with label-based approaches.

**Requested Changes:**

**Suggested (would strengthen but not mandatory):**
1. **Evaluation on larger-scale datasets:** Demonstrating performance beyond MNIST-family datasets (e.g., CIFAR-10/100 or medical imaging datasets) would strengthen claims of generalizability.
2. **Privacy discussion:** While the method avoids raw data and label sharing, it should explicitly address risks of **gradient inversion attacks** or adversarial leakage from activations/gradients, and outline mitigation strategies.
3. **Comparison with diffusion or transformer-based generative models:** Even a brief discussion of applicability would broaden the paper’s scope.
4. **Real-world latency validation:** A small-scale physical deployment (even with 10–20 devices) would better ground the latency claims beyond simulations.
5. **Ablation on GA hyperparameters:** Sensitivity analysis for population size, mutation/crossover rates, and $\beta$ (in weighting scheme) would provide more robustness evidence.

---

> ### Author Response · Authors · 2025-11-08
> **Response to Reviewer tuMr on Evaluation on Larger Scale Datasets or Medical Imaging**
>
> **Regarding the mentioned weakness:** *Experiments are limited to relatively small image datasets (MNIST-family). The scalability to modern high-dimensional data (e.g., CIFAR, ImageNet, medical images) remains untested.*
>
> and **The mentioned requested change:** *Evaluation on larger-scale datasets: Demonstrating performance beyond MNIST-family datasets (e.g., CIFAR-10/100 or medical imaging datasets) would strengthen claims of generalizability.*
>
> ### Response:
>
> We sincerely thank the reviewer for this valuable observation and for emphasizing the importance of demonstrating scalability to more complex, high-dimensional datasets. In our initial submission, we focused primarily on the **MNIST-family** datasets as a **proof of concept (PoC)** to establish the fundamental efficiency and stability of our proposed approach. These datasets allowed for rapid prototyping, controlled experimentation, and clear interpretability of the model’s underlying behaviors.
>
> We fully agree with the reviewer that this setting constitutes a limitation and that testing on larger and more complex datasets is essential to substantiate the **generalizability** and **robustness** of our method. Accordingly, we have now extended our experimental evaluation to include two new large-scale, non-IID multi-domain scenarios:
>
> 1. **High-resolution vision domain:** a two-domain setup using **CIFAR-10** and **SVHN**, representing highly **non-IID** and **heterogeneous sources**. In this experiment, we adopt the **FID score** as the main generative quality metric, as it is more suitable for higher-resolution data compared to the MNIST-family datasets. Our proposed **HuSCF-GAN** consistently achieves substantially lower FID scores, outperforming all other algorithms with margins ranging from **2× to 70× lower values**. Moreover, our approach attains the highest **classification performance**, with improvements ranging from **20% to 45% on CIFAR-10** and **10% to 60% on SVHN** compared to competing methods. These findings demonstrate the **robustness** and **adaptability** of **HuSCF-GAN** to complex, high-resolution, and heterogeneous data.
>
> 2. **Medical imaging domain:** a two-domain non-IID scenario using **BloodMNIST** and **DermaMNIST** datasets from the **MedMNIST** collection, to illustrate **real-world applicability in healthcare contexts**. Here, we continue to employ the **Inception-like data-specific generation score** due to the domain-specific image structures. Our method consistently outperforms all competing approaches across both datasets and evaluation metrics. While **PFL-GAN** achieves competitive results, it struggles under non-IID distributions. In contrast, **HuSCF-GAN** attains between **1.2× and 3× higher generation scores**, and delivers improvements in **classification metrics** ranging from **9% to 70%** over the other methods.
>
> Moreover, we have also updated **the abstract and the conclusion** of the paper accordingly to reflect the new results. Furthermore, we have revised the **Conclusion and Future Work Section 8** to explicitly outline directions for expanding this research, including:
>
> - **Evaluating the framework on more complex datasets** (higher-resolution images) and diverse modalities (e.g., time-series, text, and 3D objects).
> - **Investigating alternative generative evaluation metrics** tailored to different data modalities.
>
> We hope these additional experiments adequately address the reviewer’s concern and convincingly demonstrate the **scalability**, **flexibility**, and **real-world applicability** of our proposed approach beyond the MNIST-family datasets.

---

> ### Author Response · Authors · 2025-11-08
> **Response to Reviewer tuMr on Reliance on a central server**
>
> **Regarding the mentioned weakness:** *The approach depends on a central server for middle-layer hosting, which may limit its applicability in fully decentralized settings.*
>
> ### Response:
>
> We thank the reviewer for this insightful comment. We acknowledge that **HuSCF-GAN** relies on a **central server** to perform key coordination tasks such as **central layers computations**, **clustering**, **federated aggregation**, and execution of the **genetic algorithm**. Therefore, it does not function in a fully decentralized manner.
>
> Our rationale for describing the approach as **distributed** follows established conventions in the literature, where numerous studies have referred to architectures employing a central server—yet distributing data or portions of computation across multiple devices through **federated** or **split learning**—as **distributed** or **decentralized**. In such contexts, the term **distributed** reflects the dispersion of **data**, **computation**, or both across multiple nodes, even when a coordinating server is present.
>
> Nevertheless, we agree that **HuSCF-GAN** is **not fully decentralized**, and we now explicitly acknowledge this as a limitation in the **Limitations subsection (Section 7.1)** of the **Discussion**. Furthermore, we have clarified that the proposed framework adopts a **federated** rather than a fully decentralized topology.
>
> We have also included that as a direction for **future work**, the framework should adapt to work **without the aid of a central server** by dynamically adjusting the **selection of cutpoints** and distributing computation on **end devices only**.
>
> For additional clarification and contextual references, please refer to our ([Response to Reviewer 4ec9 on reliance on a central server](https://openreview.net/forum?id=rpbL7pfPYH&noteId=okIgcAeixz)).
>
> We hope that our response was satisfactory and addressed your comments.

---

> ### Author Response · Authors · 2025-11-08
> **Response to Reviewer tuMr on privacy discussion (1/2)**
>
> **Regarding the mentioned weakness:** *Privacy is preserved at the label/data level, but advanced adversarial leakage attacks (e.g., gradient inversion) are not discussed.*
>
> and **The mentioned requested change:** *Privacy discussion: While the method avoids raw data and label sharing, it should explicitly address risks of gradient inversion attacks or adversarial leakage from activations/gradients, and outline mitigation strategies.*
>
> ### Response
>
> We thank the reviewer for highlighting this important aspect of **privacy**. In the revised manuscript, we have added **Subsection 7.2** in the **Discussion** section to explicitly address potential privacy risks and outline mitigation strategies. While our approach avoids sharing **raw data** and **labels**—thus preserving privacy at the data level—it is important to acknowledge that systems based on **split federated learning** may still be vulnerable to **advanced adversarial attacks**, including **gradient inversion** and other forms of leakage from **intermediate activations** or **gradients**.
>
> In this subsection, we discuss several classes of potential threats:
>
> - **Data reconstruction attacks**, such as feature space hijacking, model inversion, feature reconstruction, and GAN-based attacks, where adversaries attempt to recover private client data from shared activations or gradients [1–6].
>
> - **Label inference attacks**, including gradient-based and smashed-data-based methods, where adversaries attempt to infer sensitive label information from client updates or intermediate activations [7–9].
>
> - **Other threats**, including adversarial or model-poisoning attacks that can disrupt model performance or corrupt learned representations [10–11,17].
>
> To mitigate these risks, we outline several strategies [12–16], such as:
>
> - Employing **homomorphic encryption** to protect data in transit.
> - Applying **differential privacy** to perturb gradients or activations.
> - Using **function secret sharing** for secure computation.
> - Introducing **randomized activations or layers** to reduce predictability.
> - Implementing **anomaly detection** to monitor for malicious updates.
>
> We hope that this added discussion satisfactorily addresses the reviewer's concerns.
>
> The references are in the following response.

---

> > ### Author Response · Authors · 2025-11-08
> > **Response to Reviewer tuMr on privacy discussion (2/2)**
> >
> > ### References
> > [1] D. Pasquini, G. Ateniese, and M. Bernaschi. Unleashing the tiger: Inference attacks on split learning. In Proceedings of the 2021 ACM SIGSAC Conference on Computer and Communications Security, pp. 2113–2129, 2021.
> >
> > [2] G. Gawron and P. Stubbings. Feature space hijacking attacks against differentially private split learning. arXiv preprint arXiv:2201.04018, 2022.
> >
> > [3] E. Erdoğan, A. Küpçü, and A. E. Çiçek. Unsplit: Data-oblivious model inversion, model stealing, and label inference attacks against split learning. In Proceedings of the 21st Workshop on Privacy in the Electronic Society, pp. 115–124, 2022.
> >
> > [4] X. Xu, M. Yang, W. Yi, Z. Li, J. Wang, H. Hu, Y. Zhuang, and Y. Liu. A stealthy wrongdoer: Feature-oriented reconstruction attack against split learning. In Proceedings of the IEEE/CVF Conference on Computer Vision and Pattern Recognition, pp. 12130–12139, 2024.
> >
> > [5] P. Ye, Z. Jiang, W. Wang, B. Li, and B. Li. Feature reconstruction attacks and countermeasures of DNN training in vertical federated learning. IEEE Transactions on Dependable and Secure Computing, 2024.
> >
> > [6] B. Zeng, S. Luo, F. Yu, G. Yang, K. Zhao, and L. Wang. GAN-based data reconstruction attacks in split learning. Neural Networks, 185:107150, 2025.
> >
> > [7] J. Liu, X. Lyu, Q. Cui, and X. Tao. Similarity-based label inference attack against training and inference of split learning. IEEE Transactions on Information Forensics and Security, 19:2881–2895, 2024.
> >
> > [8] K. Zhao, X. Chuo, F. Yu, B. Zeng, Z. Pang, and L. Wang. Splitaum: Auxiliary model-based label inference attack against split learning. IEEE Transactions on Network and Service Management, 2024.
> >
> > [9] X. Zhu, X. Luo, Y. Wu, Y. Jiang, X. Xiao, and B. C. Ooi. Passive inference attacks on split learning via adversarial regularization. arXiv preprint arXiv:2310.10483, 2023.
> >
> > [10] X. Wu, H. Yuan, X. Li, J. Ni, and R. Lu. Evaluating security and robustness for split federated learning against poisoning attacks. IEEE Transactions on Information Forensics and Security, 2024.
> >
> > [11] Y. He, C. Hu, Y. Pu, J. Chen, and X. Li. Advusl: Targeted adversarial attack against U-shaped split learning. In 2024 IEEE 21st International Conference on Mobile Ad-Hoc and Smart Systems (MASS), pp. 357–363. IEEE, 2024.
> >
> > [12] A. Kokaj and E. Mollakuqe. Mathematical proposal for securing split learning using homomorphic encryption and zero-knowledge proofs. Applied Sciences, 15(6):2913, 2025.
> >
> > [13] N. D. Pham, K. T. Phan, and N. Chilamkurti. Enhancing accuracy–privacy trade-off in differentially private split learning. IEEE Transactions on Emerging Topics in Computational Intelligence, 2024.
> >
> > [14] T. Khan, M. Budzys, and A. Michalas. Make split, not hijack: Preventing feature-space hijacking attacks in split learning. In Proceedings of the 29th ACM Symposium on Access Control Models and Technologies, pp. 19–30, 2024.
> >
> > [15] Y. Mao, Z. Xin, Z. Li, J. Hong, Q. Yang, and S. Zhong. Secure split learning against property inference, data reconstruction, and feature space hijacking attacks. In European Symposium on Research in Computer Security, pp. 23–43. Springer, 2023.
> >
> > [16] E. Erdoğan, U. Tekşen, M. S. Çeliktenyıldız, A. Küpçü, and A. E. Çiçek. Splitout: Out-of-the-box training-hijacking detection in split learning via outlier detection. In International Conference on Cryptology and Network Security, pp. 118–142. Springer, 2024.
> >
> > [17] K. Gamal, A. Gaber, and H. Amer. Federated learning based multilingual emoji prediction in clean and attack scenarios. arXiv preprint arXiv:2304.01005 (2023).

---

> ### Author Response · Authors · 2025-11-08
> **Response to Reviewer tuMr on comparison with other generative models**
>
> **Regarding the mentioned requested change:** *Comparison with diffusion or transformer-based generative models: Even a brief discussion of applicability would broaden the paper’s scope.*
>
> ### Response
>
> We thank the reviewer for this insightful suggestion. We initially chose **GANs** as the foundational model for our study because their dual-network architecture (**generator** and **discriminator**) presents a higher level of structural complexity, which makes them a compelling test case for **distributed** and **privacy-preserving frameworks**. The need to coordinate two interdependent networks provides a natural challenge for **multi-device computation** and **cut-point selection**, allowing us to rigorously evaluate how our framework manages both **privacy** and **computational distribution**.
>
> However, we fully agree that a discussion on the applicability of our framework to other generative architectures—such as **transformers**, **diffusion models**, and **large language models (LLMs)**—is essential to demonstrate its **generalizability** and broader potential.
>
> In the revised manuscript, we have added **Subsection 7.3** in the **Discussion** to elaborate on how **HuSCF-GAN** can be adapted to these models. While our current work focuses on GANs, the underlying principles of **HuSCF-GAN**—keeping **inputs, outputs, and labels strictly on the client side** to preserve **data privacy**, while distributing **computational load** according to client capabilities—can be naturally extended to other architectures. The adaptation primarily affects the **cut-point selection** and related algorithmic steps as follows:
>
> - **Cut-point selection:** For single-network models such as **encoder-only** (e.g., BERT), **decoder-only** (e.g., GPT), or **diffusion models**, the number of cut points would typically be two (one near the input and one near the output to keep the input data and their labels at the client), instead of four as in GANs. However, our framework should allow for **dynamic cut-point selection** in the future, and future work could extend this to determine the **optimal number of cut points** adaptively based on **model complexity** or **device capabilities**.
>
> - **Split learning:** This component of the framework would remain unchanged, enabling **distributed training and inference across devices** while maintaining **privacy guarantees**.
>
> - **Clustering:** The clustering step would still be applied, using **embeddings of the corresponding data type** (e.g., text, image, or multimodal embeddings), thereby retaining the same **conceptual foundation**.
>
> - **KLD divergence and federated learning:** Both these components would follow the same methodology, ensuring consistent **alignment and aggregation across clients** regardless of model type.
>
> Applying **HuSCF-GAN** to **LLMs** opens exciting possibilities—for instance, allowing **personal devices** such as smartphones to participate in **large-scale text model training** without exposing sensitive user data, while utilizing otherwise idle **computational resources**. Furthermore, our **clustering mechanism** could enable **domain-specific specialization**, grouping data or models by thematic areas (e.g., medical, geographic, or general knowledge).
>
> Similarly, extending the framework to **diffusion models** and **vision transformers** would enable **privacy-preserving collaborative learning** in diverse **generative scenarios**. By sharing **intermediate parameters**, these models can jointly improve performance while preserving **local specialization**. Incorporating this discussion broadens the scope of **HuSCF-GAN**, demonstrating its **adaptability beyond GANs** to a wide range of **modern generative learning architectures**.
>
> We hope that we have addressed the reviewer's concerns.

---

> ### Author Response · Authors · 2025-11-08
> **Response to Reviewer tuMr on real-world latency validation**
>
> **Regarding the mentioned requested change:** *Real-world latency validation: A small-scale physical deployment (even with 10–20 devices) would better ground the latency claims beyond simulations.*
>
> ### Response
>
> We sincerely thank the reviewer for this insightful and constructive suggestion. Our current evaluation is based on **comprehensive simulations** involving **100 heterogeneous devices**, which we believe provide a realistic and **scalable representation of real-world distributed environments**. This simulation framework was carefully designed to capture a wide range of practical factors, including **heterogeneous device capabilities**, **varying network latencies**, and **computational capacities**, thereby offering a reliable approximation of real deployment scenarios.
>
> We fully acknowledge, however, that **simulation-based studies** cannot completely substitute for the richness and unpredictability of an **actual physical deployment**. Conducting a **real-world test-bed experiment** would undoubtedly strengthen the **empirical grounding** of our latency and performance claims. Unfortunately, due to **current resource and infrastructure constraints** and **limited compute resources**, we were unable to perform such physical experiments within the scope of this work.
>
> To address this important point, we have revised the manuscript to explicitly discuss this **limitation** and have included a statement in the **Future Work** section emphasizing that **developing and evaluating a real test-bed deployment** constitutes a **valuable and natural next step** to further validate and refine our system’s **real-world performance**.

---

> ### Author Response · Authors · 2025-11-08
> **Response to Reviewer tuMr on ablation on GA hyperparameters**
>
> **Regarding the mentioned requested change:** *Ablation on GA hyperparameters: Sensitivity analysis for population size, mutation/crossover rates, and $\beta$ (in weighting scheme) would provide more robustness evidence.*
>
> ### Response
>
> We thank the reviewer for this helpful suggestion. In the revised manuscript, we have added a detailed **ablation study** on **genetic algorithm (GA) hyperparameters** in **Appendix B**. In this study, we investigate the impact of three key GA hyperparameters—**Population Size**, **Crossover Rate**, and **Mutation Rate**—by varying each individually while keeping the others fixed at their baseline values.
>
> Specifically:
> - **Population Size (PS)** determines the number of candidate solutions (possible cutpoints per all clients) maintained in each generation.
> - **Crossover Rate (CR)** controls the probability of combining pairs of solutions to produce new offspring.
> - **Mutation Rate (MR)** defines the likelihood of introducing random changes to individual solutions to promote diversity and prevent getting stuck in a **local optima**.
>
> This approach enables us to isolate the impact of each parameter on the system’s performance and assess their relative significance. Since the GA is responsible for determining the **model cut points**, it directly affects **system latency**, which serves as the primary evaluation metric in this analysis.
>
> The selected parameter ranges were chosen based on their inherent characteristics:
> - **Population sizes**: 500 to 2000, capturing effects of smaller vs. larger candidate solution sets on performance.
> - **Crossover rates**: 0.3 to 0.9, evaluating the influence of low to high rates of producing new offsprings from parent solutions.
> - **Mutation rates**: 0.001 to 0.1, assessing the impact of introducing minimal to moderate variance into the search process..
>
> The results of this study are summarized in the table below (**Table 24** in the revised manuscript), reporting **system latency** for different hyperparameter configurations:
>
> | GA Hyperparameters | PS=1000, CR=0.7, MR=0.01 | PS=1000, CR=0.3, MR=0.01 | PS=1000, CR=0.5, MR=0.01 | PS=1000, CR=0.9, MR=0.01 | PS=1000, CR=0.7, MR=0.001 | PS=1000, CR=0.7, MR=0.05 | PS=1000, CR=0.7, MR=0.1 | PS=100, CR=0.7, MR=0.01 | PS=500, CR=0.7, MR=0.01 | PS=2000, CR=0.7, MR=0.01 |
> |------------------|--------------------------|--------------------------|--------------------------|--------------------------|----------------------------|--------------------------|------------------------|--------------------------|--------------------------|---------------------------|
> | **Latency (s)**  | **7.8**                  | 8.21                     | **7.8**                  | **7.8**                  | **7.8**                    | 8.3                      | 9.7                    | 8.22                     | 7.9                      | 8.13                     |
>
> Our findings show that while variations in GA hyperparameters have minimal impact on other performance metrics, their effect on **latency** is both noticeable and significant, highlighting the importance of careful **tuning** in **latency-sensitive scenarios**. The lowest latency value achieved across all configurations was **7.8 seconds**, corresponding to the hyperparameter settings used in our main experiments.
>
> Key observations:
> - **Reducing Population Size** (e.g., to 500 or lower) deteriorates the objective function (latency) due to limited diversity of candidate solutions.
> - **Excessively increasing Population Size** (e.g., beyond 2000) yields negligible improvement in latency while substantially increasing **computational complexity**.
> - **Low Crossover Rates** (0.3 or lower) negatively impact latency by limiting diversity introduced through recombination, while excessively high rates (0.9 or higher) add unnecessary computational overhead.
> - **Mutation Rate:** Higher values (≥0.1) degrade performance due to excessive randomness; very low values (≤0.001) increase the risk of premature convergence to **local optima**.
>
> Overall, these findings confirm the importance of maintaining a **balance between efficiency and solution quality**, achieved through the selected hyperparameter configurations.

---

> ### Author Response · Authors · 2025-11-08
> **Response to Reviewer tuMr on Broader Impact Concern and Additional comments**
>
> **Regarding the mentioned Broader Impact Concerns:**
>
> We thank the reviewer for this valuable recommendation. We agree that explicitly discussing the potential risks and mitigation strategies in the **Broader Impact** section is important for responsible dissemination of our work.
>
> In the revised manuscript, we have expanded the **Broader Impact** subsection to clearly address both the benefits and risks associated with deploying **HuSCF-GAN**. Specifically, we now highlight that while the framework enables distributed generative AI across heterogeneous devices and privacy-constrained environments, there is a risk that shared activations or gradients could be exploited to reconstruct sensitive input data. To mitigate these risks, we emphasize the need for careful deployment and recommend integrating privacy-preserving techniques such as **homomorphic encryption, function secret sharing, differential privacy, and zero-knowledge proofs (ZKPs)**.
>
> We believe that these additions make the **Broader Impact** section more comprehensive and balanced, and we hope that this revision satisfactorily addresses the reviewer’s concern.
>
> **Regarding the additional comment:** *The paper is well-written and structured, though dense in equations. Adding more intuitive diagrams (e.g., GA optimization process, communication flow) would improve accessibility.*
>
> ### Response
>
> We thank the reviewer for this helpful suggestion. In response, we have added **Figure 5, Figure 6, and Figure 7** to enhance the intuitive understanding of the proposed approach. **Figure 5** presents a flow diagram illustrating the key components of the framework and their interactions, while **Figure 6** depicts the reduction strategy applied to the **Genetic Algorithm** search space. **Figure 7** represents the operation of **Heterogenous U shaped Split learning**. These additions aim to make the methodology more accessible and visually clear.
>
> **Regarding the additional comment:** *The results section is strong; however, including confidence intervals or statistical significance testing would further substantiate performance claims.*
>
> ### Response
>
> We thank the reviewer for this valuable suggestion. We agree that including measures of statistical confidence strengthens the reliability of the reported results. Accordingly, in the revised manuscript, we have added **95% Wald confidence intervals** to all performance metrics presented in the **Results** section and corresponding tables. These additions provide a clearer indication of the statistical robustness and consistency of our findings across multiple experimental runs. These confidence intervals remained consistent with our previously shown results. We believe this enhancement further substantiates our performance claims and improves the overall rigor of the results.
>
> We hope that this revision satisfactorily addresses the reviewer’s comment.
>
> **Regarding the additional comment:** *Future work suggestions are thoughtful; in particular, exploring applicability to diffusion models and LLMs is an exciting direction that would extend the contribution substantially.*
>
> ### Response
>
> We sincerely thank the reviewer for their kind words. We agree that exploring the applicability of our approach to **diffusion models** and **LLMs** is an exciting direction, which is why we included these points in our **Future Work** section.

---

### Review · Reviewer_4ec9 · 2025-09-15

**Summary Of Contributions:**

This paper addresses a complex set of challenges in training generative AI models, specifically GANs, in distributed environments. The authors identify that real-world distributed learning is often constrained by data heterogeneity (non-IID), device heterogeneity (varying compute and communication capabilities), multi-domain data sources, and strict data-sharing and privacy constraints.

To tackle these challenges, the paper proposes a novel distributed generative AI framework named HuSCF-GAN. This framework integrates several techniques in a cohesive manner:

1. To handle device heterogeneity, the model is dynamically split between the client and a server. The system employs a genetic algorithm to determine the optimal "cut points" for each client to minimize overall training latency. The U-shaped architecture ensures that raw data and labels remain on the client, enhancing privacy

2. To adapt to multi-domain data, the server clusters clients based on the activation similarity from an intermediate layer of the discriminator, grouping clients with similar data distributions.

3. To address data heterogeneity, the federated aggregation within each cluster is weighted not only by dataset size but also by a Kullback-Leibler Divergence (KLD) score calculated from activations. This approach avoids the need to share label distributions, further preserving privacy.

**Weaknesses**

1. The paper acknowledges that the search space for the genetic algorithm grows exponentially with the number of clients and uses a client down-sampling approach to simplify the optimization. While validated on a small scale, the effectiveness and sub-optimality gap of this approximation in very large-scale systems (e.g., thousands of clients) warrants further discussion.

2. Although the approach offloads computation from clients, it still relies on a central server for model splitting, clustering, and federated aggregation. This positions it as a federated, not a fully decentralized, topology, a nuance that could be discussed more explicitly.

**Audience:**

Yes

**Audience Explanation:**

Yes, the target audience for this paper is quite broad, as it sits at the intersection of several popular and active research areas, including generative AI, federated learning, edge computing, and privacy-preserving machine learning.

**Broader Impact Concerns:**

N.A.

**Claims And Evidence:**

Yes

**Claims Explanation:**

1. The experiments, conducted across six scenarios of increasing complexity, systematically demonstrate that the proposed method outperforms or significantly outperforms multiple baselines on image generation scores and classification metrics. The performance advantage is particularly evident in complex multi-domain, non-IID settings.

2. The latency comparison clearly shows that the method achieves an order-of-magnitude latency reduction compared to approaches that train the full model on the client, such as FedGAN and PFL-GAN, which is critical for resource-constrained edge devices.

3. The design of the U-shaped split learning architecture and the activation-based KLD calculation mechanistically ensure that raw data and labels do not leave the client, providing clear evidence for the privacy-preserving aspects of the framework.

**Requested Changes:**

1. I would recommend the authors add a brief discussion on the potential challenges of implementing such a complex system (combining a genetic algorithm, clustering, and federated learning), such as hyperparameter sensitivity and the overhead of coordinating the different components.

2. It would be beneficial to elaborate on the approximation strategy used for the genetic algorithm. For instance, how does the down-sampling ratio affect the quality of the final solution? Can any guarantees be made about the proximity to an optimal solution in large-scale client scenarios?

3. The experiments are based on MNIST and its variants, which is appropriate for a proof-of-concept. Acknowledging this as a limitation in the conclusion and speculating on the potential performance and challenges on more complex, real-world datasets (e.g., medical images, autonomous driving scenes) would be a valuable addition.

---

> ### Author Response · Authors · 2025-11-08
> **Response to Reviewer 4ec9 on the reduction strategy (1/3)**
>
> **Regarding the mentioned weakness:** *The paper acknowledges that the search space for the genetic algorithm grows exponentially with the number of clients and uses a client down-sampling approach to simplify the optimization. While validated on a small scale, the effectiveness and sub-optimality gap of this approximation in very large-scale systems (e.g., thousands of clients) warrants further discussion.*
>
> **the requested change:** *It would be beneficial to elaborate on the approximation strategy used for the genetic algorithm. For instance, how does the down-sampling ratio affect the quality of the final solution? Can any guarantees be made about the proximity to an optimal solution in large-scale client scenarios?*
>
> We thank the reviewer for the insightful comment and the time invested in evaluating our work. We fully agree that the approximation strategy used for the **Genetic Algorithm (GA)** required further elaboration and clarification. Accordingly, in the revised manuscript, we have expanded our explanation and refined the method to improve **scalability** and clarify its foundation.
> Below, we first motivate the use of a **GA** over a full exhaustive search, then discuss the **scalability limitations** of the GA in its full search space, and finally present our **profile-based reduction strategy** along with its guarantees and assumptions.
>
> **Motivation for Using a Genetic Algorithm.**
>
> The **cutpoint selection optimization problem** in our system is inherently combinatorial and computationally intractable to solve via exhaustive search, as the search space grows exponentially with both the number of clients and the number of layers. To make this problem tractable, we employ a **Genetic Algorithm (GA)**, which offers an efficient heuristic approach capable of exploring large solution spaces and finding **near-optimal configurations** with less computational effort than a full search.
>
> **Challenge of GA Scalability.**
>
> The challenge is that the problem space still expands exponentially with the number of clients. When using the GA, we will still face **high computational cost**, **slower convergence**, and an increased risk of getting trapped in **local optima**—especially in large-scale systems involving hundreds or thousands of clients. Therefore, improving the **scalability** of the GA itself becomes essential for enabling its practical application in **large deployments**.
>
> **Profile-Based Reduction Strategy.**
>
> To overcome these limitations, we revised our **down-sampling approach** into a **profile-based reduction strategy**. Instead of using a fixed client sampling ratio, we now reduce the **GA search space** even further to the number of **unique device profiles** (refer to **Figure 6** and **subsection 4.3** in the revised manuscript), where each profile groups clients that share identical **computational** and **communication characteristics**.
>
> The difference between our **profile-based reduction strategy** and the previous manuscript’s **downsampling strategy** lies in how each approach reduces the **Genetic Algorithm (GA) search space**. The **downsampling strategy** mitigates complexity by sampling a smaller subset of clients according to a fixed ratio while preserving the proportional distribution of different client profiles. This method approximates the behavior of the full system but continues to operate on **individual clients**, which means the search space can still grow exponentially as the number of clients increases.
>
> In contrast, the **profile-based reduction strategy** introduces a **structural** rather than **quantitative simplification**. Instead of sampling clients, it aggregates all clients that share identical **computational** and **communication characteristics** into **unique profiles**. The GA then performs optimization over these **profiles** instead of individual clients, effectively reducing the search space from the total number of clients to the number of distinct **profiles**. This approach achieves a substantially greater reduction in **computational cost** while preserving the **diversity** and **representativeness** of client behaviors in the optimization process. Moreover, unlike client-based or downsampling approaches, the **profile-based strategy** is inherently resistant to **exponential growth** in the search space, since it scales with the number of **profiles**—typically far smaller than the total number of devices, even in **large-scale deployments** where clients can always be clustered into a limited set of **representative profiles**.
>
> For completeness, it is worth noting that the **client-based approach** serves as the **baseline configuration**, where the GA operates over the full set of clients without any reduction. Each client is treated as an **independent entity** within the search space, ensuring maximum **fidelity** to the original system but resulting in the highest **computational complexity** among all three strategies.

---

> ### Author Response · Authors · 2025-11-08
> **Response to Reviewer 4ec9 on the reduction strategy (2/3)**
>
> The table below presents a comparison of the **Genetic Algorithm (GA)** performance under the different strategies implemented in our system using **100 devices**. The results clearly demonstrate that the **profile-based reduction strategy** achieved the **lowest latency** within the **smallest number of generations**, confirming its **efficiency** and **fast convergence**. In contrast, the **downsampling strategy** required approximately **five times more generations** to reach the same latency level. Finally, the **client-based approach** exhibited a significantly higher latency of **8.26** and required nearly **forty times more generations**. This outcome highlights the **high computational complexity** of the client-based method, where the increased search space leads to **slower convergence** and a higher likelihood of the algorithm becoming trapped in **local optima**, rather than indicating the inaccessibility of lower-latency solutions.
>
> | Strategy                  | Achieved Latency (s) | Number of generations till convergence |
> |----------------------------|--------------------|---------------------------------------|
> | **Profile-based**          | 7.8                | 12                                    |
> | **Downsampling (scale = 4)** | 7.8                | 60                                    |
> | **Client-based**           | 8.26               | 488                                   |
>
> ---
>
> ### **Impact on Algorithm Complexity**
>
> As the number of clients increases, the **computational complexity** of the **profile-based reduction strategy** remains largely unaffected because it scales with the **number of profiles**, not the total number of clients. In this approach, the **Genetic Algorithm (GA)** operates on a **reduced search space** defined by the **unique device profiles** rather than individual clients. Therefore, the overall **algorithmic complexity decreases** relative to any client-based optimization method and only increases if the number of **distinct profiles** grows.
>
> Although our system assumes that clients within the same profile share identical **computational** and **communication capabilities**, this assumption is for a **Proof of Concept**. In realistic **large-scale deployments**, even as the total number of devices grows substantially, these devices can still be effectively clustered into a set of **representative profiles** based on shared characteristics. Consequently, the number of profiles typically remains much smaller than the number of clients, ensuring that the **profile-reduced GA** remains highly **scalable** and **computationally efficient**, even in very large networks.
>
> ---
>
> ### **Impact on Solution Quality**
>
> As the number of clients scales, the **profile-based reduction strategy** consistently maintains — and in some cases improves — the **quality of the solution** while significantly reducing **computational complexity**. Since the **Genetic Algorithm (GA)** operates over a **reduced search space** defined by **profiles** rather than individual clients, it achieves **faster** and **more stable convergence** while maintaining reasonable quality.
>
> Experimental results with **100 devices** demonstrate this effect clearly. The **profile-based strategy** not only reached a **good latency** in far fewer generations than both the **downsampling** and **client-based approaches**, but it also achieved a **lower final latency** than the **client-based configuration**. This improvement arises from the **smaller and more structured search space**, which enables the GA to explore solutions more effectively and **avoid getting trapped in local optima**—a common issue in large, complex search spaces.
>
> ---
>
> ### **Effect of System Synchronicity**
>
> If the proposed system operates in a **synchronous setting**, in which the server waits for all clients participating in a given layer—effectively waiting for the **slowest client**—before proceeding to the next layer, the **profile-based reduction strategy** maintains **reachability** to the **best latency achievable** by the client-based GA. This is because any configuration change that could reduce overall latency either (i) provides no net benefit when applied to an individual client in a synchronous context, or (ii) can be equivalently represented at the **profile level**, since clients within a profile share identical **computational** and **communication characteristics**.
>
> In contrast, if the system were to operate under an **asynchronous setting**—where the server does not wait for slower clients—the **profile-based strategy** might lose **reachability** to certain better solutions that the client-based GA could represent. However, even in such a case, it would still deliver **competitive latency** while maintaining significantly lower **computational complexity**.

---

> ### Author Response · Authors · 2025-11-08
> **Response to Reviewer 4ec9 on the reduction strategy (3/3)**
>
> ### **Effect of Client Profiling Assumptions**
>
> The effect of **client profiling** depends on the assumptions made about the **similarity of clients** within each profile. In our current system, we assume that all clients within a profile have **identical computational and communication capabilities**. Under this assumption, there is **no loss of information**, and the **profile-based reduction strategy** can reach the same set of solutions as the **client-based GA**, maintaining **solution quality** and **reachability**.
>
> In more realistic **large-scale scenarios**, however, this assumption can be relaxed by clustering clients with **similar** (rather than identical) capabilities into **common profiles**. In this case, some **loss of information** is introduced, as each profile represents an **averaged** or **generalized version** of the clients it contains. While this may slightly affect the **reachability** to some better solutions achievable by the **client-based GA**, it is still expected to yield **high-quality results** with substantially lower **computational complexity** and **execution time**, making it a **practical** and **scalable approach** for **large deployments**.
>
> We hope this expanded explanation and the accompanying elaboration (**Appendix D**) satisfactorily address the reviewer’s concerns regarding the **scalability**, **quality**, and **practical validity** of our **reduction strategy**.

---

> ### Author Response · Authors · 2025-11-08
> **Response to Reviewer 4ec9 on reliance on a central server**
>
> **Regarding the mentioned weakness:**  *Although the approach offloads computation from clients, it still relies on a central server for model splitting, clustering, and federated aggregation. This positions it as a federated, not a fully decentralized, topology, a nuance that could be discussed more explicitly.*
>
> ### Response:
>
> We thank the reviewer for this insightful comment, which has helped us refine and clarify our contribution. Our objective was not to design a fully decentralized system, but rather to leverage underutilized computational resources while minimizing overall system latency. To achieve this, we incorporate a central server that coordinates and facilitates these objectives.
>
> While we agree with the reviewer’s observation, our use of the term distributed is consistent with established literature, where several studies have described architectures that employ a central server—yet distribute data or portions of computation across multiple devices through federated or split learning—as distributed or decentralized systems [1,2,3]. In such contexts, the distributed nature arises from the allocation of data, computation, or both across multiple devices.
>
> That said, we fully acknowledge the reviewer’s concern that our proposed architecture is not fully centralized in the sense of relying on a central server model splitting, clustering, and aggregation. We agree that this nuance should be explicitly highlighted as both a limitation and a direction for future research.
>
> Accordingly, in the revised manuscript we have:
> - Added a Discussion section (**Section 7**) with a Limitations **subsection (7.1)**, where this point is explicitly addressed. The following paragraph has been included:
>
>      *“With respect to our approach, several limitations should be acknowledged. First, although HuSCF-GAN leverages heterogeneous and underutilized devices, it does not operate in a fully decentralized manner. The framework still relies on a central server to support these devices and to handle critical tasks such as clustering, federation, and the execution of genetic algorithms. The server is also responsible for performing the forward and backward pass computations for the server-side layers.”*
>
>
>
> - Made this limitation more visible in the **Conclusion and Future Work section (Section 8)**, where we had already noted that a promising research direction is distributing the model across multiple edge devices without relying on a central server. This section now emphasizes more clearly that achieving a fully decentralized framework—by dynamically selecting cut points and fully leveraging underutilized devices—is a key direction for extending this work.
>
>
>
> By adding this clarification, We hope these revisions clarify the distinction between **HuSCF-GAN’s** current federated topology and a fully decentralized architecture, and that this addresses the reviewer’s concern satisfactorily.
>
> [1] Thapa, Chandra, et al. "Splitfed: When federated learning meets split learning." Proceedings of the AAAI conference on artificial intelligence. Vol. 36. No. 8. 2022.
>
> [2] Radovič, Boris, et al. "Towards a Unified Framework for Split Learning." Proceedings of the 5th Workshop on Machine Learning and Systems. 2025.
>
> [3] Yurdem, Betul, et al. "Federated learning: Overview, strategies, applications, tools and future directions." Heliyon 10.19 (2024).

---

> ### Author Response · Authors · 2025-11-08
> **Response to Reviewer 4ec9 on discussion on the potential challenges**
>
> **Regarding the mentioned requested change:** *I would recommend the authors add a brief discussion on the potential challenges of implementing such a complex system (combining a genetic algorithm, clustering, and federated learning), such as hyperparameter sensitivity and the overhead of coordinating the different components.*
>
> ### Response
>
> We thank the reviewer for this insightful and constructive suggestion. We would like to clarify that the integration of multiple components—namely the **Genetic Algorithm (GA)**, **Federated Learning**, **U-shaped Split Learning**, **Kullback–Leibler Divergence (KLD)** calculations, and **clustering**—was a deliberate design choice aimed at addressing a broad range of real-world challenges. These include **data heterogeneity, device heterogeneity, multi-domain learning, resource-constrained environments**, and **strict privacy requirements**. Tackling all these aspects simultaneously necessitates a multi-component architecture such as **HuSCF-GAN**.
>
> We acknowledge, however, that combining these diverse techniques introduces additional **system complexity**. In particular, the process of selecting **optimal cut points using the Genetic Algorithm** may lead to **computational overhead**. Nevertheless, this optimization is performed only once at the beginning of the algorithm, and thus its impact is negligible compared to the potential **latency** that would arise on end devices if the cut points were not selected appropriately.
>
> In response to the reviewer’s comment, we have expanded the **Limitations** subsection (7.1) to explicitly discuss these **implementation challenges** such as **system complexity, scalability, and hyperparameter tuning**. Furthermore, we conducted an **ablation study on the GA hyperparameters** in **Appendix B** to assess sensitivity across **crossover rate, mutation rate, and population size**—parameters that respectively control how genetic information is exchanged between candidate solutions, how random variations are introduced to maintain diversity, and how many candidate solutions are maintained per generation. Our experiments demonstrated that the chosen configuration yielded the **lowest latency setup**, confirming the **robustness of our design** (for further details and quantitative results, please refer to  our [Response to Reviewer tuMr on ablation on GA hyperparameters](https://openreview.net/forum?id=rpbL7pfPYH&noteId=RRCLRwDPaO)).
>
> We hope that these revisions satisfactorily address the reviewer’s concerns.

---

> ### Author Response · Authors · 2025-11-08
> **Response to Reviewer 4ec9 on experimenting on more complex datasets**
>
> **Regarding the mentioned requested change:** *The experiments are based on MNIST and its variants, which is appropriate for a proof-of-concept. Acknowledging this as a limitation in the conclusion and speculating on the potential performance and challenges on more complex, real-world datasets (e.g., medical images, autonomous driving scenes) would be a valuable addition.*
>
> ### Response
>
> We sincerely thank the reviewer for this valuable and insightful comment. As correctly noted, our initial experiments were conducted on **MNIST** and its variants, serving as a controlled proof-of-concept to validate the core **efficiency** and **stability** of the proposed approach. We fully acknowledge that this represents a **limitation**.
>
> To address this concern more concretely, we have extended our experimental evaluation beyond the **MNIST** family, as presented in **Results Subsubsections 6.1.7 and 6.1.8**. Specifically, we now include experiments on **CIFAR-10/SVHN** (non-IID vision domains) and **BloodMNIST/DermaMNIST** (medical imaging domains). Our method achieves up to **70× lower FID scores** and **20–60% higher classification accuracy** on the former, and **1.2×–3× higher generation scores** with **9–70% accuracy improvements** on the latter, thereby demonstrating strong **scalability** and **generalization capability** across diverse and complex domains.
>
> Furthermore, we have revised the **Conclusion** and **Future Work Section 8** to explicitly outline directions for expanding this research, including:
>
> - **Evaluating the framework on more complex datasets** (higher-resolution images) and diverse modalities (e.g., time-series, text, and 3D objects).
> - **Investigating alternative generative evaluation metrics** tailored to different data modalities.
>
> For additional details and comprehensive experimental results, we kindly refer the reviewer to our  [Response to Reviewer TuMr on evaluation on Larger Scale Datasets or Medical Imaging](https://openreview.net/forum?id=rpbL7pfPYH&noteId=p5NA2iHudn).
>
> We hope these additions satisfactorily address the concern.

---

### Review · Reviewer_EDsn · 2025-10-26

**Summary Of Contributions:**

The paper studies an important problem of federated learning with possible insufficient computational costs. The authors propose a decentralized GAN training algorithm.

**Additional Comments:**

NA

**Audience:**

Yes

**Audience Explanation:**

Federated learning is a popular research area in machine learning.

**Claims And Evidence:**

Yes

**Claims Explanation:**

1. The paper is well-written and easy to follow.
2. The method is sound and the figure is clear.
3. Experiments show the effectiveness of the proposed method.

**Requested Changes:**

1. It would be better to include more experiments with more different datasets, such as time series and audio.
2. It would be better to provide a case study to intuitively show the effectiveness of the proposals.
3. It is suggested to provide computational complexity analysis, i.e., space and time complexities.

---

> ### Author Response · Authors · 2025-11-08
> **Response to Reviewer EDsn on experimenting with more different datasets**
>
> **Regarding the mentioned requested change:** *It would be better to include more experiments with more different datasets, such as time series and audio.*
>
> ### Response
>
> We sincerely thank the reviewer for this valuable suggestion. As noted, our original experiments were designed as a **proof of concept** to validate the core **efficiency** and **stability** of our proposed framework. We fully agree that extending the evaluation to additional **data modalities**—such as **audio** and **time series**—is essential for a more comprehensive demonstration of **generalizability**.
>
> To this end, we have incorporated an additional experimental scenario exploring a different **data modality (audio)**, as presented in **Results Subsubsection 6.1.9**. Specifically, we utilize the **AudioMNIST dataset**, which contains **spoken digits from zero to nine**, under a **non-IID federated configuration** involving **100 clients** with varying label exclusions. The audio files are **preprocessed into 128-bin mel-spectrograms**, converted to **log-scaled power representations**, and resized to **28×28 grayscale images** to align with our **convolution-based model architecture**.
>
> **HuSCF-GAN** outperforms competing approaches with improvements ranging from **1.5% to 13%** across all **classification metrics**, proving the **adaptability of HuSCF-GAN** to different modalities. It reached a **98% accuracy** for a classifier trained on **synthetic generated data only**.
>
> While this setup effectively demonstrates the feasibility of extending our framework to the **audio domain**, we acknowledge that the **generated audio quality** remains limited due to the **low-resolution input representation**. Moreover, we recognize the importance of employing **dedicated audio evaluation metrics** (e.g., perceptual or spectral similarity measures) to more accurately assess **generative quality** and employing **dedicated audio model architectures** to get better results. We have explicitly noted this and identified it as an important direction for **future work**.
>
> In addition, to achieve **higher fidelity** in audio generation, **modality-specific architectures**—such as **audio transformers** or **waveform-based generative models**—would be more appropriate and are part of our intended **future exploration**.
>
> Finally, we would also like to highlight that we extended our evaluation to **higher-resolution visual datasets** (**CIFAR-10/SVHN**) and **medical imaging datasets** (**BloodMNIST/DermaMNIST**), as detailed in **Results Subsubsections 6.1.7 and 6.1.8**, respectively. These results further demonstrate the **scalability** and **adaptability** of our framework across diverse domains. For more details, we kindly refer the reviewer to our [Response to Reviewer tuMr on Evaluation on Larger Scale Datasets or Medical Imaging](https://openreview.net/forum?id=rpbL7pfPYH&noteId=p5NA2iHudn).
>
> We hope these additions satisfactorily address the concern and illustrate the **cross-modal potential** of our proposed approach.

---

> ### Author Response · Authors · 2025-11-08
> **Response to Reviewer EDsn on providing a case study to show the effectiveness of the proposals**
>
> **Regarding the mentioned requested change:** *It would be better to provide a case study to intuitively show the effectiveness of the proposals.*
>
> ### Response
>
> We thank the reviewer for this insightful comment. Indeed, the rapid industrial adoption of **AI assistants** and **copilots**—such as **Microsoft Copilot, Google Assistant**, and **OpenAI’s ChatGPT-based copilots**—illustrates a clear trend toward **distributed, intelligent systems** that operate across **heterogeneous edge devices**. Many of these assistants are equipped with **onboard cameras and sensors** to interpret and generate **visual data** in real-world settings. In such scenarios, each assistant operates in a **distinct environment** (e.g., home, industrial, or outdoor) and collects **sensitive, domain-specific data** that cannot be centralized due to **privacy** or **bandwidth constraints**. Our proposed **HuSCF-GAN** framework naturally fits this emerging paradigm: it enables these **distributed assistants** to **collaboratively train generative models** while keeping **data local**, efficiently handling **heterogeneous device capabilities** and **split points**. This connection underscores both the **practical relevance** and the **near-term applicability** of our approach in **real-world AI assistant ecosystems**.
>
> The **Illustrative Use Case** (which we included in the revised manuscript in **Subsection 7.4** in **Discussion Section 7**) demonstrates how **HuSCF-GAN** effectively addresses several **real-world challenges**:
>
> - It enables assistants to utilize their existing **onboard computational resources** for **federated training**, reducing reliance on **centralized cloud infrastructure**.
>
> - It handles **device and data heterogeneity** by adaptively balancing **training contributions** based on each assistant’s **computational capacity** and available **image data**.
>
> - It **clusters visual domains** (e.g., indoor vs.\ outdoor environments) to support both **domain-specific** and **cross-domain image generation and understanding**.
>
> - It preserves **data privacy**, as **no raw images** are shared between assistants, only **learned model parameters**.
>
> We hope this addition offers a **clear and intuitive demonstration** of how **HuSCF-GAN** can be applied in **realistic multi-assistant environments**. While this represents just one possible use case, we believe the framework holds potential for many other **applications** and **extensions**. To encourage further exploration and innovation, we plan to **open-source our code**, enabling the **research community** to build upon and expand our work.

---

> ### Author Response · Authors · 2025-11-08
> **Response to Reviewer EDsn on providing complexity analysis (1/2)**
>
> **Regarding the mentioned requested change:** *It is suggested to provide computational complexity analysis, i.e., space and time complexities.*
>
> ### Response
>
> We sincerely thank the reviewer for this valuable suggestion. We agree that a detailed analysis of **computational** and **space complexity** is essential for assessing the **efficiency** and **scalability** of the proposed framework. While the computational analysis was originally included in the **Appendix**, we acknowledge that it may not have been sufficiently visible and could benefit from clearer presentation, Also, the **Space complexity analysis** wasn't included.
>
> To address this, we have moved and expanded the analysis into a dedicated subsection (**Subsection 6.4, “Complexity Analysis”**) within the **Results section**. This new subsection now explicitly presents both **computational (time)** and **space complexities** for the proposed **HuSCF-GAN** and the baseline **FedGAN**, separately discussing **client-side** and **server-side** costs.
>
> A summary of the key results is as follows:
>
> ### **Computational (Time) Complexity**
>
> **Client-side:**
> - **HuSCF-GAN** partitions the **generator** and **discriminator**, so each client only processes its **local segments**—reducing computation substantially compared to **FedGAN**, where clients handle the entire model.
> - HuSCF-GAN: $ \\mathcal{O}\\left( \\underbrace{Ib\\left(n_{G_{H,k}} + n_{G_{T,k}} + n_{D_{H,k}} + n_{D_{T,k}}\\right)}_{\\text{\\small Client-side segments training iteration}} \\right) $
>
> Where $n_{G_{H,k}}$ , $n_{G_{T,k}}$ , $n_{D_{H,k}}$ , and $n_{D_{T,k}}$ denote the number of parameters in the generator and discriminator **head** and **tail** respectively for client $k$. $I$ represents the total number of **iterations**, and $b$ represents the **batch size**.
>
> - FedGAN: $ \\mathcal{O}\\left( \\underbrace{Ib\\left(n_{G} + n_{D}\\right)}_{\\text{\\small Client models training iteration}} \\right) $
>
>   Where $n_{G}$ and $n_{D}$ denote the full number of parameters in the **generator** and **discriminator** respectively for client $k$, with $n_G = n_{G_{H}} + n_{G_{T}} + n_{G_{S}}$ and $n_D = n_{D_{H}} + n_{D_{T}} + n_{D_{S}}$.
>   Such that $n_G + n_D \\gg n_{G_H} +  n_{G_T}+ n_{D_H} +  n_{D_T}$, since the left-hand side includes the right-hand side components along with the **server parameters**, and given that the **server** in our setup is assumed to be more capable than the **client devices**, it can accommodate a larger portion of the **model workload**. Consequently, the number of **parameters stored on the server** is significantly higher than those on the clients. While client workloads may vary based on their individual capabilities, the server typically possesses substantially greater **computational resources** than most devices.
>
> This decomposition lowers **per-client computation** in our experiments, enabling training on **resource-constrained edge devices**.
>
> **Server-side:**
> - The **server** executes the **shared model segments** and **aggregation steps**. Although this increases its **computational workload** compared to **FedGAN**’s purely aggregative role, it effectively offloads heavy computation from the clients:
>
> - HuSCF-GAN:
> $\\mathcal{O} ( \\underbrace{Ib\\left(\\sum_i N_{G,i} n_{G_{S,i}} + \\sum_i N_{D,i} n_{D_{S,i}}\\right)}_{\\text{\\small Server-side training iteration}} $
>
>  &ensp; &ensp; &ensp; &ensp; &ensp; &ensp;$+ \\underbrace{MN\\left(\\max_k n_{G_{H,k}} + \\max_k n_{G_{T,k}} + \\max_k n_{D_{H,k}} + \\max_k n_{D_{T,k}}\\right)}_{\\text{\\small Client-side federated aggregation}})$
>
>   Where $n_{G_{S,i}}$ and $n_{D_{S,i}}$ represent the number of parameters of layer $i$ in the **server generator** and **discriminator segments** respectively, and $N_{G,i}$ and $N_{D,i}$ represent the number of clients participating in layer $i$ in the **server generator** and **discriminator segments** respectively. $M$ represents the number of **rounds**, and $N$ represents the total number of **clients**.
>
> - FedGAN: $ \\mathcal{O}\\left( \\underbrace{MN\\left(n_{G} + n_{D}\\right)}_{\\text{\\small Client models federated aggregation}} \\right) $
>
> This **redistribution** increases the **server computational load** while taking advantage of the server’s greater **computational capacity**.

---

> ### Author Response · Authors · 2025-11-08
> **Response to Reviewer EDsn on providing complexity analysis (2/2)**
>
> ### **Space Complexity**
>
> **Client-side:**
> Each client stores only local model segments and corresponding activations:
>
> - **HuSCF-GAN:**
> $\mathcal{O}\Big( \underbrace{n_{G_{H,k}} + n_{G_{T,k}} + n_{D_{H,k}} + n_{D_{T,k}}}_{\text{\small Parameters of client-side segments}}$
>
> &ensp;&ensp;&ensp;$+ \underbrace{b (A_{G_{H,k}} + A_{G_{T,k}} + A_{D_{H,k}} + A_{D_{T,k}})}_{\text{\small Activations of client-side segments}} \Big)$
>
> Where $A_{G_{H,k}}$, $A_{G_{T,k}}$, $A_{D_{H,k}}$, and $A_{D_{T,k}}$ denote activation sizes per sample for the generator and discriminator heads and tails respectively for client $k$.
>
> - **FedGAN:**
> $\mathcal{O}\Big( \underbrace{n_G + n_D}_{\text{\small Parameters of client models}}$
>
> &ensp;&ensp;&ensp;&ensp;$+\underbrace{b (A_G + A_D)}_{\text{\small Activations of client models}} \Big)$
>
> Where $A_G$ and $A_D$ are the full activation sizes for the generator and discriminator respectively.
> It is important to note that $A_G = A_{G_H} + A_{G_S} + A_{G_T}$ and $A_D = A_{D_H} + A_{D_S} + A_{D_T}$, such that $A_G + A_D \gg A_{G_H} + A_{G_T} + A_{D_H} + A_{D_T}$.
> Since the left-hand side includes the right-hand side activations along with the server activations, and given that the server in our setup is assumed to be more capable than the client devices, it can accommodate a larger portion of the model memory. Consequently, the activation size stored on the server is significantly higher than those on the clients. While client load may vary based on their individual capabilities, the server typically possesses substantially greater resources than most devices, thus can hold larger activations.
>
> This design reduces the client memory footprint, enabling participation of low-resource or underutilized edge devices.
>
> ---
>
> **Server-side:**
> The server stores parameters and activations for the shared model segments and maintains temporary copies of the largest client segments during aggregation:
>
> - **HuSCF-GAN:**
> $\mathcal{O}\Big( \underbrace{\sum_i (n_{G_{S,i}} + n_{D_{S,i}})}_{\text{\small Parameters of server-side segments}}$
>
> &ensp;&ensp;&ensp;&ensp;$+ \underbrace{b \sum_i (N_{G,i} A_{G_{S,i}} + N_{D,i} A_{D_{S,i}})}_{\text{\small Activations of server-side segments}}$
>
> &ensp;&ensp;&ensp;&ensp;$+ \underbrace{\max_k (n_{G_{H,k}} + n_{G_{T,k}} + n_{D_{H,k}} + n_{D_{T,k}})}_{\text{\small Activations of client-side segments for aggregation}}\Big)$
>
> &ensp;&ensp;&ensp;&ensp;$A_{G_{S,i}}$ and $A_{D_{S,i}}$ represent activation sizes for the $i$-th layer of the server generator and discriminator segments.
>
> - **FedGAN:**
> $$
> \mathcal{O}\Big( \underbrace{n_G + n_D}_{\text{\small Parameters of client models for aggregation}} \Big)
> $$
>
> This increases the server’s memory usage but allows clients to operate under tighter memory budgets—enhancing system scalability and robustness in federated environments.
>
> We hope that our complexity analysis satisfactorily addresses the reviewer’s concern.

---

### Author Response · Authors · 2025-11-08
**Changes in the Revised Manuscript**

We sincerely thank the reviewers for their valuable comments and insightful suggestions, which have greatly helped us improve the quality and clarity of our paper. We have carefully responded to the reviewers' comments, edited the manuscript accordingly, and uploaded a revised version that addresses the requested changes and incorporates the reviewers’ insights.

**Key modifications include:**

- **Additional Experiments:** We have included three new experiments to test the generalizability of our approach:
  - Higher-resolution datasets
  - Medical imaging datasets
  - Audio datasets

- **Updated Abstract and Conclusion:** Both the abstract and conclusion have been updated to reflect the results of the added experiments.

- **New Discussion Section:** We added a comprehensive discussion covering:
  - Limitations of the approach
  - Potential risks and mitigation strategies
  - Adaptability to other generative models
  - An illustrative use case

- **Ablation Study:** We included an ablation study on Genetic Algorithm (GA) hyperparameters in **Appendix B**.

- **Statistical Analysis:** We added 95% Wald confidence intervals to all reported metrics.

- **Broader Impact Section:** Expanded to explicitly cover potential risks and corresponding mitigation strategies.

- **Figures and Diagrams:** Added Figure 5, Figure 6, and Figure 7 to enhance intuitive understanding of the methodology.

- **Complexity Analysis:** Moved the computational (time) complexity analysis to the **Results** section, presented it more clearly, and added a detailed **space complexity analysis**.

- **Profile-Reduced GA:** Further elaboration on the profile-based GA approach has been added in **Appendix D**.

We hope that these revisions satisfactorily address the reviewers’ comments and improve the clarity, rigor, and accessibility of our work.

---

### Decision · Action_Editor_evbv · 2026-01-04

**Recommendation:** Accept as is

**Audience:**

Yes

**Audience Explanation:**

The paper proposes HuSCF-GAN, a new distributed generative learning framework that addresses several practical challenges such as device heterogeneity, non-IID data, and privacy constraints. This would be of interest to researchers in distributed learning and generative modeling fields.

**Claims And Evidence:**

Yes

**Claims Explanation:**

The paper presents technically solid framework for distributed generative training. There was a consensus among the reviewers that the claims made by the authors are supported by the experimental evaluations.

---

> ### Author Response · Authors · 2026-01-16
> **Camera-Ready Versiom**
>
> We thank the Action Editor and the reviewer for the decision and their valuable support and insights throughout the process, which further strengthened our work.
>
> We confirm that we have submitted our camera-ready version